# C-COMPASS: a user-friendly neural network tool profiles cell compartments at protein and lipid levels

Daniel T. Haas[1,2], Daniel Weindl [3,4], Pamela Kakimoto[1,2], Eva-Maria Trautmann [1,2], Julia P. Schessner[5], Xia Mao[6], Mathias J. Gerl [7], Maximilian Gerwien[8], Timo D. Müller[1,2,9], Christian Klose[7], Xiping Cheng[6], Jan Hasenauer [3,4] & Natalie Krahmer [1,2]✉

Systematic proteomic organelle profiling methods including protein correlation profiling and LOPIT have advanced our understanding of cellular compartmentalization. To manage the complexity of organelle profiling data, we introduce C-COMPASS, a user-friendly open-source software that employs a neural network-based regression model to predict the spatial cellular distribution of proteins. C-COMPASS handles complex multilocalization patterns and integrates protein abundance to model organelle composition changes across conditions. We apply C-COMPASS to mice with humanized livers to elucidate organelle remodeling during metabolic perturbations. Additionally, by training neural networks with co-generated marker protein profiles, C-COMPASS extends spatial profiling to lipids, overcoming the lack of organelle-specific lipid markers, allowing for determination of localization and tracking of lipid species across different compartments. This provides integrated snapshots of organelle lipid and protein compositions. Overall, C-COMPASS offers an accessible tool for multiomic studies of organelle dynamics without needing advanced computational skills, empowering researchers to explore new questions in lipidomics, proteomics and organelle biology.

Cellular compartmentalization, essential for eukaryotic cell organization, creates distinct functional modules to facilitate biochemical reactions, segregate toxic metabolites and regulate signal transduction[1,2]. This compartmentalization is dynamic, tailored to the unique functions of different cell types, and plays a crucial role in cellular differentiation[3,4]. Understanding these compartmental changes is key for elucidating cellular mechanisms and dysfunction in diseases. Proteomic characterization of organelles is complicated by challenges in achieving pure organelle isolation due to interorganelle interactions, a problem amplified by high-sensitivity proteomics that detects major portions of the proteome even in highly purified samples[5]. Advanced proteomic organelle profiling techniques such as protein correlation profiling (PCP)[6,7], localization of organelle proteins by isotope tagging (LOPIT)[8], SubCellBarCode[9] and dynamic organellar maps (DOMs)[10] have addressed these challenges. These methods are based on differential or density gradient centrifugation to separate cellular components

[1]Institute for Diabetes and Obesity, Helmholtz Diabetes Center, Helmholtz Munich, Neuherberg, Germany. [2]German Center for Diabetes Research (DZD), Neuherberg, Germany. [3]Faculty of Mathematics and Natural Sciences, University of Bonn, Bonn, Germany. [4]Institute of Computational Biology, Helmholtz Zentrum München, Neuherberg, Germany. [5]Department of Proteomics and Signal Transduction, Max Planck Institute of Biochemistry, Martinsried, Germany. [6]Regeneron Pharmaceuticals, Inc., Tarrytown, NY, USA. [7]Lipotype GmbH, Dresden, Germany. [8]Max Delbrück Center for Molecular Medicine, Berlin, Germany. [9]Walther Straub Institute of Pharmacology and Toxicology, Ludwig Maximilian University of Munich, Munich, Germany. ✉e-mail: natalie.krahmer@helmholtz-munich.de

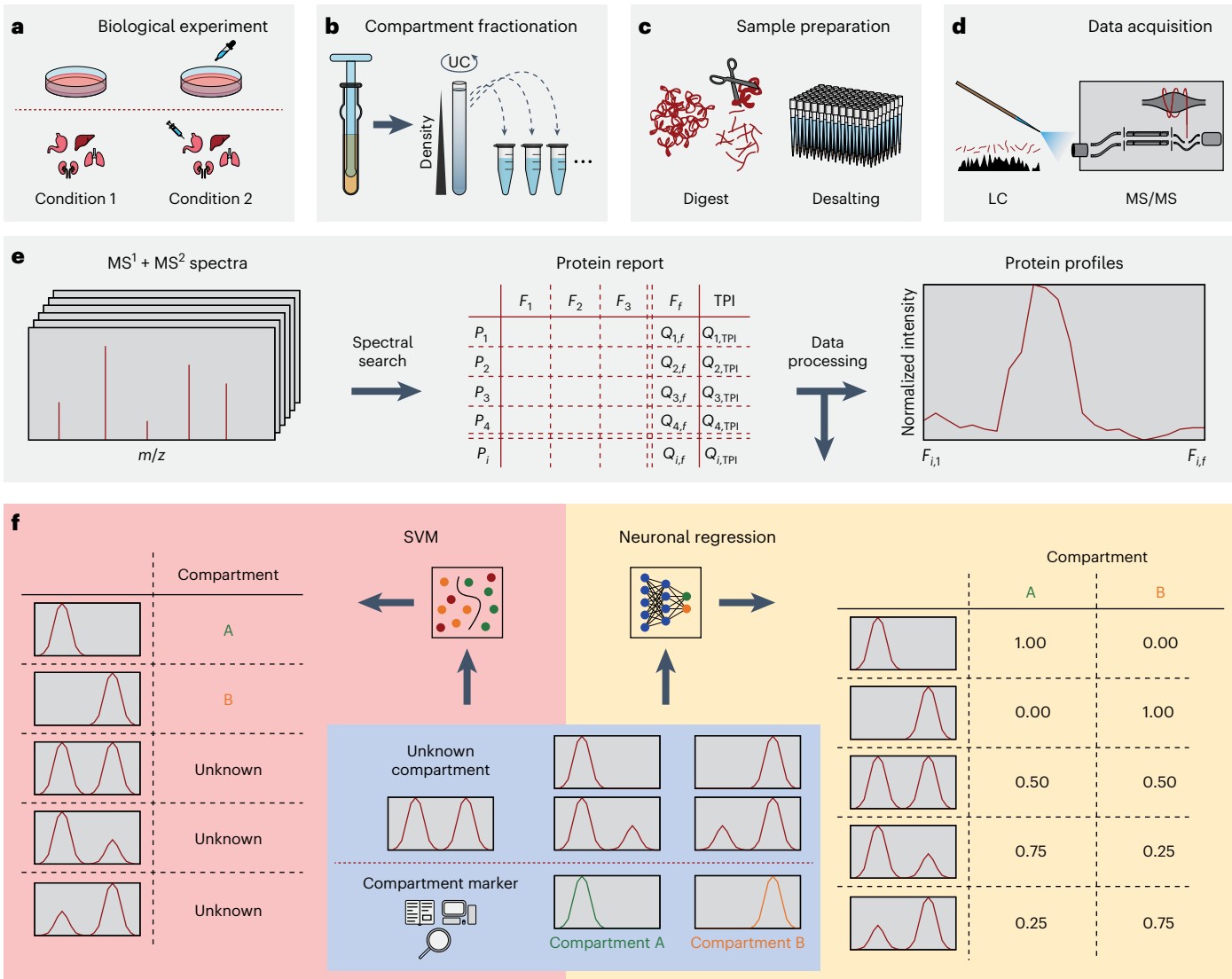

**Fig. 1 | Overview of workflow to generate organellar maps. a**, Cells or tissue samples are collected from the biological conditions of interest in biological replicates. **b**, Samples are lysed while keeping organelles intact, and compartments are separated, for example, based on density gradient ultra centrifugation (UC). **c,d**, Compartment fractions are prepared for LC–MS/MS, and proteins are quantified. **e**, Abundance information for proteins ($P_1$–$P_i$) in all organelle fractions ($F_1$–$F_f$) and total proteome information (TPI) is obtained via standard LC–MS/MS workflows. Quantitative information from pivot report tables is used to generate abundance profiles by scaling to an area of 1 or by min–max scaling across the fractions. **f**, Machine learning-based localization predictions require marker proteins specific to single compartments. Traditional methods such as SVM are suitable for binary predictions, struggling with multiply localized proteins, whereas neuronal network-based regression models can assess contributions from multiple compartments to a protein profile.

by size and density, followed by liquid chromatography tandem mass spectrometry (LC–MS/MS) to quantify protein abundance across fractions (Fig. 1a–d). This generates profiles indicative of organelle-specific protein distribution, helping to distinguish true compartmental proteins from contaminants (Fig. 1e). These techniques provide localization data for thousands of proteins across multiple compartments in a single experiment[11–13].

Streamlined cellular fractionation workflows combined with high-throughput LC–MS/MS enable mapping protein localization across biological conditions. These complex datasets contain abundance profiles for thousands of proteins, with approximately 50–60% exhibiting multiple localizations[14]. Recent computational advances, which include DOM-ABC[10] and SubCellBarCode[15] (both employing support vector machines; SVMs), MetaMass[16] (*k*-means clustering), TRANSPIRE[17] (stochastic variational Gaussian process classifier), BANDLE[18] (Bayesian framework) and TAGM-MCMC[19] (T-augmented Gaussian mixture Bayesian model with Markov chain Monte Carlo techniques),

address this complexity. While some of these tools use uncertainty or probabilistic estimations to infer multiple localizations, none of these tools predict quantitative protein distribution across compartments, integrate organelle and protein abundance data essential for analyzing organelle composition or combine proteomic and lipidomic data layers. Additionally, many tools require advanced coding skills, limiting accessibility.

Therefore, we introduce Cellular COMPartment clASSifier (C-COMPASS), an open-source software with a graphical interface that provides a reproducible pipeline for predicting proteome-wide spatial distributions from fractionation data (Fig. 1f). Unlike previous methods limited to single-organelle prediction or probabilistic outputs, C-COMPASS uses neural network-based regression to quantify protein multilocalization patterns across cellular compartments, addressing the complexity of subcellular organization. It also integrates protein and organelle abundance data from coanalyzed whole-cell proteomes to model compositional changes in compartments. Using C-COMPASS,

we generated spatial proteomic maps from humanized mouse livers across metabolic states, capturing organelle reprogramming.

Moreover, C-COMPASS extends beyond proteomics: by applying neural networks to concurrent proteomic and lipidomic data, it predicts lipid subcellular localization, enabling comprehensive mapping of organelle lipid composition and identifying organelle-specific lipid species, overcoming the limitation of the lack of established lipid markers for organelles. As a systematic tool to coanalyze the spatial distribution of proteins and lipids, C-COMPASS can be used to assess their dynamic interplay. It is a versatile, user-friendly deep learning platform that enables researchers in proteomics, lipidomics and cell biology to explore organelle dynamics across various tissues and cell types under biological or disease conditions.

## Results

### Computational workflow

C-COMPASS uses experimentally generated protein abundance profiles and is independent of LC–MS/MS acquisition methods. To annotate protein localizations from cellular fractionation data, C-COMPASS requires (1) a pivot table of identified proteins and their intensities across fractions and (2) a marker set of proteins typically localized to single compartments.

For this, we provide predefined marker lists, derived from experimentally validated datasets[20,21]. These sources are preferred over gene ontology (GO) annotations, which often overlap in content and thus provide less specific information. Our provided list (Supplementary Information) encompasses all major membrane-enclosed organelles and includes two additional categories: protein complexes and protein synthesis. Protein complexes, such as large chaperone complexes or mTOR, sediment at higher densities than soluble proteins, allowing us to resolve them. The compartment, which we named protein synthesis, can be categorized as a membraneless organelle that comprises ribosomes, mRNA-binding proteins and other components involved in translation. Alternatively, users can import custom lists, which can include both membranous and membraneless structures. Resolution depends on two factors: (1) the inclusion of markers in the list and (2) the ability of the fractionation procedure to separate compartments. In case an individual marker list is needed, we recommend creating it based on published, experimentally validated datasets (Supplementary Information). C-COMPASS first transforms input data into normalized profiles across all fractions for each replicate. To address class imbalances in the marker set, underrepresented classes are upsampled, and the set is augmented by simulating multiple localizations through mixing profiles from two compartments (Extended Data Fig. 1a). The core model is a five-layer neural network-based regression model trained on both experimental and synthetic profiles (Extended Data Fig. 1b). It consists of an input layer (matching the number of fractions), two dense layers (one tunable), a normalization layer ensuring that outputs sum to one per protein across compartments and an output layer. Model parameters, including learning rate and optimizer, are tuned for each run. Hyperparameter optimization is performed by splitting the set for training and validation over several iterations (Extended Data Fig. 1c). The optimal model is then trained on all marker profiles to predict spatial distribution for proteins with unknown localization.

To reduce false positives, values below compartment-specific thresholds are set to zero, and the remaining values are renormalized to sum to one, producing class contributions (CCs) that reflect protein distribution across compartments. To stabilize predictions, the process is repeated multiple times, generating ensemble averages. These condition-specific CC values form the first C-COMPASS output (Extended Data Fig. 1d). For comparative analysis, changes in CC between conditions, called relocalization, range from −1 to 1. In cases in which only two compartments are involved, the direction of relocalization can be determined based on opposing shifts in localization values. For more complex cases involving multiple compartments, only

changes in spatial distribution between conditions can be assessed. Variation across ensemble runs allows $P$-value and Cohen's $D$ calculations per compartment, yielding two metrics: the relocalization score (RLS), summarizing total change (0–2), and the distance score (DS), ranking effect sizes. These global relocalization metrics make up the second C-COMPASS output (Extended Data Fig. 1d).

To investigate how compartments adjust their composition with biological changes, C-COMPASS optionally integrates total proteome data. By normalizing CC values with corresponding protein levels and estimated organelle abundance (from summed marker intensities), changes in compartment composition can be modeled. This third C-COMPASS output offers a class-centric perspective on protein-level changes within compartments (Extended Data Fig. 1d). An increased depth of the proteome is beneficial here, as it increases the number of proteins with comprehensive data at both the proteome and localization levels. Detailed descriptions of all C-COMPASS output values are available in the online documentation.

### Demonstrating the application of the C-COMPASS pipeline

To evaluate the C-COMPASS steps comprising marker upsampling, multiorganelle localization predictions and output filtering, we applied the pipeline to a published dataset of human white adipocytes containing 5,530 proteins separated via PCP across 24 fractions[22] (complete data in Supplementary Table 1). Predictions were compared to SVM outputs from the original study using the same 12-compartment marker set (Fig. 2a,b).

We first assessed the impact of upsampling to balance marker count discrepancies across compartments. Uniform manifold approximation and projection (UMAP) visualization confirmed that upsampling preserved distinct clusters and similarity between artificial and experimental profiles (Fig. 2c). Performance was evaluated using precision (correct among predicted), recall (correct among actual) and $F_1$ score (harmonic mean) (Fig. 2d). Proteins from compartments with fewer markers benefited most, while those from well-represented compartments retained high $F_1$ scores. For example, lipid droplet (LD) proteins, initially poorly predicted ($F_1 < 0.01$), had improved prediction to 0.50 after upsampling. Mitochondrial proteins, with abundant markers, consistently showed high $F_1$ scores (0.82–0.86) without upsampling. Overall, upsampling improved precision (0.44–0.60), recall (0.47–0.66) and $F_1$ score (0.46–0.63).

To further evaluate organelle classification, we analyzed the prefiltered distribution of class predictions for marker proteins. These, typically localized to single compartments, showed high prediction values for their respective compartments and low values elsewhere, aligning with biological expectations (Extended Data Fig. 2a). Mitochondrial proteins, known for distinct targeting signals, exhibited strong mitochondrial predictions and minimal misclassification (Fig. 2e), yielding the highest accuracy among compartments. Among 583 mitochondrial predictions, only two false positives were identified (Fig. 2f). By contrast, LD proteins exhibited lower prediction accuracy, likely due to limited resolution, overlapping profiles or the inherent characteristics of LD proteins themselves, which are highly dynamic and often multilocalized. Specifically, LD proteins showed higher prediction scores for both the endoplasmic reticulum (ER) and the cytosol, aligning with their known targeting pathways from either the cytosol or the ER (CYTOLD and ERTOLD)[23] (Fig. 2e). Ranking LD predictions across all markers identified four false positives among 26 predicted LD proteins (Fig. 2f). Overall, profiles of C-COMPASS predictions closely matched marker profiles for their respective compartments (Fig. 2g). To assess prediction reliability, we compared correctly predicted marker proteins between C-COMPASS and SVMs[22] and validated uniquely assigned proteins using databases such as MitoCarta[24] and LD Knowledge Portal (LDKP)[25]. C-COMPASS and SVMs overlapped in 53.7% of mitochondrial classifications. C-COMPASS showed greater overlap with marker proteins (97.1%) than SVMs (83.9%), indicating a higher reliability.

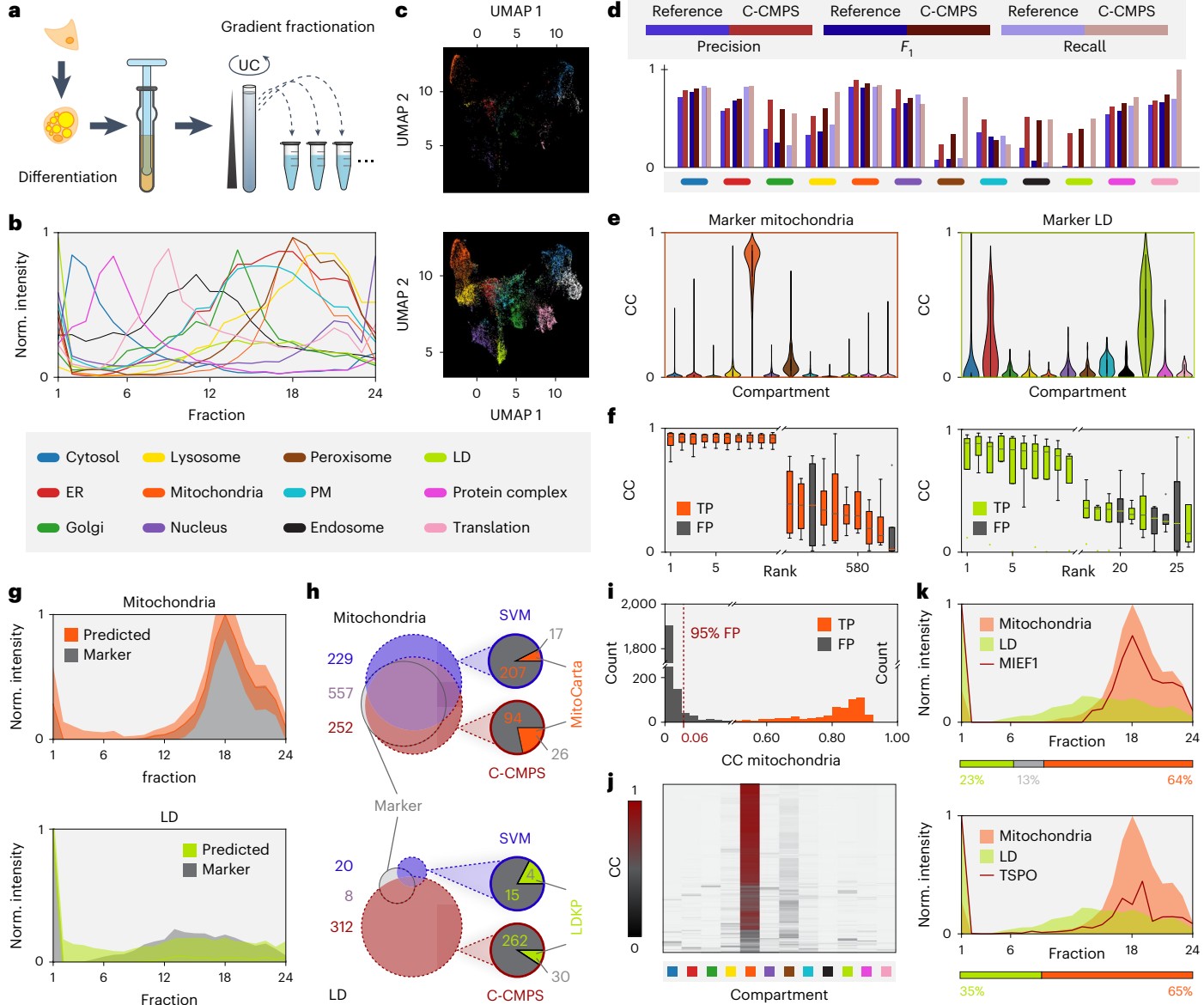

**Fig. 2 | Application of the C-COMPASS pipeline. a**, Scheme to generate an organellar map of adipocytes via PCP[58]. **b**, Profiles of compartment markers (median from four biological replicates, min–max scaled). Norm., normalized; PM, plasma membrane. **c**, UMAPs displaying data points across all replicates for compartment markers. UMAPs trained (n_neighbors = 10, n_components = 2, metric = 'cosine', min_dist = 0.1, spread = 1.0, random_state = 42, learning_rate = 1.0, n_epochs = none, init = 'spectral' and transform_seed = 42) on an upsampled dataset (bottom) were applied to represent the original marker proteins (top). White data points represent the endosome. **d**, Precision, $F_1$ and recall metrics without upsampling (blue) and with C-COMPASS (C-CMPS) upsampling and mixing (red). Proteins were assigned to compartments based on the highest neural network outputs. **e**, For mitochondria and LD, raw neural network outputs from all neurons derived from marker proteins are summarized. Violins are width scaled, inner lines represent quartiles, and dots represent median values. **f**, Box plots display ranked network outputs for mitochondria and LD marker proteins from four biological replicates, each containing three optimization rounds with ten training and prediction runs, resulting in 30 values for each replicate per box. True positives (TP) are colored,

and false positives (FP) are gray. Medians are shown with middle lines, boxes indicate interquartile ranges (IQR), whiskers extend to 1.5 times the IQR, and dots represent outliers. **g**, Min–max-scaled median protein profiles. Marker proteins are shown in gray, with colored predicted median profiles based on highest neural network values. Error bands indicate s.d. **h**, Venn diagrams comparing proteins with the highest neural network output values for mitochondria and LD from C-COMPASS (red) with original SVM predictions by Klingelhuber et al.[22] (blue). Colorless circles show intersections with marker proteins. C-COMPASS and SVM unique predictions are compared with MitoCarta[24] and LDKP[25]. **i**, Histogram of neural network output values for mitochondria. Orange bars represent mitochondrial markers, gray bars represent markers for other organelles, and the red dashed line shows the filtering threshold. **j**, Heatmap of CC values per compartment, filtered for proteins at least partially localized to mitochondria. **k**, Example min–max-scaled protein profiles for two proteins localized to mitochondria and LD. Colored areas in line plots represent median marker profiles for mitochondria (orange) and LD (green). Horizontal bars indicate compartment distribution as predicted by C-COMPASS.

---

Additionally, 26 of the 120 proteins uniquely assigned to mitochondria by C-COMPASS but absent from the marker set were validated by MitoCarta, suggesting that they are likely true mitochondrial proteins (Fig. 2h). By contrast, only 17 of the 229 proteins uniquely assigned by

SVMs were found in MitoCarta. For LDs, C-COMPASS predicted 320 proteins, with 38 confirmed (eight marker proteins, 30 by LDKP) (Fig. 2h), indicating improved LD protein identification. Overall, C-COMPASS outperformed SVMs in correctly assigning marker proteins in 11 of 12

compartments (Extended Data Fig. 2b), indicating strongly enhanced performance and reliability. The overall $F_1$ score of C-COMPASS (0.84) slightly exceeded that of SVMs (0.80) (Extended Data Fig. 2c).

Finally, we evaluated the effectiveness of the filtering step. Some marker proteins from other compartments showed low mitochondrial prediction values, indicating potential false positives. Applying a 95% false positive rate threshold per compartment effectively removed false positives while retaining true positives, as confirmed for mitochondrial markers (Fig. 2i). The filtered values, the CCs, were then used to predict multiple organelle localizations, identifying both exclusively localized and multilocalized proteins. This revealed potential organelle contact site proteins overlooked by SVMs (Fig. 2j). For example, the mitochondrial outer membrane protein MIEF1, involved in fission regulation at contact sites[26], and the mitochondrial cholesterol transporter TSPO, which mediates cholesterol redistribution[27], were both found to colocalize with mitochondria and LDs (Fig. 2k), suggesting roles in LD–mitochondria communication.

## Broad applicability of C-COMPASS across organelle separation workflows

To assess C-COMPASS's applicability across organelle separation methods and its reliability in detecting protein relocalization, we applied it to the HyperLOPIT dataset from Mulvey et al.[28], which combines differential centrifugation with tandem mass tag labeling proteomics to analyze protein localization changes in THP-1 cells after lipopolysaccharide (LPS)-induced immune activation (Fig. 3a).

Despite the different experimental workflow (Fig. 3b, Extended Data Fig. 3a and complete data in Supplementary Table 2), C-COMPASS achieved $F_1$ scores of 0.87–1.00 (average of 0.96 weighted on marker protein count per compartment) in the control condition and 0.66–1.00 (weighted average of 0.93) for the LPS-treated condition, indicating high prediction reliability (Fig. 3c and Extended Data Fig. 3b). We found that 59.5% of proteins exhibited multiple localizations (Extended Data Fig. 3c). CC values across compartments aligned with biological context (Fig. 3d and Extended Data Fig. 3d,e). For example, secretory pathway proteins frequently localized to the Golgi apparatus, endosomes and the plasma membrane while many mitochondrial proteins showed exclusive localization. A comparison matrix with Mulvey et al.'s data[28] showed strong agreement (Fig. 3e and Extended Data Fig. 3f), confirming C-COMPASS's accuracy across fractionation methods beyond PCP. Notably, C-COMPASS outperformed the original methodology by identifying multiple localizations previously missed. For instance, proteins initially categorized solely as lysosomal were found to simultaneously localize to the plasma membrane and those labeled as nuclear were additionally found in ribosomes, chromatin and the cytosol. These biologically plausible multilocalizations suggest that C-COMPASS captures true multicompartment distributions. Moreover, 81% of proteins categorized as unknown in the original study were found to localize to multiple compartments by C-COMPASS, demonstrating its capability to overcome challenges associated with assigning definitive localizations to these proteins through its ability to predict multilocalization patterns.

We further validated C-COMPASS for identifying protein relocalization. We filtered proteins with an RLS > 1 after LPS treatment, indicating a 50% change in organelle assignment, and plotted the C-COMPASS distance metric against the RLS $P$ value to identify highly reliable hits (Extended Data Fig. 3g). These proteins were enriched in pathways known to be activated by LPS, including MHC-I[29] and the unfolded protein response[30,31] in addition to vesicular transport and cytoskeletal remodeling described in the original study (Fig. 3f). In concordance with the original study, C-COMPASS detected a high number of relocalization events, especially from the lysosome to the plasma membrane and from the cytosol to the nucleus (Fig. 3g), supporting the reliability of C-COMPASS in detecting biologically plausible relocalization events.

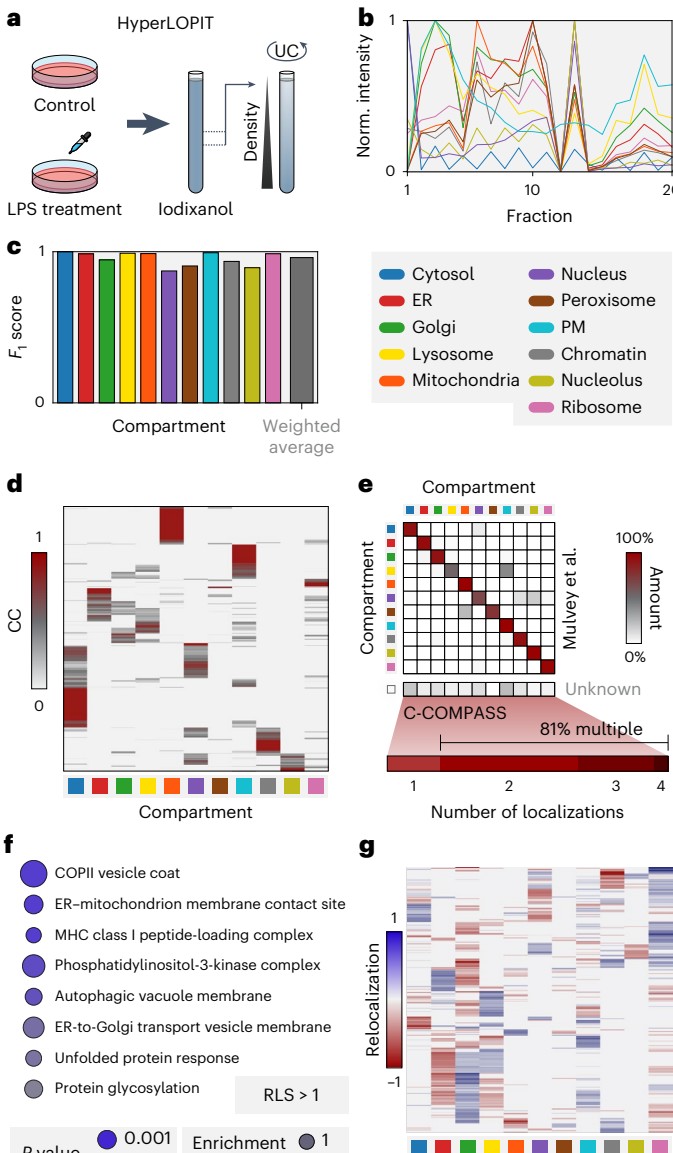

**Fig. 3 | The application of C-COMPASS to different fractionation workflows. a**, Experimental design to generate a cellular map of LPS treatment via HyperLOPIT by Mulvey et al.[28]. **b**, Profiles for compartment markers, with line values representing median values derived from median profiles across replicates for the control condition. Norm., normalized. **c**, Bar plot showing $F_1$ scores for each compartment in the control condition. Main organelle assignments were determined using the highest CC value per protein. The gray bar represents the average $F_1$ value weighted by batch sizes of marker proteins. **d**, Heatmap of CC values across all compartments in the control condition. **e**, Correlation matrix comparing original compartment associations by Mulvey et al.[28] with C-COMPASS main compartment predictions for the control condition. The lower bar shows the percentage of proteins with varying numbers of localizations for proteins not assigned in the original study. **f**, GO annotation, KEGG and keyword enrichment analysis for proteins with RLS > 1 upon LPS treatment (one-sided Fisher's exact test, false discovery rate < 0.15). **g**, Heatmap of relocalization values for each compartment across all proteins upon LPS treatment.

## Impact of fractionation resolution on prediction accuracy

The experimental design for C-COMPASS analyses requires balancing resolution with practical limitations, including mass spectrometry (MS) measurement time and workload. To systematically assess how the number of fractions affects performance, we created simulated datasets that closely reflect gradient centrifugation experiments

(Methods). By varying both fraction counts and compartment numbers, we evaluated localization predictions for single-compartment and multicompartment proteins to determine how resolution impacts accuracy (Fig. 4a and complete data in Supplementary Table 3). Assessing prediction quality from experimental datasets is challenging, as the ground truth for localizations is typically not available, especially for multiple localization patterns. Therefore, to estimate performance, we used a theoretical model that reflects ideal experimental behavior. We performed simulations in which multilocalization profiles are generated by mixing compartment-specific features in defined ratios. These simulations allow us to evaluate C-COMPASS's performance under controlled conditions. For real datasets, we assume comparable accuracy, provided that the experimental data quality is optimal.

Prediction accuracy improved with an increasing number of fractions, as indicated by reduced mean prediction error (Fig. 4b). Correspondingly, the proportion of proteins with high prediction accuracy, defined by a localization error (LE) below 0.1, increased across various simulated compartment numbers (Fig. 4c). However, when the fraction-to-compartment-ratio was low, approximately 6% of proteins remained unassigned due to C-COMPASS's built-in filter, which removes low-confidence predictions caused by insufficient resolution (Fig. 4d). $F_1$ scores further confirmed that compartment resolution depends directly on the number of fractions. While $F_1$ values remained above 0.9 for up to 12 compartments with 16 fractions, the highest scores occurred when the number of fractions was at least twice the number of compartments (Fig. 4e), highlighting the need to balance these parameters in the experimental design. We further evaluated C-COMPASS on complex localization patterns using the simulation with eight compartments and 16 fractions, where proteins were assigned to multiple compartments with increment contributions of 0.25. Predicted CC values closely matched expected distributions (Fig. 4f), demonstrating accurate quantification of multilocalizations. To identify sources of misclassification, we computed mean prediction values for proteins assigned to single compartments (Fig. 4g), revealing that inaccuracies were predominantly associated with overlapping fractionation profiles (Fig. 4h), whereas well-separated compartments showed minimal false positives. We further assessed C-COMPASS's performance in detecting and quantifying protein relocalization based on simulated data, which showed strong concordance between predicted and true localization changes. Compartment-specific relocalization error (RLE) remained low (maximum of 0.048) (Fig. 4i). The highest errors occurred between compartments with overlapping profiles (Fig. 4j), while relocalizations between distinct compartments were more accurately resolved.

In sum, these results indicate that limitations in localization and relocalization accuracy primarily arise from overlapping profiles, which can be mitigated by increasing fractionation resolution. For optimal performance, we recommend using at least twice as many fractions as the number of compartments to be resolved.

## Comparison of C-COMPASS with existing organelle proteomics analysis pipelines

Having validated the performance of C-COMPASS using simulated datasets, we next employed these datasets to compare its capabilities with those of other computational tools for spatial proteomics analysis via predictive models of protein localization (see Extended Data Table 1 for a detailed feature comparison). We selected BANDLE and DOM-ABC for direct benchmarking, as both generate output formats most comparable to those of C-COMPASS. While all are theoretically applicable across various workflows and provide compartment-specific values per protein, BANDLE and DOM-ABC yield probability-based classifications, whereas C-COMPASS quantitatively estimates protein distributions across organelles.

To assess localization accuracy, we focused on 1,139 proteins exclusively localized to a single compartment (complete data in Supplementary Table 4). Of these, 720 were correctly assigned by all three methods, while 16 were misclassified by each (Fig. 4k). C-COMPASS and BANDLE outperformed DOM-ABC in terms of the number of correctly predicted proteins. However, DOM-ABC applied a stricter internal filtering strategy, excluding more proteins to reduce false positives (Fig. 4l). When evaluated using standard classification metrics (precision, recall and $F_1$ score), C-COMPASS achieved the highest performance for single-compartment localizations (Fig. 4m). To further assess tool performance, we examined prediction accuracy across proteins with increasing numbers of simulated localizations (Fig. 4n). All methods performed comparably for proteins with single localizations, although C-COMPASS exhibited slightly higher variance. In dual- and triple-localization cases, C-COMPASS consistently outperformed BANDLE and DOM-ABC, showing lower prediction error in these complex scenarios (Fig. 4o). Finally, we compared relocalization accuracy between C-COMPASS and BANDLE (DOM-ABC was excluded due to incomparable output). C-COMPASS consistently achieved lower RLEs, confirming strength in detecting dynamic compartment changes under varying conditions (Fig. 4p). In summary, all tools robustly predict single localizations; however, C-COMPASS demonstrates superior performance in capturing multilocalization patterns.

## Organelle composition modeling in a mouse model with humanized liver

Organelle composition is determined by three main factors: (1) protein localization, (2) protein abundance and (3) organelle abundance.

**Fig. 4 | Evaluation of C-COMPASS performance across fraction numbers and comparison with other organelle proteomic tools. a**, Simulated marker profiles shown across different numbers of fractions. Profiles are shown for simulations with minimum and maximum numbers of compartments and fractions. Norm., normalized; int., intensity. **b,c**, Line plot of mean prediction error (**b**) and line plot of the number of proteins with a prediction error below 0.1 across increasing numbers of simulated fractions (**c**). Errors were calculated as the sum of absolute differences across compartments, divided by 2. **d**, Scatterplot showing the percentage of proteins that remained unassigned by C-COMPASS, plotted against the ratio of simulated fractions to compartments. **e**, $F_1$ scores for simulations (sim.) containing 16 fractions and varying numbers of compartments. **f**, Predicted CC values for proteins with different simulated localization amounts. Blue circles indicate the mean, black lines show the median, and gray lines represent the IQR. **g**, Matrix showing average predicted CC values per compartment (*y* axis) for proteins simulated to localize to a single compartment (*x* axis). Empty cells represent values of 0. **h**, Normalized intensity profiles of simulated marker proteins for the first condition of the dataset used in **g**. **i**, Heatmap of RLEs. Matrix shows origin compartments (*y* axis) and target compartments (*x* axis) for relocalizing and non-relocalizing proteins. Av. abs., average absolute. **j**, Normalized intensity profiles of marker proteins for the second condition of the dataset used in **g,i**. **k**, Venn diagram showing the overlap of correctly classified single-localization (loc.) proteins across three analysis tools. **l**, Number of proteins successfully analyzed (colored bars) and those excluded due to method-specific filtering. **m**, Prediction metrics for proteins simulated with single localizations across three methods. **n**, Box plots of prediction errors across tools, categorized by proteins with one, two or three simulated localizations (*n* = 878, single, DOM-ABC, yellow; *n* = 1,139, single, BANDLE, blue; *n* = 1,131, single, C-COMPASS, red; *n* = 1,254, double, DOM-ABC, yellow; *n* = 1,612, double, BANDLE, blue; *n* = 1,607, double, C-COMPASS, red; *n* = 345, triple, DOM-ABC, yellow; *n* = 439, triple, BANDLE, blue; *n* = 437, triple, C-COMPASS, red). **o**, Total prediction error per protein for each method, computed as the sum of absolute errors across compartments, divided by 2 (*n* = 2,477, DOM; *n* = 3,252, BANDLE; *n* = 3,175, C-COMPASS). **p**, Prediction error for protein relocalization events, between predicted and simulated changes in localization between the conditions (*n* = 1,179, BANDLE, 0.95; *n* = 1,106, BANDLE, 0.99; *n* = 2,092, C-COMPASS). BANDLE relocalizations were obtained by thresholding the differential localization (DL) score at 0.95 or 0.99. **n**–**p**, Medians are shown with middle lines, boxes indicate IQR, whiskers extend to 1.5 times the IQR, and dots represent outliers.

C-COMPASS integrates these to assess organelle-specific protein abundance changes by combining localization changes and protein expression levels derived from the total proteome. This integration allows estimation of organelle abundance changes and normalization of results. We generated organelle maps in mice with humanized livers, studying organelle remodeling in human hepatocytes under various metabolic conditions, providing organelle proteomics data in a mouse model with humanized liver (complete data in Supplementary Table 5).

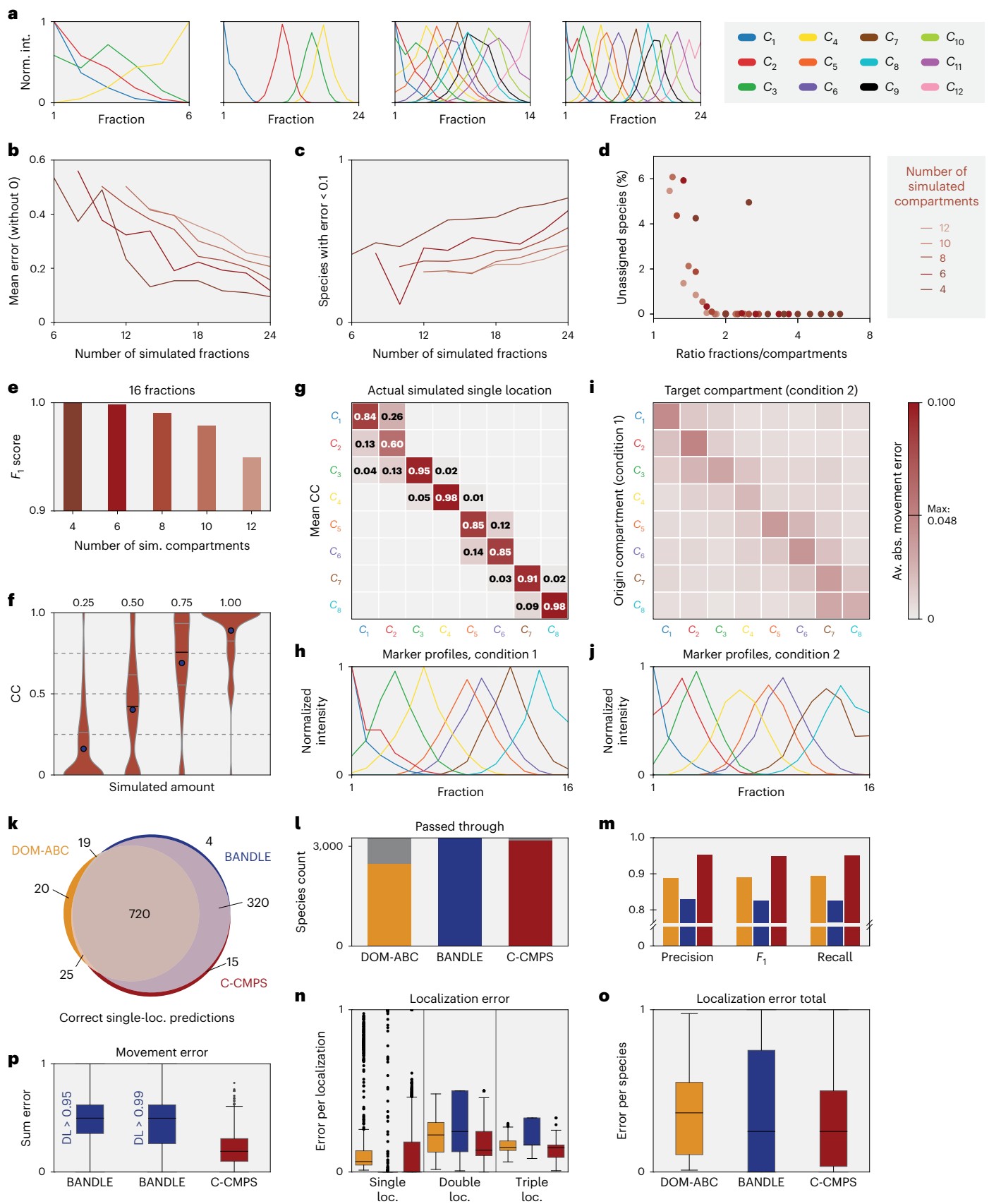

We used the FRG (deficient in FAH, RAG2 and IL2RG) mouse model[32] in which the liver was humanized by engrafting FAH-proficient human hepatocytes carrying a *PNPLA3* mutation, known to exacerbate lipid accumulation[33], to evaluate the capability of C-COMPASS in mapping protein relocalization and organelle remodeling under different metabolic perturbations. We compared cellular architecture in humanized livers on a standard chow diet, on a high-fat, high-fructose (HFHF) diet and after fasting in HFHF-fed mice (Fig. 5a). Streamlining the PCP protocol from 24 to 14 fractions, we maintained the ability to identify clusters corresponding to 11 compartments, although resolution was slightly reduced (Fig. 5b,c and Extended Data Fig. 4a,b). C-COMPASS identified 80–100% of marker proteins within their classes (Fig. 5d and Extended Data Fig. 4c,d). Distribution of proteins with single and multiple localizations was comparable to that in previously analyzed datasets (Fig. 5e–g and Extended Data Figs. 4e and 5a–c). We consistently identified 7,690 proteins across metabolic conditions, with half maintaining the same main organelle prediction between conditions. Around a quarter exhibited minor changes (RLS < 1), while the rest were evenly split between fully (RLS = 2) and partially relocalized (RLS between 1 and 2) proteins (Fig. 5h). Of the ~2,000 proteins with RLS > 1, 1,300 overlapped in both comparisons (Extended Data Fig. 6a). All compartments were affected to varying extents, with LDs showing a particularly high percentage of relocalization events (Fig. 5i,j and Extended Data Fig. 6b,c).

To identify candidates with the most robust relocalization, we analyzed the C-COMPASS distance score (DS) between conditions and corresponding *P* values for proteins with an RLS > 1 (Fig. 5k and Extended Data Fig. 6d). Among these candidates, we found proteins previously reported to undergo starvation-induced relocalization, such as PLIN5, which facilitates fatty acid mobilization[34–36]. PLIN5 was fully localized to LDs under the chow condition, relocalized to the cytosol under HFHF feeding and returned to LDs after fasting (Fig. 5l). Additionally, we validated the nuclear translocation of glucokinase regulator (GCKR) (Extended Data Fig. 6e), which sequesters glucokinase in the nucleus under low-glucose conditions[37]. Beyond these known examples, C-COMPASS enabled the discovery of previously uncharacterized fasting-induced protein relocalization, including that of SINHCAF, RBBP7 and ASF1A, accessory subunits of the histone deacetylase SIN3–HDAC complex[38,39]. These proteins, which are mainly cytosolic or lysosomal in the fed state, relocalize to the nucleus during fasting (Fig. 5m–o and Extended Data Fig. 7a). Furthermore, our analysis revealed fasting-induced relocalization of the conserved oligomeric Golgi (COG) complex[40], specifically the dissociation of its lobe B subcomplex, which shifts from the Golgi apparatus to the cytosol (Extended Data Fig. 7b). Dysregulation of COG subunits is associated with altered vesicular trafficking and secretion, implying that COG complex remodeling during fasting might impact these cellular processes[41]. These findings underscore the capability of C-COMPASS in uncovering new protein relocalization events, providing insights into the dynamic regulation of protein localization.

Next, we applied the C-COMPASS pipeline to assess cellular changes in an organelle-centric manner by integrating data on protein localization, levels and compartment quantities from total proteome analyses. This approach allowed us to adjust protein distribution across compartments and identify proteins with altered levels in specific organelles, predicting functional changes. For instance, in mitochondria, enrichment analysis of GO annotations, Kyoto Encyclopedia of Genes and Genomes (KEGG) pathways and keywords within distinct protein clusters revealed functional reprogramming under different metabolic conditions (Fig. 5p). For example, a decreased respiratory capacity was observed in the HFHF-fasted state, consistent with previous findings of reduced mitochondrial respiration during fasting[42]. While changes in the mitochondrial proteome were mainly due to alterations in protein levels, the LD proteome was significantly influenced by protein relocalization events (Fig. 5q). To visualize fasting-induced changes in the LD proteome, we plotted fold changes in protein levels on LDs against LD-specific relocalization values, assessing how relocalization contributes to overall protein changes within this organelle. Most proteins either increased on LDs due to enhanced targeting or decreased with reduced LD localization. For example, ARL8A, a protein that targets LDs[43,44] and a homolog of ARL8B, which has been recently discovered to facilitate LD-mediated lipophagy[45], exhibited increased abundance and relocalization to LDs. Conversely, the ubiquitin-conjugating enzyme UBE2A relocated away from LDs, accompanied by a decrease in its LD-specific levels.

**Fig. 5 | An organellar map of the humanized liver. a**, Mice with humanized liver (repopulated with human hepatocytes after nitisione (NTBC) withdrawl induced cell death in FRG mice) were fed either a chow diet or an HFHF diet for 5 weeks, and livers were collected in an ad libitum fed state. A third group on the HFHF diet was subjected to an overnight fast. Livers from all three conditions were used for PCP. KO, knockout. **b**, Hierarchical clustering of protein profiles across 14 fractions in the chow state. Colored areas indicate clusters with the highest enrichment scores for specific compartment marker annotations. Norm., normalized. **c**, Profiles of compartment markers, with line values representing median values derived from median profiles across replicates in the chow state. **d**, Bar plot showing the number of true positive localization predictions by checking for positive CC values for each class in the chow state. The gray bar represents the average true positive value weighted by batch sizes of marker proteins. W. av., weighted average. **e**, Number of main localization assignments to compartments in the chow state. **f**, Number of proteins assigned to single or multiple compartments in the chow state. **g**, Heatmap of CC values across all compartments in the chow state. **h**, Venn diagram indicating proteins detected in all conditions and pie charts showing the number of proteins within specific RLS value ranges for the comparison of chow versus HFHF (top) and chow versus HFHF-fasted (bottom) conditions. **i**, Heatmap of relocalization values for each compartment across proteins with RLS > 1, comparing HFHF with HFHF-fasted conditions. **j**, Sankey diagram illustrating localization changes from chow to HFHF to HFHF-fasted conditions across all compartments, filtered for RLS > 0 for at least one transition while maintaining the origin and target for each relocalization. Bar charts summarize the proteins listed for one transition in the Sankey diagram, showing the number of proteins with RLS = 0 (dark red), RLS in the range of 0 to 1 (bright red), RLS between 1 and 2 (bright blue) and RLS = 2 (dark blue). **k**, Scatterplot displaying *P* values from a two-sided Welch's *t*-test based on ensemble network output values plotted against DS values for proteins with RLS > 1 for the comparison of HFHF versus HFHF-fasted conditions. The gray dashed line indicates a *P* value of 0.05. **l**, Protein profiles of PLIN5 across the indicated metabolic conditions, overlaid with median marker profiles for LDs and the cytosol. Bars on the right show CC distributions for PLIN5 between these two compartments. **m**, Protein profiles (lines) of selected candidates for HFHF and HFHF-fasted conditions, overlaid with median marker profiles (areas). Bars show CC distributions for these proteins across the displayed compartments. **n**, Relocalization heatmap for the selected proteins across the displayed compartments. **o**, Cluster network of selected biologically relevant proteins, including the displayed candidates based on the STRING database. Connection lines represent known and predicted interactions as well as information from text mining. Thickness reflects the number of distinct types of connections between the nodes. **p**, Unsupervised hierarchical clustering of mitochondrial proteins, corrected for protein expression levels and organelle abundance across metabolic conditions. GO annotations, KEGG pathways and keywords enriched in the compartments are indicated (one-sided Fisher's exact text, false discovery rate < 0.15). Compl., complex; electr., electron; respir., respiratory; transp., transport. **q**, Scatterplot of proteome changes in LDs between HFHF and HFHF-fasted conditions. Cohen's *D* values (*D*) for LDs versus fold changes (FC; relocalization values corrected by expression and abundance). Proteins with relocalization > 0.5 are red; those with relocalization < −0.5 are blue. Red in the top right shows higher LD localization and abundance in the HFHF-fasted condition; blue in the bottom left shows the opposite.

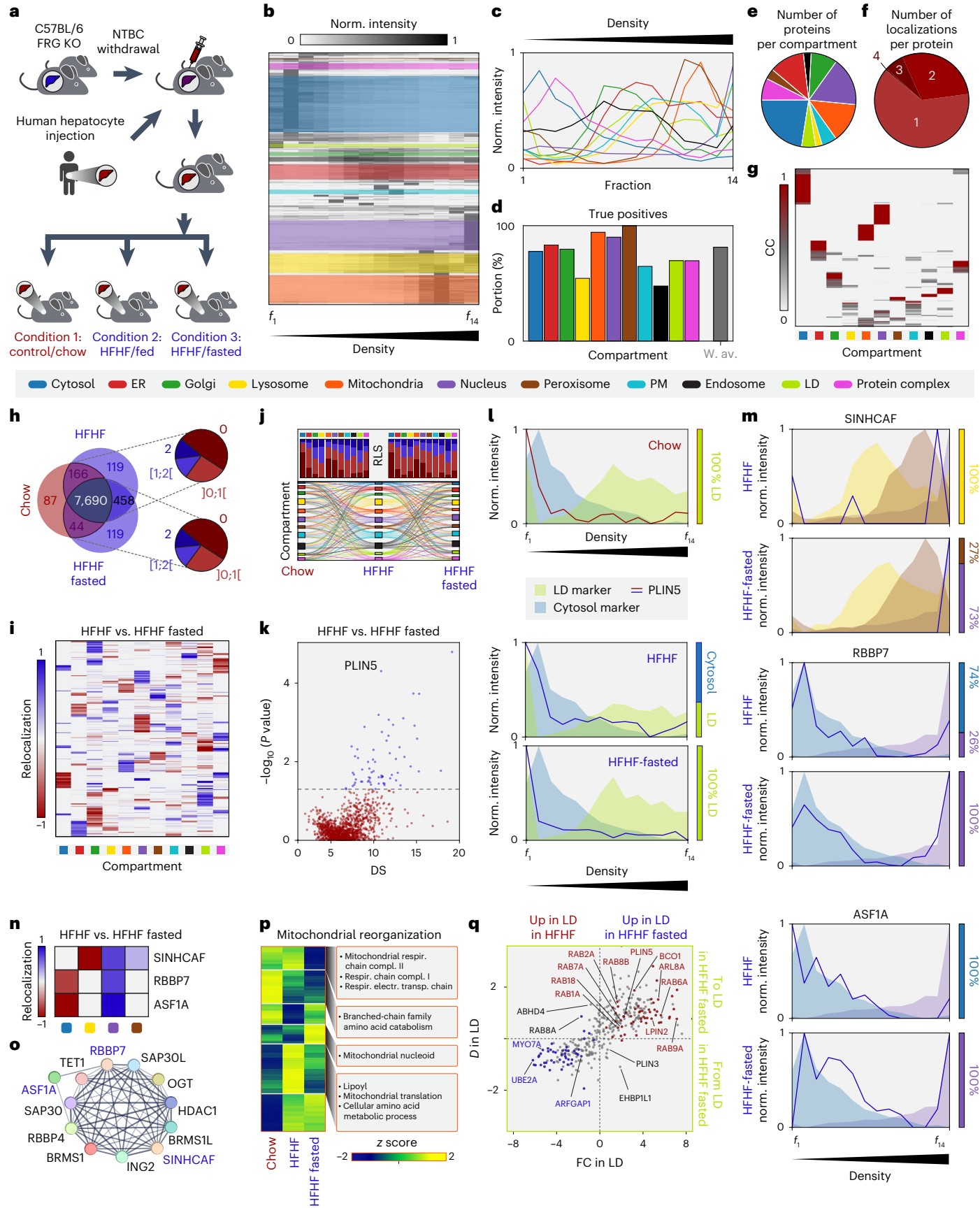

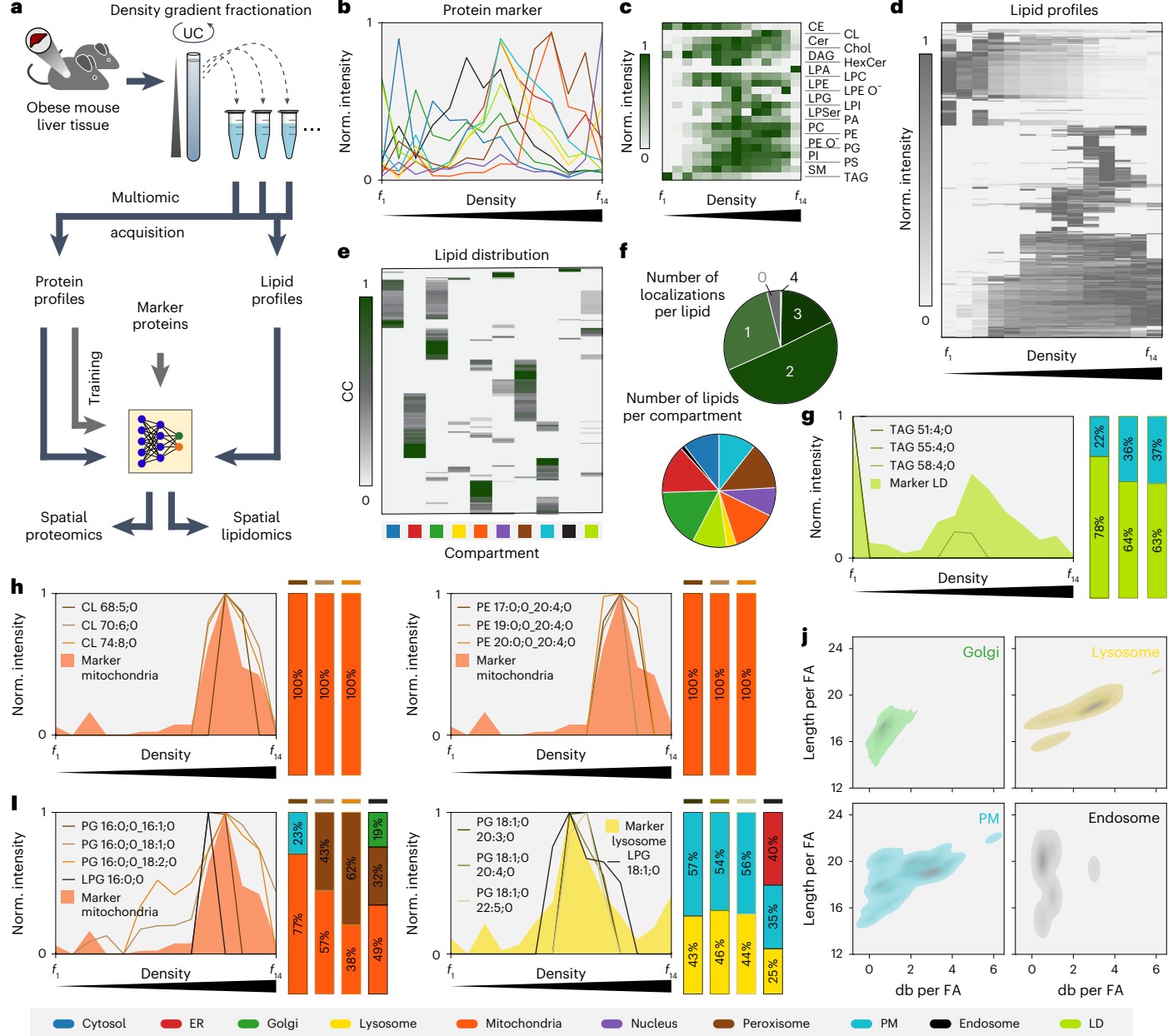

**Fig. 6 | A combined protein and lipid map of the steatotic liver. a**, Experimental setup scheme for integrating proteomic and lipidomic data. Liver samples from obese mice were used for the standard PCP workflow. All fractionation samples and total proteome samples underwent both proteomics and lipidomics analysis. After training the neural network with marker profiles derived from the proteome, both datasets were fed into the neural network for spatial predictions. **b**, Profiles of compartment markers, with line values representing median values derived from median profiles across replicates. **c**, Hierarchical clustering heatmap of different lipid classes across 14 fractions. CE, cholesterol ester; Cer, ceramide; CL, cardiolipin; chol, cholesterol; DAG, diacylglycerol; LPA, lysophosphatidate; LPC, lysophosphatidylcholine; LPE, lysophosphatidylethanolamine; LPI, lysophosphatidylinositol; LPSer, lysophosphatidylserine; PA, phosphatidate; PI, phosphatidylinositol;

PS, phosphatidylserine; SM, sphingomyelin; TAG, triacylglycerol. **d**, Hierarchical clustering heatmap of all lipid profiles across 14 fractions. **e**, Heatmap of CC values across all compartments for all identified lipids. **f**, Number of lipids assigned to single or multiple compartments and the number of identified lipid localization assignments to compartments. **g**–**i**, Indicated median protein marker profiles for different compartments (areas) overlaid with distinct examples of known and unknown compartment-specific lipid species (lines). Vertical bars indicate the percentage distribution of these lipid species across different compartments for the selected examples. **j**, Kernel density estimation plots showing distribution of carbon chain lengths and the number of double bonds (db) per fatty acid (FA) across selected compartments (bandwidth adjustment = 0.5). Values are normalized by CC values and weighted by intensities in the total lysate. Norm., normalized.

## Multiomic integration: coprofiling of organelle lipidomes and proteomes

Organelle functions are determined not only by specific proteins but also by distinct lipid compositions. Structural and quantitative variations in lipids influence membrane properties such as thickness, fluidity, curvature and surface charge. However, systematic lipid organelle

profiling has been hindered by the scarcity of organelle-specific lipid markers, resulting in poor knowledge of lipid species distribution. To address this, we evaluated C-COMPASS's ability to integrate lipidomics data and predict lipid localization using neural networks trained with marker protein profiles (complete data in Supplementary Table 6). We performed parallel proteomic and lipidomic analyses on organelle

gradients from livers of steatotic mice on a high-fat diet, using 14 fractions (Fig. 6a). High-throughput shotgun lipidomics[46,47] quantified 411 lipid species across 22 classes. Principal-component analysis revealed that the primary component separating the samples was the organellar fraction, indicating distinct, organelle-specific lipid compositions (Extended Data Fig. 8a). Organelle separation was independently confirmed by protein marker profiles (Fig. 6b). Comparison of lipid class profiles with protein marker profiles showed that triglycerides (TAGs) accumulated in the LD-rich top fraction (Fig. 6c). Cardiolipins (CLs), mitochondrial lipids[48], peaked in fractions 10 to 13, aligning with mitochondrial marker proteins. Hexosylceramides (HexCer), the abundance of which is increased in lysosomes[49], overlaid with the lysosomal marker profile. Phosphatidylcholine (PC) and phosphatidylethanolamine (PE), major membrane phospholipids, exhibited broad peaks across all membrane organelle fractions. Compared to PC, PE showed an enrichment in mitochondrial fractions, confirming higher PE content of the inner mitochondrial membrane than of other cellular membranes[50]. Hierarchical clustering of lipid species profiles indicated nonuniform distributions across organelle fractions with distinct clusters corresponding to specific organelle marker protein peaks (Fig. 6d).

Using C-COMPASS, we trained neural networks with protein marker profiles to predict lipid species CC values (Fig. 6e). Compared to proteins, a higher proportion of lipids demonstrated multiple localizations. The highest lipid complexity, indicated by the highest number of different lipids, was found in LDs, the Golgi apparatus, mitochondria and peroxisomes (Fig. 6f). Validation of C-COMPASS predictions showed that TAGs were largely predicted to localize in LDs (Fig. 6g), although the previously reported co-flotation with the Golgi apparatus in the steatotic liver[12] posed challenges (Extended Data Fig. 4b). CLs and certain PE species aligning with mitochondrial protein markers were correctly assigned to mitochondria (Fig. 6h). Ceramides, primarily synthesized in the ER[51], were predicted to localize in the ER (Extended Data Fig. 8b). In addition to confirming organelle-specific lipid localizations, our study identified previously unidentified lipid species that were selectively enriched in specific organelles. For instance, we detected enrichment of distinct phosphatidylglycerol (PG) and lysophosphatidylglycerol (LPG) species, which serve as precursors for organelle specific lipids such as cardiolipids or BMP (bis(monoacylglycero) phosphate)[52–57], at specific cellular compartments. PG and LPG with an 18:1 fatty acid at the sn-1 position localized to lysosomes, contrasting with PG(16:1) in mitochondria (Fig. 6i), suggesting fatty acid chain dependent sorting mechanisms.

Next, we used C-COMPASS outputs to (1) assess distinct lipidomic characteristics, focusing on fatty acid length and unsaturation, and (2) predict organelle lipidomes. First, to visualize differences in carbon chain length and unsaturation between organelles, we used density plots reflecting the presence and abundance of fatty acid chains. Consistent with previously reported increases in membrane thickness along the secretory pathway[57], our analysis showed that organelles involved later in the pathway, such as endosomes, lysosomes and the plasma membrane, exhibited a higher number of carbon atoms and greater unsaturation than the Golgi apparatus, which functions earlier in the pathway (Fig. 6j and Extended Data Fig. 8c). Alternatively, to analyze the lipid class length and unsaturation profiles for each organelle, we calculated the weighted mean of the number of double bonds and carbon atoms in the fatty acid moieties for each lipid class (Extended Data Fig. 8d). These weighted averages incorporate information on both lipid class and species abundances in the total lipidome and individual organelle lipidomes (Methods). While there is generally a positive correlation between fatty acid chain length and degree of unsaturation across most lipid classes, organelle-distinct patterns emerged upon comparison. For example, PG species are longer and less saturated in lysosomes than in mitochondria whereas the opposite is observed for PC, with lysosomes having shorter, more saturated fatty

acids. Additionally, PE and ether-linked PE are longer and less saturated in the plasma membrane. We next predicted organelle lipidomes by integrating lipid localization and abundance data from total tissue lipidome analyses. Applied to LDs and mitochondria, the predicted lipid compositions aligned well with reported data (Extended Data Fig. 8e), accurately reflecting levels of PC and PE in mitochondria, indicating that C-COMPASS enables the creation of integrated cellular maps at the protein and lipid levels for a comprehensive view of cellular architecture.

## Discussion

Here, we introduce C-COMPASS, an open-source software designed to analyze proteome-wide spatial distributions across cellular compartments using a neural network-based regression model. C-COMPASS provides a unique and complementary solution to existing spatial proteomic tools. Unlike most methods that use probabilistic classifiers (for example, SVMs, $k$-means, Bayesian inference) to assign proteins to a single compartment based on likelihood, C-COMPASS employs a neural network-based multiclass, multilabel regression approach. This enables quantitative estimation of protein distribution across multiple compartments, capturing multilocalization patterns. While some tools allow condition comparisons, they are typically limited to pairwise, qualitative analyses. By contrast, C-COMPASS is the only tool among those compared that provides quantitative rather than probabilistic predictions for multilocalized proteins. C-COMPASS improves prediction accuracy over traditional SVM-based methods[12,21], especially for underrepresented organelles. The software is compatible with various fractionation and labeling methods, making it a versatile tool for multicompartmental analysis. C-COMPASS includes a graphical user interface and is available as both a PyPI package and a standalone executable, requiring no command-line or R-based setup. Together, these features make C-COMPASS a powerful and user-friendly tool accessible to a broad scientific community and applicable to diverse biological questions across various model organisms.

We show that C-COMPASS can extend beyond proteomics and co-integrate lipidomics to co-map protein and lipid compositions of organelles. We use it to create, to the best of our knowledge, the first systematic organelle map at the lipid level. By concurrently mapping protein and lipid compositions, C-COMPASS overcomes the challenge of lipid assignment due to the lack of spatial lipid markers, identifying organelle-specific lipid species and variations in membrane lipid composition. Traditional studies limited to either proteomics or lipidomics have constrained our understanding of cellular functions and dynamics, as proteins and lipids jointly shape the structure, function and dynamics of membranes and organelles. Proteins rely on specific lipid environments for proper localization and activity, while lipids are regulated by protein interactions and signaling pathways. Our dual mapping approach facilitates the investigation of how lipid species influence protein function, membrane organization and signaling pathways, offering insights into how changes in lipid composition affect protein roles and contribute to cellular dysfunction and disease. C-COMPASS allows researchers to capture static and dynamic snapshots of organelle composition, compare temporal processes and assess the impact of perturbations, such as lipid transfer protein knockouts. Moreover, in the future, it can be extended to metabolic tracing approaches for lipid flux analyses, which track lipid transfer across compartments over time.

Despite its strengths, users should be aware of certain limitations. Similar to other tools, C-COMPASS predictions depend on the resolution of the experimental data and may be less accurate for compartments with overlapping or cofractionating profiles, such as the Golgi apparatus and LDs in the steatotic liver. Therefore, it is important to assess profile uniqueness and compartment-specific error rates before applying the workflow. Optimizing fractionation depth or adjusting the number of fractions can improve performance but increases workload

and resource demands. C-COMPASS currently uses zero imputation for missing values, which we consider appropriate for fractionation data, although future versions could support alternative strategies. Another challenge is the underrepresentation of multilocalized proteins in training data: while ~30% of profiles may reflect mixed localization, only 5% of training data do. Still, model performance remains robust due to the distinctiveness of most compartment profiles. C-COMPASS detects relative relocalization but does not provide absolute quantification, as estimates depend on experimental factors such as protein yield and MS loading. Therefore, validation using orthogonal methods is recommended. Additionally, biological variables such as changes in tissue cell type composition (for example, in the steatotic liver) can affect fractionation profiles and must be considered for accurate interpretation.

## Online content

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

## Methods

### Mouse models

Six-week-old female C57BL/6N FRG KO mice repopulated with human hepatocytes (HHF13022, PNPLA3$^{II48M}$ +−/−+), purchased from Yecuris, were housed at Regeneron Pharmaceuticals, New York Medical College under controlled conditions (12-h light–dark cycle, 22 ± 1 °C, 60–70% humidity). The mice were fed ad libitum with either standard low-protein chow (5LJ5, Purina Laboratory Rodent Research Diets, 5001) or an HFHF diet (D09100310iA18080901i, Research Diets) for 5 weeks. For the fasted condition, food was withdrawn overnight for 18 h. All mice were housed in an AAALAC-accredited mouse production barrier facility at Regeneron Pharmaceuticals, New York Medical College. At the age of 8 months, mice were euthanized. All procedures were approved by New York Medical College's Regeneron Pharmaceuticals' Institutional Animal Care and Use Committee and were conducted in accordance with the Guide for the Care and Use of Laboratory Animals, Eighth Edition and the AVMA Guidelines for Euthanasia of Animals: 2013 or 2020 Edition.

Male C57BL/6J mice for combined lipid correlation profiling and PCP were double housed and kept under constant ambient conditions of 22 ± 1 °C, 45–65% humidity and a 12-h–12-h light–dark cycle, with lights on from 6 a.m. until 6 p.m. Mice had free access to water and were fed ad libitum with a high-fat diet (D12331, 58% kcal fat, Research Diets). At the age of 43 weeks, mice were euthanized and livers were perfused postmortem with ice-cold PBS before subjecting them to organelle fractionation. Experiments were performed in accordance with the animal protection law of the European Union and upon permission from the local animal ethics committee of the Government of Upper Bavaria, Germany.

### Organelle fractionation

Half of a mouse liver was used for each experiment. PCP was performed with three to four biological replicates. PBS-perfused livers were isolated and homogenized on ice using a tissue homogenizer in 2.5 ml of buffer (20% sucrose, 20 mM Tris, pH 7.4, 0.5 mM EDTA, 5 mM KCl, 3 mM MgCl$_2$, protease inhibitor and phosphatase inhibitor cocktail from Roche). A 500-µl aliquot of the lysate was reserved for total liver proteome and lipidome analysis. The remaining 2 ml of lysate was centrifuged at 500$g$ for 15 min to pellet the nuclei. The 2 ml of supernatant was placed on top of a continuous 11-ml 20–55% sucrose gradient in 20 mM Tris, pH 7.4, 0.5 mM EDTA, 5 mM KCl and 3 mM MgCl$_2$ with protease and phosphatase inhibitors. Subcellular organelles were separated by sucrose density centrifugation at 100,000$g$ for 3 h at 4 °C using a Beckman SW 41 Ti rotor. LDs were isolated by collecting the 1-ml top fraction with a tube slicer (Beckman Coulter). The remaining 1-ml gradient fractions were collected sequentially with a pipette from top to bottom. In total, 14 fractions were collected from each biological replicate and condition for proteomic and lipidomic analysis.

### Proteomic analysis

**Proteomic sample preparation.** The protein concentration of samples was measured using the BCA Protein Assay (Thermo, 23225). A total of 50 µg of protein was adjusted (2% sodium deoxycholate and 100 mM Tris-HCl, pH 8.5) and boiled for 5 min at 95 °C and 1,000 rpm. Samples were sonicated using a Diagenode Bioruptor for 15 cycles of 30 s. Proteins were digested overnight at 37 °C and 1,000 rpm with trypsin (Sigma, T6567) and LysC (Wako, 129-02541) at a 1:50 protein-to-enzyme ratio. Proteins were reduced and alkylated with 10 mM Tris(2-carboxyethyl)phosphine and 40 mM chloroacetamide at 40 °C in the dark for 10 min. Peptides were then acidified by adding an equal volume of isopropanol and 2% trifluoroacetic acid (TFA). After centrifugation at 15,000$g$ for 10 min, supernatants were loaded onto activated triple-layer styrene-divinylbenzene reversed-phase sulfonated StageTips (3M Empore). The peptides were washed

sequentially with 100 µl ethyl acetate containing 1% TFA, 100 µl 30% methanol containing 1% TFA, and 150 µl 0.2% TFA and then eluted with 60 µl elution buffer (80% acetonitrile and 5% ammonium hydroxide). Peptides were lyophilized and dissolved in 10 µl MS loading buffer (2% acetonitrile and 0.1% TFA).

**Proteomic data acquisition.** For LC–MS/MS analysis, 500 ng of peptides were analyzed using an Orbitrap Exploris 480 (Thermo Fisher Scientific) equipped with a nano-electrospray ion source and FAIMS (Thermo Fisher Scientific). Fractionation samples were subjected to a single CV at −50 V, while total proteome samples used dual CVs at −50 and −70 V. The system was coupled with an EASY-nLC 1200 HPLC (Thermo Fisher Scientific). Separation of peptides occurred at 60 °C on a 50-cm column with an inner diameter of 75 µm, packed in house with ReproSil-Pur C18-AQ 1.9-µm resin (Dr. Maisch). The gradient was set to 60 min for fractionation samples and 115 min for total proteome samples, employing reversed-phase chromatography with a binary buffer system: buffer A (0.1% formic acid) and buffer B (80% acetonitrile and 0.1% formic acid). Initially, buffer B started at 5% and increased to 45% over 45 min for fractionation samples and 95 min for total proteome samples, followed by a washout phase at 95%, all at a flow rate of 300 nl min$^{-1}$. Peptides were ionized via electrospray ionization and transferred to the gas phase. Data acquisition used a DIA tandem MS method with variable window sizes. Each cycle included one MS$^1$ scan (300–1,650 $m/z$, maximum ion fill time of 45 ms, normalized AGC target of 300%, $R$ = 120,000 at 200 $m/z$), followed by fragment scans of 66 unequally spaced windows for humanized liver fractionation samples and 33 windows for high-fat diet mouse liver fractionation samples and all total proteome samples (fill time of 22 ms, normalized AGC target of 1,000%, normalized HCD collision energy of 30%, $R$ = 15,000). Spectra were recorded in profile mode with positive polarity.

**Proteomic raw data processing.** DIA raw data files were processed using Spectronaut 18 (Copernicus, Biognosys) with the directDIA+ deep-mode option, searching against the UniProt databases for humans (UP000005640_9606) and mice (UP000000589_10090). Trypsin P cleavage allowed peptide lengths of seven to 52 amino acids with up to two missed cleavages. Fixed modifications included carbamidomethylation, and variable modifications included methionine oxidation and N-terminal acetylation. The analysis selected three to six best N-fragment ions per peptide with a precursor and protein $q$-value cutoff of 1%. Global normalization used median quantities corrected for MS intensity drift.

### Lipidomic analysis

**Lipid sample preparation.** MS-based lipid analysis was performed by Lipotype as described[59]. Lipids were extracted using a chloroform–methanol procedure[60]. Samples were spiked with an internal lipid standard mixture containing CL 14:0/14:0/14:0/14:0, ceramide 18:1;2/17:0, diacylglycerol 17:0/17:0, HexCer 18:1;2/12:0, lysophosphatidate 17:0, LPC 12:0, LPE 17:1, LPG 17:1, lysophosphatidylinositol 17:1, lysophosphatidylserine 17:1, phosphatidate 17:0/17:0, PC 15:0/18:1 D$_7$, PE 17:0/17:0, PG 17:0/17:0, phosphatidylinositol 16:0/16:0, phosphatidylserine 17:0/17:0, cholesterol ester 16:0 D$_7$, sphingomyelin 18:1;2/12:0;0, triacylglycerol 17:0/17:0/17:0 and cholesterol D$_6$. After extraction, the organic phase was transferred to an infusion plate and dried in a speed vacuum concentrator. The dry extract was resuspended in 7.5 mM ammonium formate in chloroform–methanol–propanol (1:2:4, vol/vol/vol). All liquid-handling steps were performed using the Hamilton Robotics STARlet robotic platform with the Anti-Droplet Control feature for organic solvent pipetting.

**Lipidomic data acquisition.** Samples were analyzed by direct infusion on a Q Exactive mass spectrometer (Thermo Scientific) equipped with a TriVersa NanoMate ion source (Advion Biosciences). Samples were

analyzed in both positive and negative ion modes with a resolution of $R_{m/z=200} = 280,000$ for MS and $R_{m/z=200} = 17,500$ for MS/MS experiments in a single acquisition. MS/MS was triggered by an inclusion list encompassing corresponding MS mass ranges scanned in 1-Da increments[46]. Both MS and MS/MS data were combined to monitor cholesterol ester, cholesterol, diacylglycerol and triacylglycerol ions as ammonium adducts; LPC, LPC O⁻, PC and PC O⁻ as formiate adducts; and CL, lysophosphatidylserine, phosphatidate, PE, PE O⁻, PG, phosphatidylinositol and phosphatidylserine as deprotonated anions. MS only was used to monitor lysophosphatidate, LPE, LPE O⁻, LPG and lysophosphatidylinositol as deprotonated anions and ceramide, HexCer and sphingomyelin as formiate adducts.

**Lipidomic raw data processing.** Data were analyzed with in-house developed lipid identification software based on LipidXplorer[47,61]. Data postprocessing and normalization were performed using an in-house developed data management system. Only lipid identifications with a signal-to-noise ratio >5 and a signal intensity fivefold higher than that in the corresponding blank samples were considered for further data analysis.

**Mathematical problem formulation.** In our approach, each biochemical species (protein or lipid) is measured across multiple fractions in one or more biological replicates. Specifically, we measure $S$ species in $F$ fractions and $R$ replicates. For species $s = 1,...,S$ in replicate $r = 1,...R$, the measured profile is an F-dimensional vector:

$$p^{(s,r)} = \left( p_1^{(s,r)}, ..., p_F^{(s,r)} \right) \in \mathbb{R}_+^F,$$

where $p_f^{(s,r)}$ represents the abundance in fraction $f$, for $f = 1,...F$.

We assume that each profile can be explained by a combination of $C$ underlying 'compartment' profiles $p^{(c)} \in \mathbb{R}_+^F$ (one for each compartment $c$). Thus,

$$p^{(s,r)} = \sum_{c=1}^C w_c^{(s,r)} p^{(c)},$$

where the weights $w_c^{(s,r)}$ indicate how much species $s$ in replicate $r$ contributes to compartment $c$. In practice, $w_c^{(s,r)} \geq 0$, and we often interpret $\sum_{c=1}^C w_c^{(s,r)} = 1$.

For certain well-characterized species, it is known that they reside almost entirely in a single compartment. Their measured profiles can serve as references ($p_{\text{ref}}^{(s,r)}$) for compartment $c$. However, these references can vary substantially even for species apparently belonging to the same compartment, suggesting a distribution of possible 'subprofiles' for each compartment:

$$p^{(c)} \approx \text{probability distribution over profiles in compartment } c.$$

This idea, that each compartment encompasses a range of similar profiles rather than a single canonical profile, is a key feature distinguishing C-COMPASS from other methods (for example, BANDLE).

### Proteomic data processing
**Proteome fractionation data processing.** Fractionation abundance data were imported into C-COMPASS, with samples assigned to condition (group), replicate $r$ and fraction $f$. Proteins identified in at least two replicates per condition were retained, and missing values were replaced with zeros. Unlike standard proteomic analyses, in which missing values typically indicate low-level expression, fractionation data reflect compartmental localization. A protein absent from a given fraction likely indicates its absence from that compartment rather than low expression. Therefore, imputing missing values with zeros is more appropriate in this context. To ensure comparability, the profiles were normalized so that the total sum across fractions $f$ equals 1:

$$\bar{p}_f^{(s,r)} = \frac{p_f^{(s,r)}}{\sum_{f=1}^F p_f^{(s,r)}},$$

where $\bar{p}_f^{(s,r)}$ represents the normalized abundance of $p_f^{(s,r)}$. This normalization method is applied in the analysis performed by C-COMPASS. However, for visualization purposes, min–max scaling as normalization was used to rescale values into the range [0, 1]:

$$\bar{p}_f^{(s,r)} = \frac{p_f^{(s,r)} - \min_f p_f^{(s,r)}}{\max_f p_f^{(s,r)} - \min_f p_f^{(s,r)}}.$$

This alternative normalization ensures a more intuitive representation of the profiles in plots.

**Total proteome data processing.** Total proteome data were assigned to a condition (group) and replicate $r$ and filtered to retain proteins identified in at least two replicates. Missing values were imputed using a normal distribution:

$$p_{\text{imputed}} \approx \mathcal{N}(\mu = \mu_{\text{valid}} - 1.8 \times \sigma_{\text{valid}}, \sigma^2 = (0.3 \times \sigma_{\text{valid}})^2).$$

Here, $\mu_{\text{valid}}$ and $\sigma_{\text{valid}}$ represent the mean and standard deviation of the valid (nonmissing) values, respectively. This approach ensures that imputed values follow a distribution centered below the valid data while maintaining a controlled variance.

**Lipidomic data processing.** Only lipids with amounts >1 pmol were reported. Molar amounts ($\acute{l}_i$ in pmol) of individual lipid (sub)species were then normalized to the total lipid content per sample, yielding molar fraction values expressed in mol%:

$$l_i = \frac{\acute{l}_i}{\sum_{i'=1}^N \acute{l}_{i'}} \times 100,$$

where $N$ represents the total number of lipid species in the sample.

**Lipidome fractionation data processing.** Lipid fractionation data were processed alongside proteomic fractionation data, with lipids considered as additional species in the dataset. Lipid abundance values ($l_f^{(s,r)}$) were integrated into the protein list, aligning their intensity values with the corresponding fraction columns. The same filtering criteria were applied, retaining only lipids identified in at least two replicates per condition. Missing values were replaced by zeros.

For normalization, lipid fractionation profiles were treated identically to proteins. The total sum across fractions was set to 1:

$$\bar{l}_f^{(s,r)} = \frac{l_f^{(s,r)}}{\sum_{f=1}^F l_f^{(s,r)}},$$

where $\bar{l}_f^{(s,r)}$ represents the normalized lipid abundance. For visualization, min–max scaling was applied to rescale values to the range [0, 1]:

$$\bar{l}_f^{(s,r)} = \frac{l_f^{(s,r)} - \min_f l_f^{(s,r)}}{\max_f l_f^{(s,r)} - \min_f l_f^{(s,r)}}.$$

**Total lipidome data processing.** Lipids identified in at least two replicates per condition were retained, and missing values were imputed using the nearest-neighbor method ($k = 1$) after log transformation and assuming a truncated normal distribution (KNN-TN):

$$l_{\text{imputed}} = \text{KNN}_1(\log(l_{\text{valid}}))_{\text{TN}}.$$

This method estimates missing values based on the most similar existing values while preserving the overall distribution.

**Synthetic data generation.** To improve the robustness of the training process, we used synthetic data, mostly generated using tailored upsampling. This addresses the imbalance of the datasets, for example, the underrepresentation of reference profiles for certain compartments (for example, LDs), and allows us to create training data for mixed profiles.

For upsampling, we first identify the compartment with the largest marker set (size $N_{\max}$). All other compartments are then 'upsampled' to reach $N_{\max}$ through the creation of artificial profiles. For a single compartment $c$, artificial profiles are generated by:

1. Selecting three random marker reference profiles $\bar{p}_{\mathrm{ref}}^{(c,i)}$ from the existing data indicated with indices $i_1$, $i_2$ and $i_3$. This is performed for each replicate $r$ individually.
2. Computing the median of these three profiles for each fraction $f$, yielding a base profile:

$$p_{\mathrm{base},f}^{(c)} = \mathrm{median}(\bar{p}_{\mathrm{ref},f}^{(c,i_1)}, \bar{p}_{\mathrm{ref},f}^{(c,i_2)}, \bar{p}_{\mathrm{ref},f}^{(c,i_3)}).$$

3. Adding random noise (drawn from a normal distribution for which the standard deviation is twice the fraction-wise standard deviation of the selected profiles):

$$p_{\mathrm{noised},f}^{(c)} = p_{\mathrm{base},f}^{(c)} + \epsilon_f,$$

with the noise $\epsilon_f$, drawn from a normal distribution with standard deviation $\sigma_f = \mathrm{s.d.}(\bar{p}_{\mathrm{ref},f}^{(c,i_1)}, \bar{p}_{\mathrm{ref},f}^{(c,i_2)}, \bar{p}_{\mathrm{ref},f}^{(c,i_3)})$:

$$\epsilon_f \approx \mathcal{N}\left(\mu = 0, \sigma^2 = (2 \times \sigma_f)^2\right).$$

4. Rescaling to keep values within a plausible range.

For the artificial profiles generated from references for compartment $c$, we set $w_c = 1$ and the remaining weight equal to zero.

Mixed-compartment profiles are produced similarly by combining profiles from two compartments (for example, 25:75, 50:50 and 75:25 ratios). For each ratio, we add 5% of $N_{\max}$ new profiles. This procedure is repeated for every replicate, ensuring a more balanced set of training examples. The ratios used for the generation are used as weights in the training process.

**Neural network architecture.** The normalized profiles $\bar{p}^{(s,r)}$ are processed independently. The network input has the dimension $F$, matching the number of fractions in that replicate. We then use two dense (fully connected) layers, followed by a rectified linear unit with the size of the number of compartments $C$ and a normalization layer:

Input layer: size $F$.
First hidden layer: dense layer with $C + 0.4 \times (F - C)$ to $C + 0.6 \times (F - C)$ neurons and a rectified linear unit activation function. The number of neurons is a hyperparameter.
Second hidden layer: dense layer with $C$ neurons and a linear activation function.
Third hidden layer: normalization layer that ensures that the outputs sum to 1, so that each output can be interpreted as a compartment weight.

The outputs of the neural network are estimates for the compartment weights $w_1^{(s,r)}, \dots, w_C^{(s,r)}$ for species $s$ and replicate $r$. The mean and standard deviation of the compartment weights across replicates are denoted as $\bar{w}_c^{(s)}$ and $\bar{\sigma}_c^{(s)}$.

**Neural network optimization.** The model is trained to minimize the mean squared error between the predictions of the network and known (or upsampled) reference assignments. For each replicate, the upsampled data are split into 80% for training and 20% for validation. Training stops early if the validation mean squared error does not improve after five consecutive epochs (patience = 5) to avoid overfitting.

We employ Keras Hyperband to select the best-performing optimizer (stochastic gradient descent, root mean square propagation or adaptive moment estimation), the learning rate (logarithmically distributed within $[10^{-4}, 10^{-1}]$) and the size of the first hidden layer. Each configuration is tested for up to 20 epochs (with a reduction factor of 3), and the best settings are selected. After determining the best hyperparameters, we retrain the model on the complete training dataset (including synthetic profiles) for ten independent runs. This entire procedure (upsampling, mixing, training, hyperparameter tuning and training and prediction) is repeated in three rounds for each condition and replicate. We then average the raw outputs across these runs and renormalize to obtain the final CC values.

**Output filtering and final network output processing.** To further reduce spurious (false positive) assignments, we apply a compartment-specific threshold $\tau_c$. For each compartment $c$, we gather the outputs that appear in marker proteins of other compartments. The 95th percentile of these 'false positive' outputs defines the threshold $\tau_c$. If the mean predicted value for a protein $\bar{w}_c^{(s)}$ is below $\tau_c$, we set it to 0:

$$\bar{w}_c^{(s,\mathrm{filtered})} = \begin{cases} \bar{w}_c^{(s)} & , \mathrm{if} \, \bar{w}_c^{(s)} \geq \tau_c, \\ 0 & , \mathrm{otherwise} \end{cases}.$$

The remaining nonzero outputs are then rescaled so that they sum to 1, yielding the CC value:

$$\mathrm{CC}_c^{(s)} = \frac{\bar{w}_c^{(s,\mathrm{filtered})}}{\sum_{\hat{c}=1}^{C} \bar{w}_{\hat{c}}^{(s,\mathrm{filtered})}}.$$

Any protein retaining exactly one nonzero compartment after this filtering is labeled as a single localization in that compartment. We note that, in principle, this filtering step can also be used already during network training.

**Static protein localization.** CC values were used for static localization statistics, where each CC value was understood as the percentage of protein localization on a compartment. In some cases, all CC values for a protein across all conditions derived 0 after the filtering step because of values that fell below the thresholds. These proteins were excluded from further analyses.

**Protein relocalization.** Relocalization events were analyzed by pairwise comparisons between conditions (groups) $g_1$ and $g_2 \in \{1, \dots, G\}$, where $G$ is the total number of conditions. For each compartment $c$, the relocalization (RL) value was computed as the difference in predicted CCs between the two conditions:

$$\mathrm{RL}_{c,g_1,g_2}^{(s)} = \mathrm{CC}_{c,g_2}^{(s)} - \mathrm{CC}_{c,g_1}^{(s)},$$

where $\mathrm{RL}_{c,g_1,g_2}^{(s)}$ represents the amount of relocalization for compartment $c$ between conditions $g_1$ and $g_2$ and is computed from the predicted CCs in compartment $c$ for conditions $g_1$ and $g_2$, $\mathrm{CC}_{c,g_1}^{(s)}$ and $\mathrm{CC}_{c,g_2}^{(s)}$.

To assess the statistical significance of relocalization events, we applied Welch's $t$-tests to the unfiltered neural network outputs $w_c^{(s,r)}$. Welch's $t$-statistic is given by:

$$t = \frac{\bar{w}_{c,g_2}^{(s)} - \bar{w}_{c,g_1}^{(s)}}{\sqrt{\frac{\sigma_{c,g_1}^{2(s)}}{R_{g_1}} + \frac{\sigma_{c,g_2}^{2(s)}}{R_{g_2}}}},$$

where $R_g$, $\bar{w}_{c,g}^{(s)}$ and $\sigma^{2(s)}_{c,g}$ denote replicate number, mean of the predicted output and variance of the predicted output, respectively. The degrees of freedom $v$ was estimated using the Welch–Satterthwaite equation. $P$ values were then calculated from the two-sided $t$-distribution, as implemented with the scipy.stats.ttest_ind function. In addition to statistical significance, we also calculated Cohen's $d$ as a standardized effect size metric for each compartment.

$$d_c^{(s)} = \frac{\bar{w}_{c,g_2}^{(s)} - \bar{w}_{c,g_1}^{(s)}}{\sqrt{\left(\sigma^{2(s)}_{c,g_1} + \sigma^{2(s)}_{c,g_2}\right)/2}}.$$

This metric accounts for both the magnitude of relocalization and the precision of the model predictions, providing a unified score to identify promising relocalization candidates.

To quantify the overall extent of relocalization between two conditions, we computed the RLS as the sum of the absolute relocalization values across all compartments:

$$\mathrm{RLS}^{(s)} = \sum_{c=1}^{C} |\mathrm{RL}_c^{(s)}|.$$

**Organelle composition.** To analyze organelle composition across conditions, we first computed compartment-targeted protein intensities by combining predicted $\mathrm{CC}_{c,g}^{(s)}$ with total proteome intensity values. For each species $s$ and condition $g$, the total proteome intensity was calculated as the mean across all replicates $R$:

$$\bar{I}_g^{(s)} = \frac{1}{R_g}\sum_{r=1}^{R_g} I_g^{(s,r)},$$

where $I_g^{(s)}$ is the total proteome intensity for species $s$ from condition $g$ in replicate $r$ and $\bar{I}_g^{(s)}$ is the mean intensity across all replicates $Rg$ for species $s$ from condition $g$.

The compartment-weighted species abundance $\mathrm{SA}_{c,g}^{(s)}$ for each species $s$, condition $g$ and compartment $c$ was then calculated as:

$$\mathrm{SA}_{c,g}^{(s)} = \mathrm{CC}_{c,g}^{(s)} \times \bar{I}_g^{(s)}.$$

To estimate the total abundance of each compartment in a given condition, these values were summed across all species $s$ per compartment $c$:

$$\mathrm{CA}_{c,g} = \sum_{s=1}^{S} \mathrm{SA}_{c,g}^{(s)},$$

where the compartment abundance $\mathrm{CA}_{c,g}$ reflects the relative representation of compartment $c$ in condition $g$, based on the total contribution of all proteins assigned to it.

Each compartment-weighted species abundance $\mathrm{SA}_{c,g}^{(s)}$ is then normalized by the compartment abundance $\mathrm{CA}_{c,g}$, giving the normalized abundance $\mathrm{NA}_{c,g}^{(s)}$:

$$\mathrm{NA}_{c,g}^{(s)} = \frac{\mathrm{SA}_{c,g}^{(s)}}{\mathrm{CA}_{c,g}}.$$

Finally, to compute changes in the levels of species on a specific compartment, we calculated the class-centric fold change ($\mathrm{CFC}_{c,g}^{(s)}$) using $\log_2$-transformed normalized values:

$$\mathrm{CFC}_{c,g}^{(s)} = \log_2(\mathrm{NA}_{c,g_2}^{(s)}) - \log_2(\mathrm{NA}_{c,g_1}^{(s)}).$$

This CFC value reflects the $\log_2$ (fold change) in compartment-specific abundance between conditions, allowing detection of shifts in organelle targeting even when global compartment abundance also changes.

**Data simulation.** To evaluate the performance of C-COMPASS, benchmark it against alternative tools and optimize analysis parameters, we generated simulated protein abundance datasets under controlled conditions. These simulations were designed to test prediction resolution, sensitivity to relocalization events and performance under different experimental conditions.

To assess resolution, we simulated datasets with a single condition and three replicates. Each dataset contained a varying number of compartments (four, six, eight, ten or 12) and a range of fractionation depths from six to 24 fractions in steps of two. For each number of compartments, we generated multiple datasets with at least two more fractions than compartments to ensure adequate separation power.

For each replicate, the full fractionation range was split into equal-sized windows corresponding to the number of compartments. Within each window, a compartment-specific peak was generated by sampling a peak center from a uniform distribution and using it as the mean ($\mu$) of a normal distribution. The standard deviation ($\sigma$) of this distribution was randomly drawn between 1.5 and 2 to introduce variation in peak shape. A random number of data points (around 200 with 10% deviation) were then simulated for each protein using the resulting distribution to generate smooth abundance profiles.

To scale profiles to realistic intensity levels, each was multiplied by a random factor between 5,000 and 8,000. To simulate variability in MS-based quantification across fractions, we introduced a per-fraction intensity fluctuation of ±50 units around the scaling factor.

Each dataset included between 75 and 125 marker proteins per compartment and an additional 100–200 proteins with single, non-annotated localizations simulated using the same procedure.

To mimic proteins with multiple subcellular localizations, we generated additive profiles by combining two or three compartment-specific profiles, weighted according to randomly assigned localization ratios. For double-localization proteins, we simulated 150–250 proteins per compartment with either 75:25 or 50:50 localization ratios, where the first value corresponds to a fixed compartment (several per compartment class) and the second to a randomly selected one. Similarly, for triple-localized proteins, we simulated 20–80 proteins per compartment using 50:25:25 ratios, with the first compartment fixed and the remaining two selected at random per protein.

To evaluate the detection of relocalization events, we simulated an additional dataset with two biological conditions. This dataset followed the same design as above, with eight compartments, 16 fractions and three replicates per condition. The same simulation procedure for single, double and triple localizations was applied independently to both conditions to allow controlled comparison and evaluation of relocalization detection performance.

To simulate realistic variability, we introduced a 4% chance for each protein to be completely missing in one replicate, independent of its presence in others. In datasets with two conditions, a 2% probability was added for proteins to be absent in one condition but present in the other. Additionally, 1% of proteins were assigned relocalization events, where their compartment localization in the second condition was randomly reassigned and new profiles were simulated accordingly.

**Statistics for comparison of fractions and tools.** To evaluate the accuracy of each tool using simulated data, we quantified localization and RLEs by comparing the known expected values (defined by the simulation parameters) with the results from C-COMPASS, BANDLE and DOM-ABC. Specifically, expected protein localization across compartments was compared to resulting CC values for C-COMPASS and to localization probabilities for BANDLE and DOM-ABC.

BANDLE protein predictions were evaluated with version 1.8.0 in conditions 1 and 2 as described in the Bioconductor documentation (https://doi.org/10.18129/B9.bioc.bandle) with adaptations. To each replicate, the analysis was performed considering 4,000 Markov chain Monte Carlo iterations, 400 burn-in iterations and nine chains.

The matrix of Dirichlet priors was defined considering 0.1% of relocalization and the suggested values for penalized complexity priors for big datasets. The BANDLE R script and parameter settings used in this study are available in the Supplementary Information.

C-COMPASS and DOM-ABC results were distributed across the eight compartments used during simulation. By contrast, BANDLE included a ninth compartment corresponding to an 'outlier' probability, which reflects the likelihood that a protein localizes to a compartment not included in the marker list. The BANDLE authors recommend a threshold of 0.99 for both the outlier probability and the compartmental assignment probability to confidently assign a protein to a single compartment[18]. Therefore, proteins with both an outlier probability >0.99 and a compartment probability >0.99 were considered fully localized to compartment 9.

LE was calculated as the absolute difference between expected and predicted localization values for each compartment across all proteins. Similarly, RLE was calculated using the difference between expected and predicted relocalization values, where relocalization is defined as the localization difference between conditions. Both LE and RLE were divided by two to account for the fact that an overestimation in one compartment necessarily leads to an underestimation in another, thereby avoiding double counting of errors.

To further assess RLE with respect to specific origin–target compartment transitions, we applied a continuous transport-based approach that conserves predicted abundance across compartments. For each protein, we computed a transport matrix by minimizing the cost of redistributing the origin localization distribution (condition 1) into the target distribution (condition 2) using a modified earth mover's distance (EMD) that ensures conservation of total signal.

This yielded a transport matrix $F = [f_{i,j}]$ per protein, where each entry reflects the amount redistributed from a specific origin compartment $i$ to a specific target compartment $j$:

$$\text{EMD}^{(P,Q)} = \min_F \sum_{i=1}^{m} \sum_{j=1}^{n} f_{ij} \times d_{ij},$$

where $P = (p_1,\ldots,p_m)$ is the origin localization distribution (condition 1), $Q = (q_1,\ldots,q_m)$ is the target localization distribution (condition 2), $d_{ij}$ is the cost of transporting one unit of mass from compartments $i$ to $j$ and $f_{ij}$ is the amount of mass moved from $i$ to $j$.

The same procedure was applied to both expected and predicted distributions. Element-wise differences between the expected and resulting transport matrices were used to calculate the per-element transport error. The final global error matrix was obtained by averaging these error matrices across all proteins present in both conditions.

**Quality assessment.** To evaluate the performance of compartment predictions, we calculated standard classification quality metrics, including precision, recall and $F_1$ score, using the scikit-learn Python package. Main protein annotations served as the ground truth, and predicted compartments were determined by selecting the compartment with the highest predicted $CC_c^{(s)}$ value for each species $s$. An exception was made for evaluation of the upsampling strategy, for which raw neural network outputs $w_c^{(\text{raw})}$ were used instead. For each compartment $c$, the quality metrics were computed as follows:

$$\text{Precision}_c = \frac{\text{TP}_c}{\text{TP}_c + \text{FP}_c}$$

$$\text{Recall}_c = \frac{\text{TP}_c}{\text{TP}_c + \text{FN}_c}$$

$$F_{1c} = 2 \times \frac{\text{precision}_c \times \text{recall}_c}{\text{precision}_c + \text{recall}_c}.$$

For each compartment $c$, $\text{TP}_c$ is the number of true positives (species correctly predicted to compartment $c$), $\text{FP}_c$ is the number of false positives (species predicted to compartment $c$ but originally annotated to a different compartment $\neq c$) and $\text{FN}_c$ is the number of false negatives (species originally annotated to compartment $c$ but predicted to a different compartment $\neq c$). The predicted compartment for each marker species was determined by selecting the compartment with the highest $CC_c^{(s)}$ value.

To assess overall performance across all compartments, $F_1$ scores were weighted by the number of marker proteins evaluated in each compartment:

$$F_{1\text{weighted}} = \frac{\sum_{c=1}^{C} n_c F_{1c}}{\sum_{c=1}^{C} n_c},$$

where $n_c$ is the number of marker proteins used for evaluation in compartment $c$.

This approach ensures that compartments with more ground truth annotations contribute proportionally to the final quality scores.

**Lipid localization after processing.** To enable meaningful comparisons of lipid characteristics across different compartments, we accounted for differences in lipid structure between lipid classes (in particular, carbon chain length, degree of saturation and the number of substituted fatty acids) for an adequate comparison. For each lipid class $\mathcal{L}$ with the lipid species $k = 1,\ldots,K$ and compartment $c$, the weighted average feature value per fatty acid was calculated as:

$$\bar{q}_f^{(c,\mathcal{L})} = \frac{1}{f_\mathcal{L} \times \sum_{k=1}^{K} (n_k^{(\mathcal{L})} CC_k^{(c,\mathcal{L})})} \times \sum_{q \in Q_c} (q \times \sum_{j=1}^{J} (n_j^\mathcal{L} CC_j)),$$

where:

$\bar{q}_f^{(c,\mathcal{L})}$ is the weighted mean feature (either chain length or saturation) per fatty acid ($f$) for compartment $c$ and lipid class $\mathcal{L}$.
$k$ is a lipid species from the lipid class $\mathcal{L}$.
$K$ are all lipid species from the lipid class $\mathcal{L}$ detected for compartment $c$.
$f_\mathcal{L}$ is the number of fatty acids per lipid species for class $\mathcal{L}$.
$n_k^{(\mathcal{L})}$ is the total molar amount of lipid species $k$ from lipid class $\mathcal{L}$.
$CC_k^{(c,\mathcal{L})}$ is the CC for lipid species $k$ from lipid class $\mathcal{L}$ on compartment $c$.
$q$ is a feature from $Q$.
$Q_c$ are all features that were identified for lipid class $\mathcal{L}$ on compartment $c$ (example: identified lipid species $k$ for lipid class $\mathcal{L}$ included species with four, six and ten double bonds per fatty acid $f$, then $Q_c = \{4, 6, 10\}$).
$j$ is a lipid species from $J$.
$J$ are all lipid species from lipid class $\mathcal{L}$ that fit criteria $q$ (example: $q = 6$ means $J$ are all lipid species $k$ from lipid class $\mathcal{L}$ with six double bonds per fatty acid $f$).
$n_j^\mathcal{L}$ is the total molar amount of lipid species $j$ from lipid class $\mathcal{L}$.
$CC_j$ is the CC of lipid species $j$ on compartment $c$.

**C-COMPASS implementation.** C-COMPASS was developed under Python v.3.8.8. The graphical user interface of C-COMPASS was developed using PySimpleGUI (v.4.55.1) in combination with Tkinter (v.8.6). Quality metrics were calculated using scikit-learn (v.0.24.1), while neural network predictions were executed using Keras (v.2.10.0) alongside TensorFlow (v.2.10.0) and KerasTuner (v.1.4.5) for hyperparameter tuning. Additionally, C-COMPASS uses several libraries including NumPy (v.1.20.1), SciPy (v.1.6.2) and pandas (v.1.2.4).

**Image generation.** All figures were created using Adobe Illustrator v.24.3. Plots were generated in Python v.3.8.8 using the following packages: NumPy (v.1.20.1), pandas (v.1.2.4), seaborn (v.0.11.1),

Matplotlib (v.3.3.4), umap-learn (v.0.5.3), adjustText (v.0.8), SciPy (v.1.6.2), scikit-learn (v.0.24.1), UpSetPlot (v.0.9.0) and Plotly (v.5.22.0). Heatmaps were also created with Perseus v.1.6.1.5.0.

## Reporting summary
Further information on research design is available in the Nature Portfolio Reporting Summary linked to this article.

## Data availability
Proteomics data generated in this study are available in the PRIDE repository (ProteomeXchange) under the accession number PXD056457. Proteomics data and C-COMPASS outputs are provided as Extended Data Table 1. An example dataset based on simulated fractionation and total proteome data as well as a ready C-COMPASS session are provided in the Supplementary Information for demonstration purposes. The R code used to create BANDLE predictions is provided in the Supplementary Information as well. Source data are provided with this paper.

## Code availability
C-COMPASS software is on GitHub at https://github.com/ICB-DCM/C-COMPASS. Additionally, C-COMPASS releases are deposited at Zenodo (https://doi.org/10.5281/zenodo.14712134)[62]. Software documentation is available at https://c-compass.readthedocs.io/. An example session for C-COMPASS, along with example data, is available on Zenodo (https://zenodo.org/records/13901167)[63]. These data can be used to load an already processed session or to import and process the data, allowing users to reproduce the results.

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

## Acknowledgements
We thank R. Farese Jr, F. Klingelhuber, K. Lilley and all members of the Krahmer and Hasenauer laboratories for discussion. We thank D. Brandt and S. Ribicic for technical assistance. These studies were supported by the German Research Foundation DFG (Emmy Noether KR5166/2 to N.K.; FOR5815 to N.K.; BATenergy TRR 333/1, 450149205 to N.K. and J.H.; Germany's Excellence Strategy 390685813, EXC 2047 and 390 873048, EXC 2151 to J.H.; TRR296, TRR152, SFB1123 and GRK 2816/1 to T.D.M.), the European Foundation for the Study of Diabetes (Future Leader Award NNF20SA0066171 to N.K.) and the University of Bonn via the Schlegel professorship (to J.H.). T.D.M. received funding from the European Research Council (ERC-CoG Trusted no. 101044445) and the German Center for Diabetes Research (DZD e.V.). The funders had no role in study design, data collection and analysis, decision to publish or preparation of the manuscript.

## Author contributions
N.K. and D.T.H. conceived the project. N.K., D.T.H. and X.C. designed experiments. D.T.H. and X.M. performed organelle fractionation. E.-M.T. performed proteomic sample preparation. D.T.H. conducted proteomic analyses. D.T.H., J.H. and N.K. designed the software structure as well as the analysis and validation strategies. D.T.H. and M.G. conceptualized data preprocessing strategies. D.T.H. and D.W. implemented software. N.K. and D.T.H. analyzed data. P.K., J.P.S. and D.T.H. carried out comparative analysis of available tools. C.K. and M.J.G. performed lipidomic analysis and data analysis. T.D.M. contributed to discussions. N.K. and D.T.H. wrote the manuscript.

## Funding

## Competing interests
T.D.M. receives research funding by Novo Nordisk and has received speaking fees from Novo Nordisk, Eli Lilly, Boehringer Ingelheim, Merck, AstraZeneca and Mercodia. All other authors declare no competing interests.

## Additional information
**Extended data** is available for this paper at https://doi.org/10.1038/s41592-025-02880-3.

**Correspondence and requests for materials** should be addressed to Natalie Krahmer.

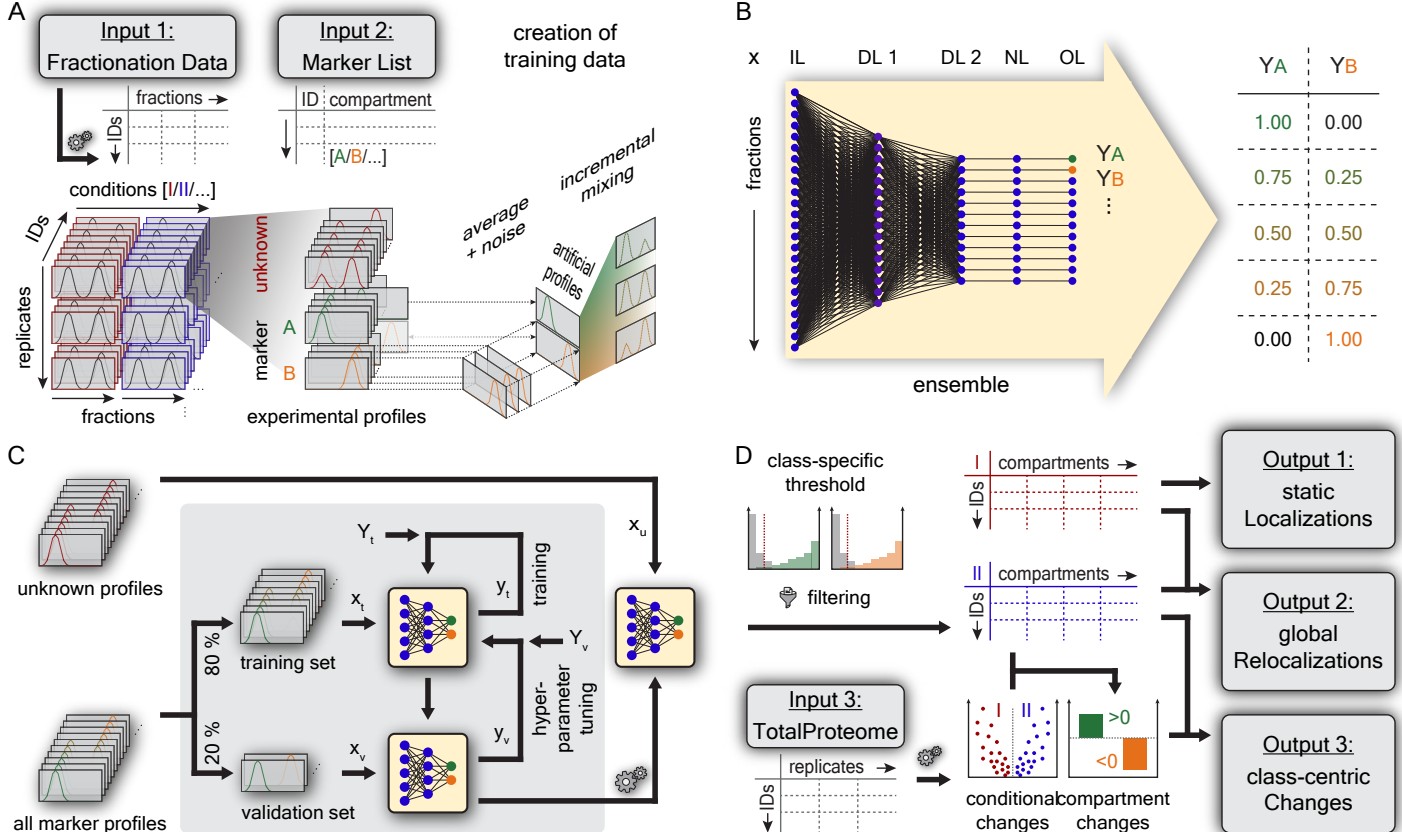

**Extended Data Fig. 1 | The C-COMPASS framework. (A)** Fractionation data is imported into C-COMPASS (Input 1) and filtered for valid values. Profiles are scaled to an area of 1, creating a data matrix with valid profiles across replicates and conditions (red and blue). These protein lists are matched with a marker list (Input 2) from literature, containing compatible identifiers and main compartment assignments. To balance marker batch sizes, artificial profiles are created by adding random noise to randomly selected profiles from underrepresented classes. Marker profiles from two compartments are incrementally mixed to represent dual localizations. **(B)** The input layer (IL) of the neural network matches the number of fractions in a replicate. Two hidden dense layers follow: the first layer (DL1) size is optimized via hyperparameter tuning, and the second layer (DL2) size corresponds to the number of compartments. A normalization layer (NL) ensures neuron data sums to 1, and the output layer

(OL) provides the network output (Y) for each compartment. **(C)** The network topology is optimized through hyperparameter tuning. Marker profiles are split into a training set to compare actual values (Yt) with network output (yt) and a validation set to evaluate performance on unseen data by comparing actual values (Yv) with network output (yv). The best topology is used for processing the entire dataset, including marker and unknown profiles. **(D)** Neural network output (y) is filtered by setting a threshold to cover 95% of false positive values per compartment, setting all values below the threshold to 0, and re-normalized to sum to 1. The results provide CC values for all compartments (Output 1), which can be compared across biological conditions to estimate relocalizations (Output 2). Total proteome values (Input 3) can be used to assess changes in protein expression levels and compartment abundance, normalizing the relocalization information to generate class-centric statistics (Output 3).

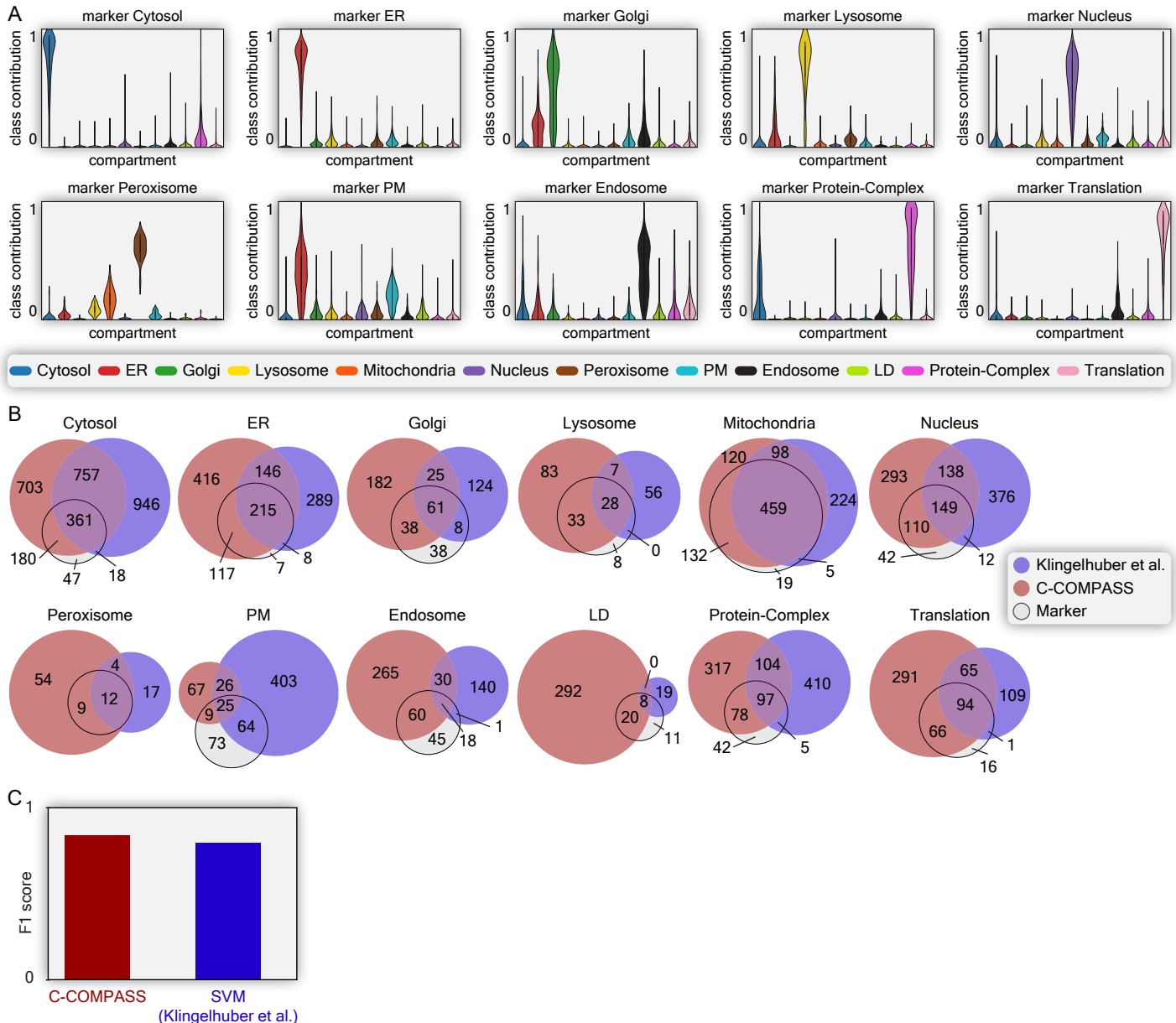

**Extended Data Fig. 2 | Comparison of C-COMPASS performance versus SVM predictions.** (**A**) Violin plots show raw neural network outputs for different compartments derived from their marker proteins. Violins are width-scaled, inner lines represent quartiles, and dots represent median values. (**B**) Venn diagrams comparing proteins with the highest neural network output values for different compartments from C-COMPASS (red) with original predictions by Klingelhuber et al.18 (blue). Colorless circles show intersections with marker proteins. (**C**) Bar plot shows the F1 scores for the original predictions and C-COMPASS predictions based on highest CC values.

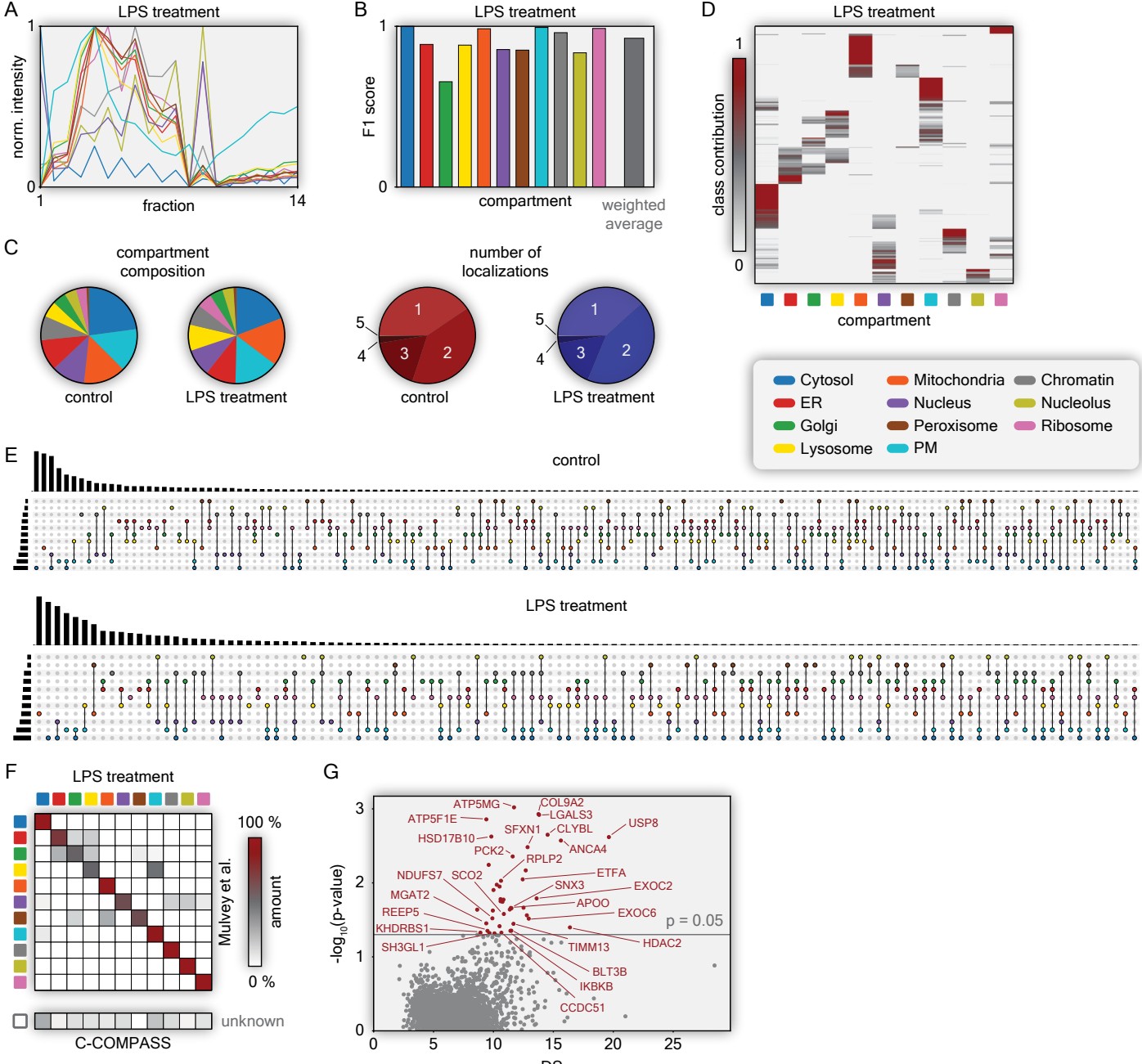

**Extended Data Fig. 3 | Application of C-COMPASS to the HyperLOPIT workflow.** (**A**) Profiles for compartment markers, with line values representing median values derived from median profiles across replicates for LPS treatment. (**B**) Bar plot showing F1 scores for each compartment for LPS treatment. Main organelle assignments were determined using the highest CC value per protein. The grey bar represents the average F1 value weighted by batch sizes of marker proteins. (**C**) Numbers of main localization assignments to compartments, and numbers of proteins assigned to single or multiple compartments for both conditions.

(**D**) Heatmap of CC values across all compartments for LPS treatment. (**E**) Lists of all identified compartment combinations and their frequencies for both conditions. (**F**) Correlation matrix comparing original compartment associations by Mulvey et al. with C-COMPASS main compartment predictions for LPS treatment. (**G**) Scatter plot displaying p-values from a two-sided Welch's t-test based on ensemble network output values against DS values for proteins with RLS > 1, highlighting the most reliable outliers for the comparison between control and LPS treatment. The grey dashed line indicates a p-value of 0.05.

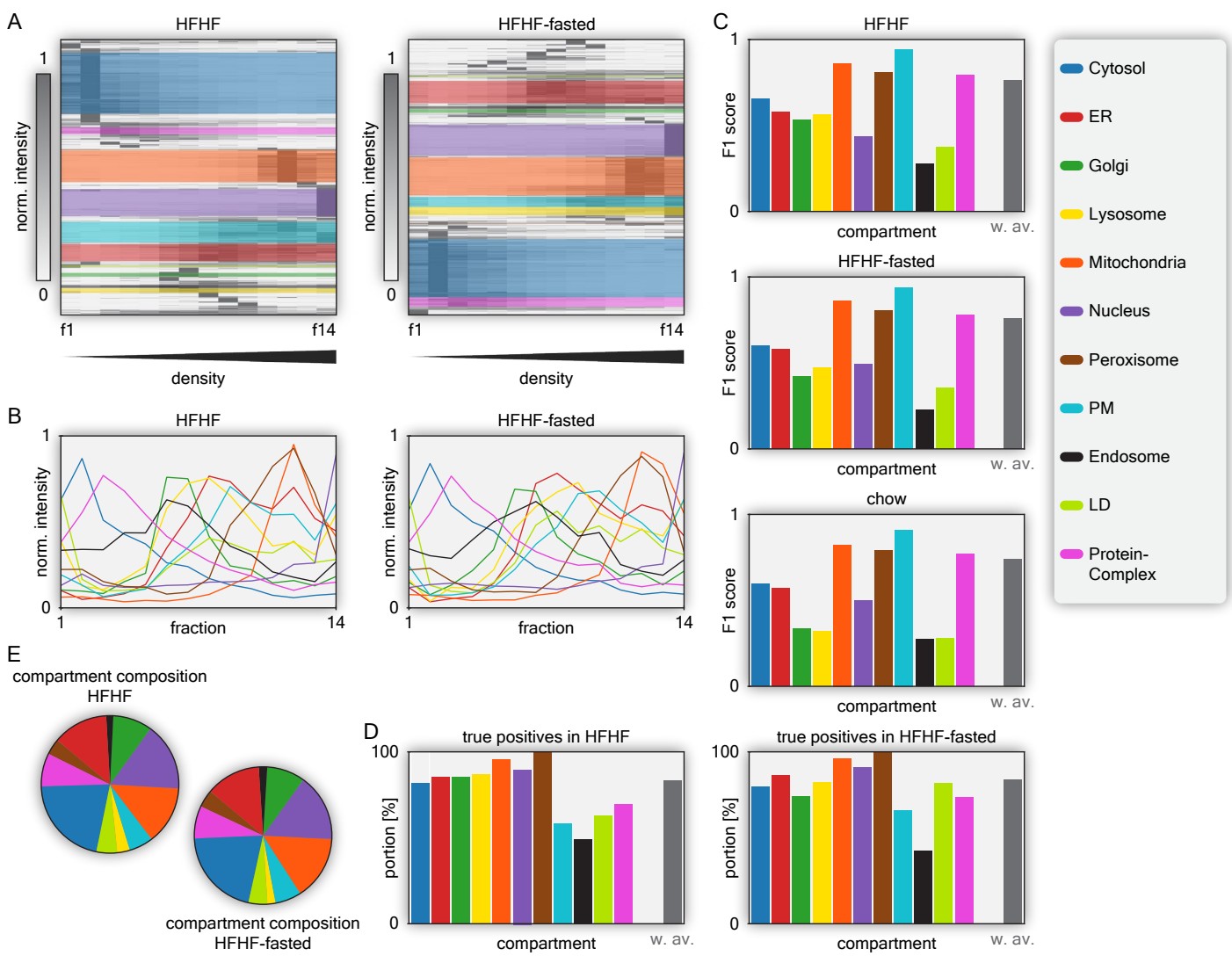

**Extended Data Fig. 4 | An organelle map of humanized liver across metabolic conditions.** (**A**) Hierarchical clustering heatmap of protein profiles across 14 fractions for HFHF and HFHF-fasted conditions. Colored areas indicate clusters with the highest enrichment scores for specific compartment marker annotations. (**B**) Profiles of compartment markers, with line values representing median values derived from median profiles across replicates for HFHF and HFHF-fasted conditions. (**C**) Bar plots showing F1 scores for each compartment across all conditions. Main organelle assignments were determined using the highest CC value per protein. The grey bar represents the average F1 value weighted by batch sizes of marker proteins. (**D**) Bar plot showing the number of true positive localization predictions by checking for positive CC values for each compartment for HFHF and HFHF-fasted conditions. The grey bar represents the average true positive rate weighted by batch sizes of marker proteins. (**E**) Numbers of main localization assignments for HFHF and HFHF-fasted conditions.

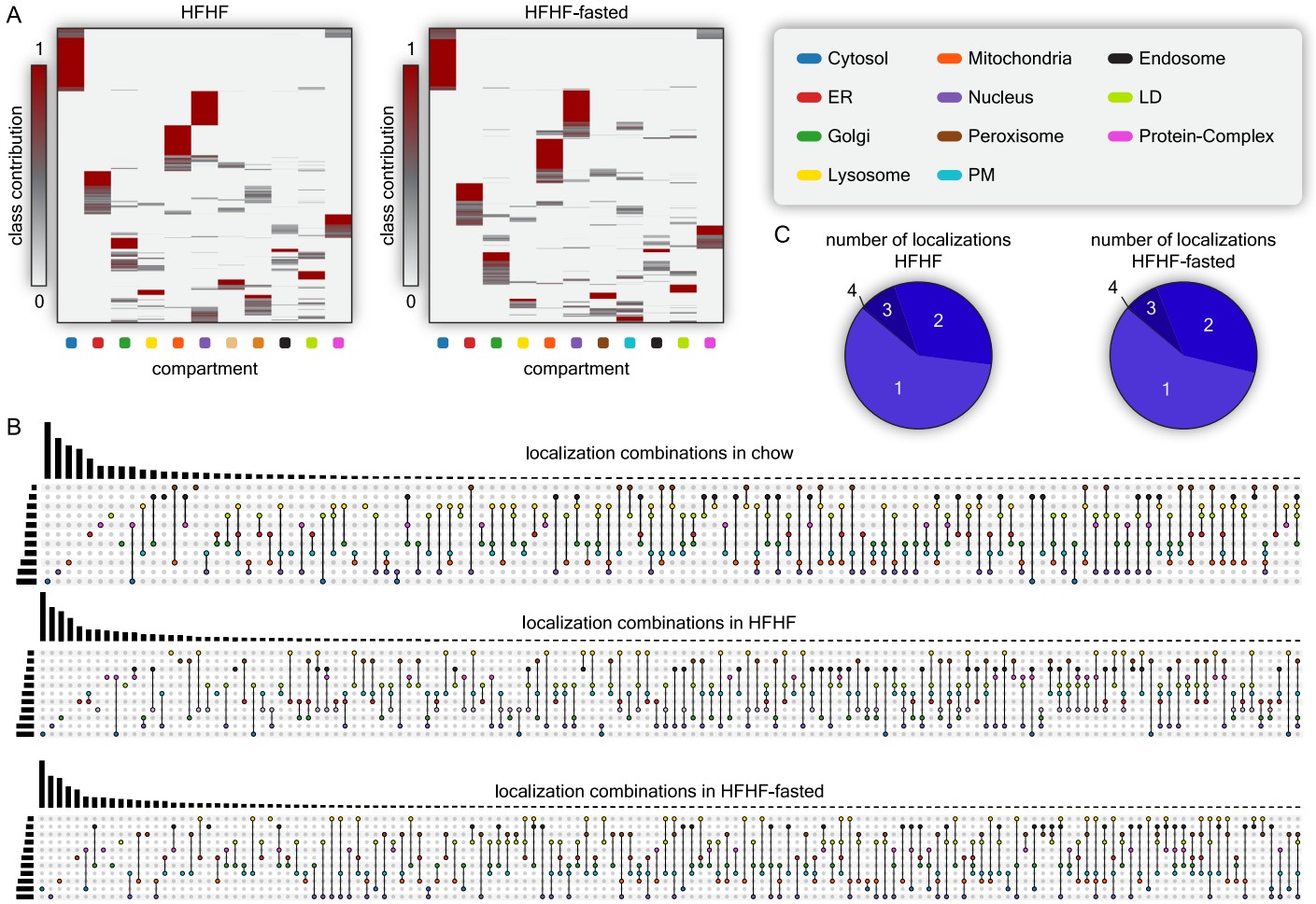

**Extended Data Fig. 5 | Compartment association and organelle map of humanized liver.** (**A**) Heatmaps of CC values across all compartments for HFHF and HFHF-fasted conditions. (**B**) Lists of all identified compartment combinations and their frequencies for chow, HFHF, and HFHF-fasted conditions. (**C**) Numbers of proteins assigned to single or multiple compartments for HFHF and HFHF-fasted conditions.

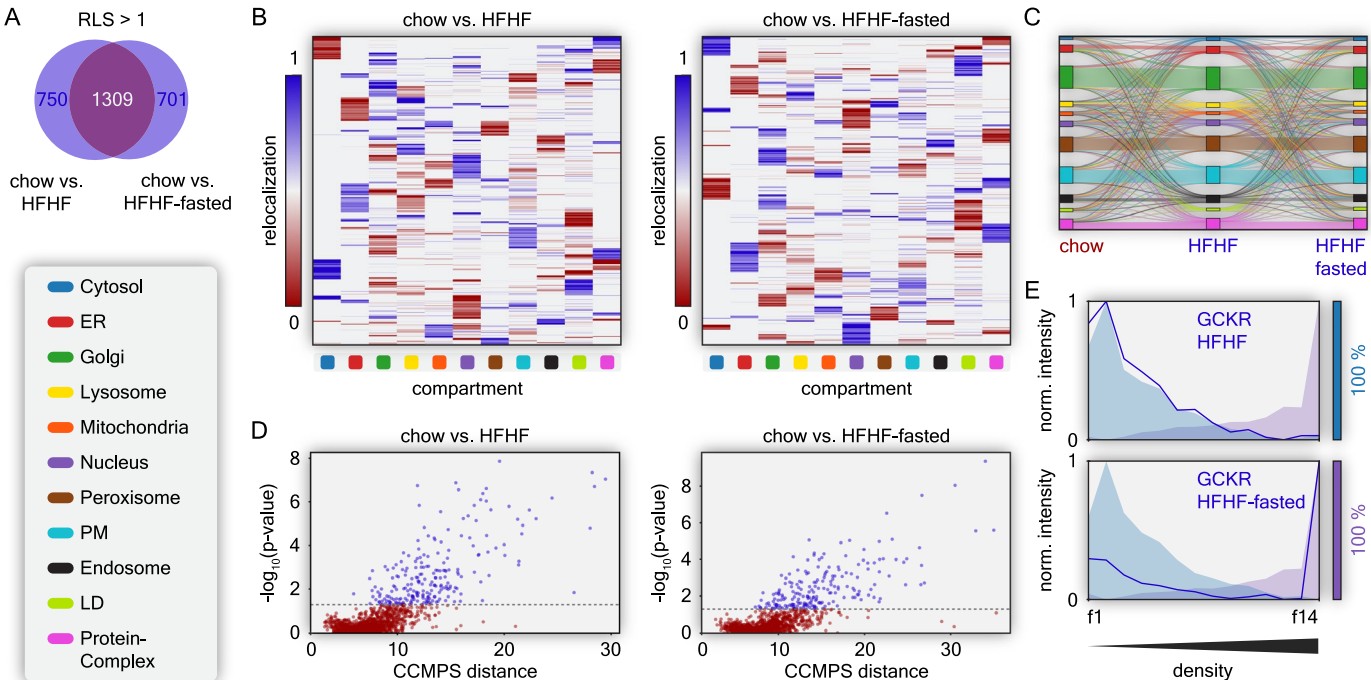

**Extended Data Fig. 6 | Protein relocalizations and compartmental changes across different metabolic conditions.** (**A**) Venn diagram showing proteins with RLS > 1 for the comparisons chow vs. HFHF and chow vs. HFHF-fasted. (**B**) Heatmaps of RL values for each compartment across proteins with RLS > 1 for the comparisons chow vs. HFHF and chow vs. HFHF-fasted. (**C**) Sankey diagram illustrating localization changes from chow to HFHF to HFHF-fasted across all compartments, including non-relocalizing proteins, maintaining the origin and target for each relocalization. (**D**) Scatter plots displaying p-values from two-sided Welch's t-tests based on ensemble network output values against DS values for proteins with RLS > 1, highlighting the most reliable outliers for the comparisons chow vs. HFHF and chow vs. HFHF-fasted. (**E**) Normalized intensity profiles for GCKR (line) and median marker profiles for Cytosol and Nucleus (areas) for HFHF and HFHF-fasted conditions. Bars are indicating the localization of GCKR.

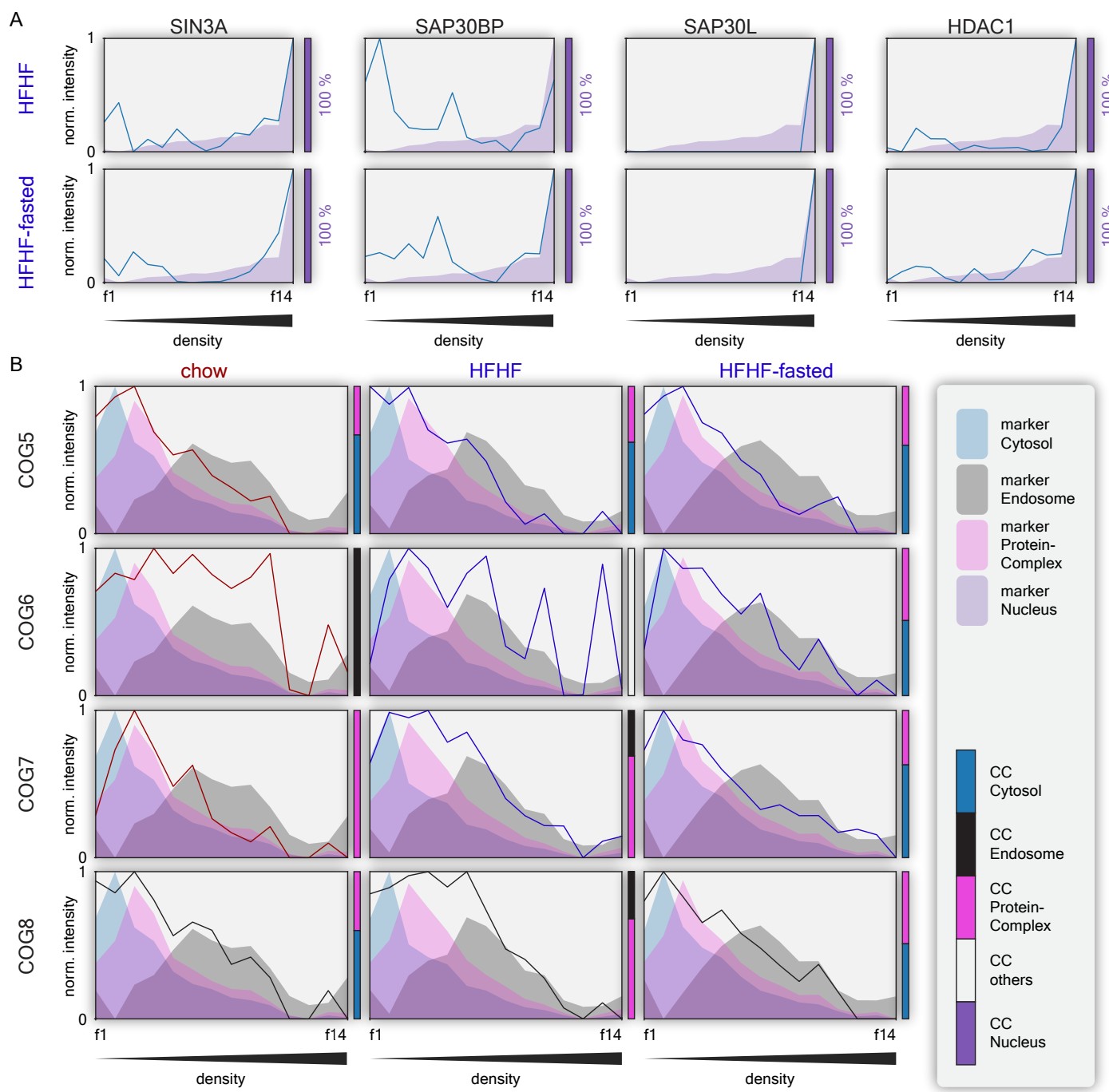

**Extended Data Fig. 7 | Metabolic state induced spatial re-organization of COG complex. (A)-(B)** Protein profiles (lines) of selected candidates for chow, HFHF, and HFHF-fasted, overlaid with median marker profiles (areas). Bars on the right show the CC distributions for these proteins across the displayed compartments.

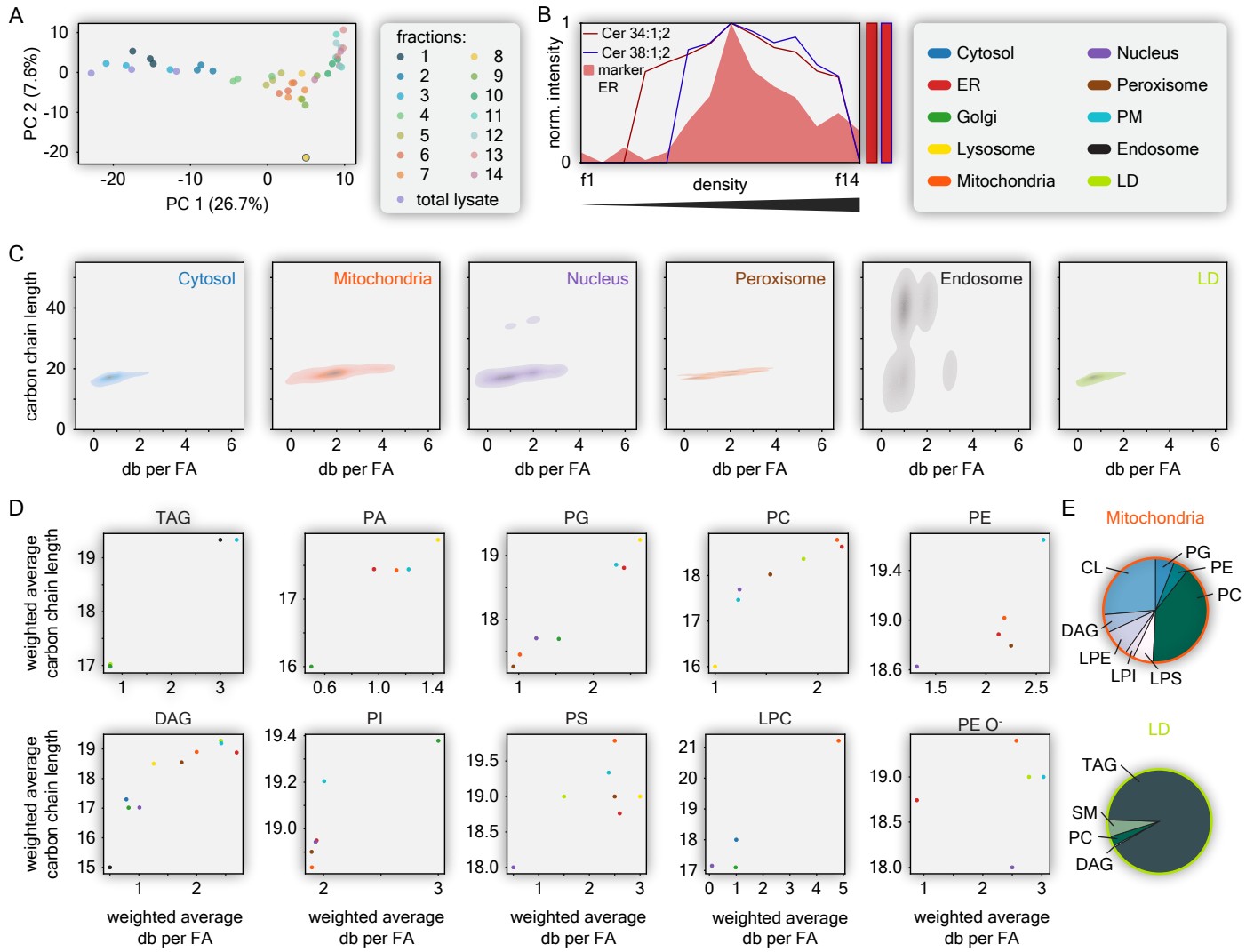

**Extended Data Fig. 8 | Modeling of organelle lipid composition and analysis of compartment specific phospholipid compositions.** (**A**) Principal Component Analysis of all measured lipidomics samples. (**B**) Median protein marker profiles for the ER (area), overlaid with distinct lipid profiles (lines). (**C**) KDE plots showing distribution of carbon chain lengths and the number of double bonds per fatty acid across selected compartments (bandwidth adjustment = 0.5).

Values are normalized by CC values and weighted by intensities in the total lysate. (**D**) Distribution of average carbon chain lengths and number of double bonds per fatty acid across different compartments, shown for various lipid classes. Average values are weighted by CC values and intensities. (**E**) Total numbers of findings for various lipid classes in Mitochondria and LDs.

**Extended Data Table 1 | Comparison of computational tools for subcellular localization prediction and spatial proteomics analysis**

| | METAMASS | SUBCELL-BARCODE | TRANSPIRE | BANDLE | DOM-ABC | C-COMPASS |
|---|---|---|---|---|---|---|
| **RELEASE** | 2016 | 2019/2022 | 2020 | 2022 | 2023 | 2025 |
| **PLATFORM** | R, MS Excel | R | Python | R | Python | Python |
| **USABILITY** | GitHub-hosted R Package (requires devtools for installation), or Excel-Sheet | Bioconductor R Package | Python Package (install from GitHub) | Bioconductor R Package | Web Application (install via PyPI, or accessible via webserver), and non-GUI application (install via PyPI) | GUI Application (install via PyPI, or run as standalone executable) |
| **PREDICTION MODEL** | K-means clustering | SVM | SVGPC* | Bayesian Model | SVM | Neural Network |
| **PREDICTION FORMAT** | Multi-class Classification | Probabalistic multi-class Classification | Multi-class Classification | Multi-class Classification | Multi-class Classification | Quantitative multi-class, multi-label regression |
| **MULTI LOCALIZATION ANALYSIS** | No | Qualitative through probabilities | No | Qualitative through uncertainty estimation and probabilites | No | Quantitative Predictions |
| **RE-LOCALIZATION PREDICTION** | Qualitative output comparison | Qualitative output comparison | Probabalistic Predictions | Qualitative against control | Qualitative | Quantitative Predictions |
| **REPLICATE HANDLING** | Not modeled | Processes replicates, no variability modeling | Processes replicates, no variability modeling | Models replicate variation | Assess reproducibility across replicates | Processes replicates independently to model variations |
| **EXPERIMENTAL VALIDATION** | Density Gradient Centrifugation | SubCellBarCode (based on Differential Centrifugation) | Differential Centrifugation, Density Gradient Centrifugation | HyperLOPIT | Differential Centrifugation | Density Gradient Centrifugation, HyperLOPIT |
| **APPLIED DATASETS** | Proteomics from cells | Proteomics from cells | Proteomics from cells | Proteomics from cells, Transcriptomics | Proteomics from cells | Proteomics & Lipidomics from tissues |
| **KEY STRENGTHS** | User-friendly Excel sheet | Fully automated pipeline | Relocalization Prediction | Detection of Multi-Localization probabilities | Easy-to-use Website | User-friendly GUI, Quantitative Predictions, Multi-Relocalization Prediction, Flexibility in experimental design |
| **LIMITATIONS** | Excel sheet created with Excel 2007, no replicate combination | Limited to an exactly defined experimental design | Limited to relocalization predictions, no support for single condition analysis | Analysis only possible for control against one condition, Extensive manual data pre-processing required | Pre-processing and parameter configuration can be challenging for novices | Compatibility for MacOS still with limitas for GUI |

This table summarizes the core features of the six representative tools MetaMass, SubCellBarCode, TRANSPIRE, BANDLE, DOM-ABC, and C-COMPASS, including platform, usability, prediction model and format, support for multilocalization and relocalization analysis, and application scope. *SVGPC: Stochastic Variational Gaussian Process Classifier.

# Reporting Summary

## Statistics

For all statistical analyses, confirm that the following items are present in the figure legend, table legend, main text, or Methods section.

| n/a | Confirmed | |
|---|---|---|
| ☐ | ☒ | The exact sample size (*n*) for each experimental group/condition, given as a discrete number and unit of measurement |
| ☐ | ☒ | A statement on whether measurements were taken from distinct samples or whether the same sample was measured repeatedly |
| ☐ | ☒ | The statistical test(s) used AND whether they are one- or two-sided<br>*Only common tests should be described solely by name; describe more complex techniques in the Methods section.* |
| ☒ | ☐ | A description of all covariates tested |
| ☐ | ☒ | A description of any assumptions or corrections, such as tests of normality and adjustment for multiple comparisons |
| ☐ | ☒ | A full description of the statistical parameters including central tendency (e.g. means) or other basic estimates (e.g. regression coefficient) AND variation (e.g. standard deviation) or associated estimates of uncertainty (e.g. confidence intervals) |
| ☐ | ☒ | For null hypothesis testing, the test statistic (e.g. *F*, *t*, *r*) with confidence intervals, effect sizes, degrees of freedom and *P* value noted<br>*Give P values as exact values whenever suitable.* |
| ☐ | ☒ | For Bayesian analysis, information on the choice of priors and Markov chain Monte Carlo settings |
| ☐ | ☒ | For hierarchical and complex designs, identification of the appropriate level for tests and full reporting of outcomes |
| ☐ | ☒ | Estimates of effect sizes (e.g. Cohen's *d*, Pearson's *r*), indicating how they were calculated |

*Our web collection on statistics for biologists contains articles on many of the points above.*

## Software and code

Policy information about availability of computer code

| Data collection | Proteomics raw data were processed by Scpectronaut 18 (Copernicus, Biognosys). Lipidomics data were analyzed with in-house developed lipid identification software based on LipidXplorer. |
|---|---|
| Data analysis | Microsoft Excel, Perseus 1.6.1.5.0, Python 3.8.8 (including NumPy (1.20.1), pandas (1.2.4), seaborn (0.11.1), Matplotlib (3.3.4), umap-learn (0.5.3), ScipPy (1.6.2), scikit-learn (0.24.1), UpSetPlot (0.9.0), Plotly (5.22.0), Keras (2.10.0), KerasTuner (1.4.5)) |

For manuscripts utilizing custom algorithms or software that are central to the research but not yet described in published literature, software must be made available to editors and reviewers. We strongly encourage code deposition in a community repository (e.g. GitHub). See the Nature Portfolio guidelines for submitting code & software for further information.

## Data

Policy information about availability of data

All manuscripts must include a data availability statement. This statement should provide the following information, where applicable:

- Accession codes, unique identifiers, or web links for publicly available datasets
- A description of any restrictions on data availability
- For clinical datasets or third party data, please ensure that the statement adheres to our policy

Proteomics data generated in this study are available on the PRIDE repository (ProteomeXchange) under the accession number PXD056457. Proteomics data and C-COMPASS outputs are provided as Extended Data Tables and Raw data for all figure is provided as source files.

Code availability

The C-COMPASS software is developed on GitHub at https://github.com/ICB-DCM/C-COMPASS. Additionally, C-COMPASS releases are deposited at Zenodo (https://doi.org/10.5281/zenodo.14712134). Software documentation is available at https://c-compass.readthedocs.io/. An example dataset based on simulated fractionation and total proteome data, as well as a ready C-COMPASS session are provided in the supplementary material for demonstration purpose. The R code used to create BANDLE predictions is provided in the supplementary material as well.

An example session for C-COMPASS, along with example data, is available on Zenodo at https://zenodo.org/records/13901167. This data can be used to load an already processed session or to import and process the data, allowing users to reproduce the results.

## Human research participants

Policy information about studies involving human research participants and Sex and Gender in Research.

| | |
|---|---|
| Reporting on sex and gender | *Use the terms sex (biological attribute) and gender (shaped by social and cultural circumstances) carefully in order to avoid confusing both terms. Indicate if findings apply to only one sex or gender; describe whether sex and gender were considered in study design whether sex and/or gender was determined based on self-reporting or assigned and methods used. Provide in the source data disaggregated sex and gender data where this information has been collected, and consent has been obtained for sharing of individual-level data; provide overall numbers in this Reporting Summary. Please state if this information has not been collected. Report sex- and gender-based analyses where performed, justify reasons for lack of sex- and gender-based analysis.* |
| Population characteristics | *Describe the covariate-relevant population characteristics of the human research participants (e.g. age, genotypic information, past and current diagnosis and treatment categories). If you filled out the behavioural & social sciences study design questions and have nothing to add here, write "See above."* |
| Recruitment | *Describe how participants were recruited. Outline any potential self-selection bias or other biases that may be present and how these are likely to impact results.* |
| Ethics oversight | *Identify the organization(s) that approved the study protocol.* |

Note that full information on the approval of the study protocol must also be provided in the manuscript.

## Field-specific reporting

Please select the one below that is the best fit for your research. If you are not sure, read the appropriate sections before making your selection.

☒ Life sciences  ☐ Behavioural & social sciences  ☐ Ecological, evolutionary & environmental sciences

For a reference copy of the document with all sections, see nature.com/documents/nr-reporting-summary-flat.pdf

## Life sciences study design

All studies must disclose on these points even when the disclosure is negative.

| | |
|---|---|
| Sample size | Sample size was determined based on experience on previous studies and what is commonly used in the field |
| Data exclusions | No samples were excluded |
| Replication | For animal experiments at least three biological replicates were performed |
| Randomization | No randomization possible |
| Blinding | No blinding possible |

## Reporting for specific materials, systems and methods

We require information from authors about some types of materials, experimental systems and methods used in many studies. Here, indicate whether each material, system or method listed is relevant to your study. If you are not sure if a list item applies to your research, read the appropriate section before selecting a response.

## Materials & experimental systems

| n/a | Involved in the study |
|---|---|
| ☒ ☐ | Antibodies |
| ☒ ☐ | Eukaryotic cell lines |
| ☒ ☐ | Palaeontology and archaeology |
| ☐ ☒ | Animals and other organisms |
| ☒ ☐ | Clinical data |
| ☒ ☐ | Dual use research of concern |

## Methods

| n/a | Involved in the study |
|---|---|
| ☒ ☐ | ChIP-seq |
| ☒ ☐ | Flow cytometry |
| ☒ ☐ | MRI-based neuroimaging |

# Animals and other research organisms

Policy information about studies involving animals; ARRIVE guidelines recommended for reporting animal research, and Sex and Gender in Research

| | |
|---|---|
| Laboratory animals | 6-week-old female C57BL/6N FRG KO mice, 43-week-old male C57BL/6J |
| Wild animals | *Provide details on animals observed in or captured in the field; report species and age where possible. Describe how animals were caught and transported and what happened to captive animals after the study (if killed, explain why and describe method; if released, say where and when) OR state that the study did not involve wild animals.* |
| Reporting on sex | Only male mice were used for experiments. |
| Field-collected samples | *For laboratory work with field-collected samples, describe all relevant parameters such as housing, maintenance, temperature, photoperiod and end-of-experiment protocol OR state that the study did not involve samples collected from the field.* |
| Ethics oversight | Guide for the Care and Use of Laboratory Animals, Eighth Edition, and the AVMA Guidelines for Euthanasia of Animals: 2013 or 2020 Editions, and local animal ethics committee of the Government of Upper Bavaria, Germany. |

Note that full information on the approval of the study protocol must also be provided in the manuscript.

