## [Peer Review File · Nature Methods]

C-COMPASS: A user-friendly neural network tool profiles cell compartments at protein and lipid levels

Corresponding Author: Dr Natalie Krahmer

Version 0:

Decision Letter:

17th Dec 2024

Dear Natalie,

Your Article entitled "C-COMPASS: A user-friendly neural network tool profiles cell compartments at protein and lipid levels" has now been seen by 2 reviewers, whose comments are attached. While they find your work of some potential interest, they have raised concerns which in our view are sufficiently important that they preclude publication of the work in Nature Methods.

We will consider looking at a revised manuscript only if further experimental data allow you to address all the major criticisms of the reviewers (unless, of course, something similar has by then been accepted at Nature Methods or appeared elsewhere). This includes submission or publication of a portion of this work somewhere else.

The required new experiments and data include, but are not limited to, more thorough benchmarking against other methods in this space, better characterization of the method, and further validation of the results. We hope you understand that until we have read the revised paper in its entirety we cannot promise that it will be sent back for peer-review.

Or, if you like, your paper may be appropriate for another journal in the Nature Portfolio. If you wish to explore the journals and transfer your manuscript please use our manuscript transfer portal. You will not have to re-supply manuscript metadata and files, unless you wish to make modifications. For more information, please see our [manuscript transfer FAQ](http://www.nature.com/authors/author_resources/transfer_manuscripts.html?WT.mc_id=EMI_NPG_1511_AUTHORTRANSF&WT.ec_id=AUTHOR) page.

If you are interested in revising this manuscript for submission to Nature Methods in the future, please contact me to discuss your appeal before making any revisions. Otherwise, we hope that you find the reviewers' comments helpful when preparing your paper for submission elsewhere.

Sincerely,
Arunima

Arunima Singh, Ph.D.
Senior Editor
Nature Methods

Reviewers' Comments:

Reviewer #1:

Remarks to the Author:

The manuscript by Natalie Kramer and coworkers presents C-COMPASS, an open-source software tool that utilizes neural network-based regression models to predict the spatial distribution of proteins and lipids across cellular compartments. The tool can analyze proteomics data obtained from organellar fractionation methods such as Protein Correlation Profiling (PCP) or Localization of Organelle Proteins by Isotope Tagging (LOPIT). The tool, which is provided as a user-friendly graphical

interface, requires input data on protein abundances across fractions and a set of predefined or custom organelle marker proteins. It was developed with the aim of overcoming some of the limitations of existing proteomics-based organellar profiling methods, which often struggle with proteins that localize to multiple compartments as well as with the integration of protein and lipid data.

MAJOR POINTS:

- Although the authors present an interesting tool to perform differential protein location analysis, they fail to present properly its differences and advantages against previous tools. For instance, in the Discussion, authors claim that "Validated with multiple datasets, C-COMPASS accurately predicts multi organelle localization and models organelle composition under different conditions, outperforming earlier single-classification methods". However, no proper comparison is performed against already published algorithms for protein localization, both for single localization, such as MetaMass (Lund-Johansen, F., de la Rosa Carrillo, D., Mehta, A. et al. MetaMass, a tool for meta-analysis of subcellular proteomics data. *Nat Methods* 13, 837–840 (2016). <https://doi.org/10.1038/nmeth.3967>) or for multiple locations such as the Bayesian mixture model described by Crook and colleagues (Crook OM, Mulvey CM, Kirk PDW, Lilley KS, Gatto L. A Bayesian mixture modelling approach for spatial proteomics. *PLoS Comput Biol*. 2018 Nov 27;14(11):e1006516. doi: 10.1371/journal.pcbi.1006516.) or the BUNDLE algorithm (Crook, O.M., Davies, C.T.R., Breckels, L.M. et al. Inferring differential subcellular localisation in comparative spatial proteomics using BUNDLE. *Nat Commun* 13, 5948 (2022). <https://doi.org/10.1038/s41467-022-33570-9>). Neither, do the authors compare C-COMPASS against the differential localization scoring used in other manuscripts from the field, such as: "Martinez-Val, A., Bekker-Jensen, D.B., Steigerwald, S. et al. Spatial-proteomics reveals phospho-signaling dynamics at subcellular resolution. *Nat Commun* 12, 7113 (2021). <https://doi.org/10.1038/s41467-021-27398-y>" or "Itzhak DN., Tyanova S., Cox J, Borner, GHH (2016) Global, quantitative and dynamic mapping of protein subcellular localization *eLife* 5:e16950. <https://doi.org/10.7554/eLife.16950>". A much more thorough benchmark needs to be done.

- Did the results of the study using C-COMPASS significantly deviate from other published tools?

- If so, what findings were specific to C-COMPASS and how did the authors ensure that the results of C-COMPASS were correct?

- And what did C-COMPASS miss that other tools found?

- In line 191, and again in line 241, the authors mentioned a reduction in the number of fractions used for organelle fractionation. They give some hints that such reduction has an effect on increase organelle overlap, but this effect is not properly assessed. What would be the smallest number of fractions required for an accurate performance of this strategy? Also, in this regard, would this strategy work for other subcellular proteomics strategies, which do not rely on organelle purification but rather on sequential enrichment?

- In line 123, the authors mention the necessity to use total proteome information. Is this strictly required? If so, it needs to be clarified that for using this tool, total proteome is required to be measured in parallel to organelle fractions. Also, in this regard; should the total proteome have the same depth as the organelle fractionation proteome? Meaning that all proteins profiled in the organelle fractions need to be identified and quantified in the total proteome as well, to be able to normalize them?

- If understood correctly, the algorithm in C-COMPASS relies on a prior list of expected organelles, with known or predefined markers. Does this mean that C-COMPASS cannot be applied to organelles, which are not defined among the ones used in this publication? For instance, can C-COMPASS analyze data of non-membrane protein condensates or stress granules?

- It is not clear, how the statistical significance of the protein relocations is calculated neither from the results nor from the methods sections? This needs to be described in much more detail. Moreover, it is not clear if multiple comparison corrections are performed in the analyses?

MINOR POINT:

- In figure 5B: which subcellular compartment does "Protein Complex" refer to?

- In the discussion, the authors mention: "Importantly, C-COMPASS extends beyond proteomics to include other omics levels such as lipidomics, transcriptomics, and metabolomics.". However, the publication only presents integration of proteomics and lipidomics. The authors should either demonstrate and benchmark the potential integration with transcriptomics, and metabolomics, or remove the sentence.

Reviewer #2:

Remarks to the Author:

Dear Editors, dear authors,

I congratulate the authors for this interesting piece of research. They provided a novel data analysis tool in the field of spatial proteomics by exploiting neural networks for assigning protein intensity profiles to cellular compartments. Their classification method further allows multiple localization, which apparently is not yet possible with the current state-of-the-art despite its biological relevance. Moreover, they leverage the output of their models to

further assess changes in these compartment distributions upon treatment or other experimental condition and attempt to assess organelle composition. To ensure reliability of their data, they carry out a thorough model training and validation procedure and prevent over-fitting. They further show that their method is not limited to proteomics analysis, but can also be applied to other omics modalities, here illustrated with a combined proteomics and lipidomics data set.

Disclaimer: I am not familiar with the spatial proteomics literature and will assume the authors provided an honest overview of the current state-of-the-art regarding the classification of biological feature expression into cellular compartments. Also, I am a computational biologist and don't have the required expertise to fully assess the experimental and technical procedure, as well as the knowledge to thoroughly assess the plausibility of the biological hypotheses regarding compartment localization of specific proteins/lipids that have been derived from the data analysis results presented in this work.

Decision on publication: I consider the content of this research project is well-suited for publication in this journal. However, I don't find it suitable for publication in its **current** state, as major and minor concerns need to be addressed first. I have three major concerns: clarity of the manuscript, software engineering, and reproducibility. I'll end this peer-review with minor comments that I deem easy to solve.

Major concern 1: clarity of the manuscript

I found several parts of the manuscript very unclear, leading me to guess what the authors mean or to ignore some parts of the manuscript. I'm convinced that investing some time improving clarity will dramatically increase the quality of the manuscript. Here is a comprehensive list of the parts that need a major revision.

- General comment: the Results section provides too many details that could be in the methods section, whereas the Methods section is too short, very difficult to read and lacks mathematical notation that would clarify what the authors attempt to compute.
- L142-145: quality assessment should also include the SVM results from the initial study. Currently, it is difficult to assess whether C-COMPASS performs better than SVM.
- L163: the authors state that 54% of the hits are confirmed by MitoCarta and this indicates improved accuracy? This reads to me as if 46% of the markers are false hits. Moreover, how does MitoCarta markers overlap with SVM results?
- L195: "...indicating high reliability in compartment predictions (Fig. 4c)". I agree, but what about the results by Mulvey et al.? Again, it is difficult to assess whether C-COMPASS performs better than the original study.
- Fig2, L280-282, L654-660: exploring the organelle composition seems very interesting, but I did not understand it. The authors should clarify the biological question they attempt to answer and how their approach tackles the question. After reading several times the part in the Methods section, I am still unable to grasp the procedure. The Methods part should be completely rewritten, and I suggest to make use of mathematical notation to avoid ambiguity. Finally, exploiting the organelle composition results with a simple cluster analysis did not convince me of its relevance.
- L634-639: I can guess what the authors mean after several reads, but the text should be clarified, especially in the Methods sections.
- L648: student t-test is used, but given the restricted distribution range of RLs (0-2), I expect the data to be heteroskedastic. I would rather use a Welch t-test, or maybe even a non-parametric test if the authors cannot prove that the sample means are normally distributed.
- L641-645: it's not clear what data is used to compute precision, recall and F1, namely how the target values are defined. Also, the main text mentions false positives (L152), but how are false

positives defined? I guess that the protein marker list is used as ground truth data, but could it be the that the 2 FP are new markers that were not or wrongly described before? Also, scikit-learn weighted averaging is used, but how were the weights assigned?

Major concern 2: software engineering

Since the title mentions that it is "a user-friendly tool" and that publication focuses on the development of a novel computational approach, I'll be picky on the software engineering. In its ****current**** state, I foresee the software to have a limited lifetime and a limited user base, but I am convinced the authors can improve this. Here are the weaknesses that I found:

- The code is provided through a personal GitHub account, with a single contributor and showing little commit activity and no community activity (no issues, no pull request). I doubt the code has been reviewed. I don't mean to offend the main author which has invested a lot of effort in writing the software, but rather highlights that the software is prone to being abandoned upon project's completion. Could the author comment on their maintenance and lifecycle plan for management of the software?
- There is no unit testing, this again questions the long term maintainability of the code.
- The software executables are only provided for Windows, forcing users with MacOS, Linux or other OS to use the tools through the command line, which may have a significant impact on the user base. Since I use Linux, I have to install it manually.
- Manual installation of the module lacks documentation and a streamlined workflow. For instance, I tried installing the dependencies using the CCMPs/requirement.txt, but I hit an awful lot of version clashes which led me to abandon C-COMPASS installation...

Note that I saw the authors provided a Zenodo repository with an example session and data, but it is still private.

Major concern 3: reproducibility

From the journal's peer review guidelines, "Is the reporting of data and methodology sufficiently detailed and transparent to enable reproducing the results?" I currently have to answer no as the Methods section is unclear and there is no script/guidelines to reproduce any of the results presented in this work. Since I couldn't run the tool, I could not assess whether these results were all generated from within C-COMPASS itself.

Minor concerns:

- L92: could the authors provide some suggestions of good sources or data bases for marker lists? How reliable do they need to be?
- L110-112: I had to read the sentences several times to understand the underlying concept. I suggest rewriting it for clarity.
- L113: "renormalized to sum to one": is it renormalized per compartment across proteins or per protein across compartments? I guess it's the latter, but it should be clarified.
- L117: the ensemble approach comes out of the blue in the main text. After reading the methods section, I find ensembling an interesting approach to stabilize the predictions, so it's worth clarifying this in the results.
- L122-123: I don't understand this sentence.
- L142-145: I find the discussion of the results in Fig3a rather dry. It's striking that there is a benefit of upsampling for some compartments and no benefit in others, why? Is it because LDs, peroxysomes, endosome have less markers? Or is it because the marker patterns are more difficult to grasp hence require more profiles? The fact that some compartments are more difficult to model than others, regardless of upsampling or not, is interesting information worth to be discussed, in my opinion. The following paragraph seems to provide part of the explanation. I'm also not convinced that scores are improved for mitochondria given the error bars, I

wouldn't use it as an example.

- L244: "with some compartments, such as the endosomes and Golgi apparatus, showing lower accuracy". Golgi shows good results in Fig5d.

- L252-254: Concluding that there are relocalization events requires more evidence. This statement assumes the cellular composition of the livers across conditions to be similar, eg maybe the diet condition may impact the proportion of adipocytes or immune cells in the liver. I'm interested in the author's perspective on this and feel it needs to be added to the discussion.

- L374-375: the sentence should reference Fig S8e.

- L568: literature and theory has shown that imputation by zero is the worst strategy for MS-based proteomics data. Could the authors comment on this? Also, why using another strategy for the total proteome?

- L571: "Mean intensity values were used for analysis." Mean across what?

- L599-600: Is there a safeguard when the difference between the number of markers between most and least abundant compartment is substantial? In other words, are there some compartments that are automatically dropped if not enough markers?

- L606: two-compartment profiles are simulated at 3 mixing ratios, each represent 5% of the batch size. Could the authors comment why 5%? From the results presented, at least 30% of the proteins (and about 75% of the lipids) in the data sets seem to display multiple compartment profiles.

I realize this is a long list of painful comments to tackle, but I really hope the authors will be able to solve the current concerns as this work truly deserves to get published.

Best regards,
Christophe Vanderaa

Remarks on code availability:

See concerns #2 and #3 in the main review box.

** For Nature Portfolio general information and news for authors, see <http://npg.nature.com/authors>.

Version 1:

Decision Letter:

29th Jan 2025

Dear Natalie,

Thank you for your letter asking us to reconsider our decision on your Article, "C-COMPASS: A user-friendly neural network tool profiles cell compartments at protein and lipid levels". After careful consideration we have decided that we are willing to consider a revised version of your manuscript that addresses the issue of benchmarking and better characterization of the method and the other points raised by the reviewers.

* include a point-by-point response to our referees and to any editorial suggestions

* please underline/highlight any additions to the text or areas with other significant changes to facilitate review of the revised manuscript

* address the points listed described below to conform to our open science requirements

* ensure it complies with our general format requirements as set out in our guide to authors at www.nature.com/naturemethods

* resubmit all the necessary files electronically by using the link below to access your home page

Link Redacted

We hope to receive your revised paper within 6-7 weeks. If you cannot send it within this time, please let us know. In this event, we will still be happy to reconsider your paper at a later date so long as nothing similar has been accepted for publication at Nature Methods or published elsewhere.

OPEN SCIENCE REQUIREMENTS

REPORTING SUMMARY AND EDITORIAL POLICY CHECKLISTS

When revising your manuscript, please submit reporting summary and editorial policy checklists.

IMAGE INTEGRITY

DATA AVAILABILITY

CODE AVAILABILITY

Please include a "Code Availability" subsection in the Online Methods which details how your custom code is made available. Only in rare cases (where code is not central to the main conclusions of the paper) is the statement "available upon request" allowed (and reasons should be specified).

MATERIALS AVAILABILITY

SUPPLEMENTARY PROTOCOL

To help facilitate reproducibility and uptake of your method, we ask you to prepare a step-by-step Supplementary Protocol for the method described in this paper. We [encourage authors to share their step-by-step experimental protocols](https://www.nature.com/nature-research/editorial-policies/reporting-standards#protocols) on a protocol sharing platform of their choice and report the protocol DOI in the reference list. Nature Portfolio's protocols.io is a free-to-use and open resource for protocols; protocols deposited onto protocols.io are citable and can be linked from the published article. More details can be found at [protocols.io](https://www.protocols.io/help/publish-articles).

ORCID

Sincerely,
Arunima

Arunima Singh, Ph.D.
Senior Editor
Nature Methods

Version 2:

Decision Letter:

Our ref: NMETH-A58158B

30th Jun 2025

Dear Natalie,

Thank you for submitting your revised manuscript "C-COMPASS: A user-friendly neural network tool profiles cell compartments at protein and lipid levels" (NMETH-A58158B). It has now been seen by the original referees and their comments are below. The reviewers find that the paper has improved in revision, and therefore we'll be happy in principle to publish it in Nature Methods, pending minor revisions to satisfy the referees' final requests and to comply with our editorial and formatting guidelines.

TRANSPARENT PEER REVIEW

ORCID

Sincerely,
Arunima

Arunima Singh, Ph.D.
Senior Editor
Nature Methods

Reviewer #1 (Remarks to the Author):

In the revised version of the manuscript by Kramer and co-workers, the authors did a good job addressing my concerns by benchmarking their tool against other similar tools available. They also assessed the effect of number of fractions and separation resolution for the performance of their tool. As is suspected, a quite big number of fractions is required for proper performance of the C-COMPASS tool. The authors also did a good job on the software part, I checked their Github project page and the tutorials and it seems a good resource. Overall, I think that this revised version has significantly improved. That said, I would appreciate a bit more effort on explaining how to interpret the outcomes of C-COMPASS biologically, but I can imagine that might be out of the scope of the current publication. I also have a few remaining concerns to the revised version:

- In their benchmark against other tools, the authors claims that C-COMPASS shows better performance for predicting dual and multiple locations. However, I failed to understand how the prediction error is calculated in these cases. Should the user provide a list of predefined proteins with multiple locations such that the true positives and false positives can be properly assessed in those cases?

- From this revised version, I can understand how the relocalization score can be used to identify relocation events. Is the magnitude of this value indicative of the magnitude of the event? Does the program provide an accompanying statistical value for this? Moreover, does the program indicate between which compartments the protein relocation is happening?

- In the lines 248 to 253 the authors claim: "Furthermore, a reorganization of the Pi3K pathway, known to be activated by LPS32, was observed. In concordance with the original study using HyperLOPIT, which utilized a T-Augmented Gaussian Mixture Model with Bayesian computation performed using a Markov Chain Monte Carlo (TAGM-MCMC) model for protein localization and relocalization, C-COMPASS detected a high number of translocation events, especially from the lysosome to the plasma membrane and from the cytosol to the nucleus (Fig. 3G), confirming its ability to detect known and validated protein relocalizations under specific treatment conditions." How does the detecting of more relocalization events confirms that their software can detect known and validated protein relocalization events? The authors should provide examples to prove this claim.

- In lines 499 to 501 the authors have rephrased their initial text to "Here, we show that C-COMPASS can extend beyond proteomics and co-integrate lipidomics to co-map protein and lipid compositions of organelles. In the future, the pipeline could also be used for integration of additional omics layers such as metabolomics or transcriptomics" This statement implies that C-COMPASS can be fully compatible with other omics layers. I can understand that from a metabolomics perspective a similar data matrix as in proteomics or lipidomics can be retrieved from subcellular fractionation experiments. However, is that also the case for transcriptomics? Does it make biological sense to perform subcellular fractionation at that level?

Reviewer #2 (Remarks to the Author):

Dear editor, dear authors,

I am impressed by the dramatic improvement of the manuscript. All my previous comments regarding manuscript clarity have been thoroughly addressed as well as my additional minor comments. I am most impressed by the improved quality of the software, which I previously could not even install and now could fully run from start to end without errors. I can attest of the user-friendliness of the software and acknowledge the efforts taken to improve the long-term maintainability of the code. Below are still a few minor comments that should be quick to fix.

L114-116: "To assess this in an unbiased manner, we recommend performing supervised hierarchical clustering followed by GO enrichment analysis prior to applying C-COMPASS." I find this sentence too short to be informative and therefore superfluous.

L703-706: this section seems obsolete or out of place.

L711-714: I do not quite agree with the answer regarding data imputation. In MS experiments, a protein is either detected or not detected in a sample. Whether the protein is not detected because it is truly absent in the sample or because of technical reasons is difficult to assess. Replacing all missing values by zero assumes the absence of technical missingness, which I highly doubt happens even in a fractionation context. However, as missing data management is outside the scope of this work and I do not foresee a dramatic impact on this manuscript's conclusions, I put this comment for the track record and do not require action by the authors.

L832-833: inconsistency in notation either it is $RL^{(s)}_c$ or $RL^{(g_m, g_n)}_c$

L841-842: inconsistency in notation either it is $\bar{w}^{(g)}_c$ or $\bar{w}^{(s,g)}_c$

L854: spooled or pooled?

L871 & L874: inconsistency in notation, $CC^{(s, g)}_c$ should be used instead of $CC^{(c, g)}_s$. Same goes for SA.

Note that I may have missed additional inconsistencies.

L980-984: could you add a brief sentence in the methods that describes how you identify false positives, false negative and true positive for experimental data? While I am satisfied with your answer to my previous comment, I feel this could be a bit more explicit for the reader as well.

In conclusion, I find the manuscript suitable for publication.

Best regards,
Christophe Vanderaa

Reviewer #2 (Remarks on code availability):

The code has been subjected to major improvement, with extensive documentation. I could reproduce their example with minimal efforts. I therefore foresee C-COMPASS to be a usable resource for the community.

Version 3:

Decision Letter:

21st Sep 2025

Dear Natalie,

I am pleased to inform you that your Article, "C-COMPASS: A user-friendly neural network tool profiles cell compartments at protein and lipid levels", has now been accepted for publication in Nature Methods. The received and accepted dates will be October 8, 2024 and September 21, 2025. This note is intended to let you know what to expect from us over the next month or so, and to let you know where to address any further questions.

Over the next few weeks, your paper will be copyedited to ensure that it conforms to Nature Methods style. Once your paper is typeset, you will receive an email with a link to choose the appropriate publishing options for your paper and our Author Services team will be in touch regarding any additional information that may be required. It is extremely important that you let us know now whether you will be difficult to contact over the next month. If this is the case, we ask that you send us the contact information (email, phone and fax) of someone who will be able to check the proofs and deal with any last-minute problems.

Authors may need to take specific actions to achieve compliance with funder and institutional open access mandates.

If your research is supported by a funder that requires immediate open access (e.g. according to [a Plan S principles](https://www.springernature.com/gp/open-science/plan-s-compliance) or the [NIH public access policy](https://www.springernature.com/gp/open-science/us-federal-agency-compliance)) then you should select the gold OA route, and we will direct you to the compliant route where possible. Because authors warrant under our subscription licensing terms that they haven't committed to licensing any version of their article under a licence inconsistent with the terms of our agreement – including the applicable embargo period – publication under the subscription model isn't suitable for authors whose funders require no embargo.

If you are active on Twitter/X or Bluesky, please e-mail me your and your coauthors' handles so that we may tag you when the paper is published.

Best regards,
Arunima

Arunima Singh, Ph.D.
Senior Editor
Nature Methods

** Visit the Springer Nature Editorial and Publishing website at http://editorial-jobs.springernature.com?utm_source=ejP_NMeth_email&utm_medium=ejP_NMeth_email&utm_campaign=ejp_Nmeth for more information about our career opportunities. If you have any questions please click [here](mailto:editorial.publishing.jobs@springernature.com). **

to the original author(s) and the source, provide a link to the Creative Commons license, and indicate if changes were made. In cases where reviewers are anonymous, credit should be given to 'Anonymous Referee' and the source. The images or other third party material in this Peer Review File are included in the article's Creative Commons license, unless indicated otherwise in a credit line to the material. If material is not included in the article's Creative Commons license and your intended use is not permitted by statutory regulation or exceeds the permitted use, you will need to obtain permission directly from the copyright holder. To view a copy of this license, visit <https://creativecommons.org/licenses/by/4.0/>

Response to the Reviewers:

In response to the reviewers' comments, we have substantially revised our manuscript. The main points are:

- We conducted an extensive *in silico* analysis to assess how the number of fractions influences prediction accuracy and method performance.
- We added a detailed comparison of C-COMPASS with existing techniques, including a comprehensive table summarizing features, similarities, and differences. Additionally, we evaluated the performance of the latest tools using the same *in silico* dataset and analyzed result differences in detail.
- We revised and clarified the results moving some content from results to methods. We rewrote the methods section completely, which now includes mathematical notations for all steps.
- By now, the code has undergone significant improvements in quality, performance and documentation.
- To enhance software development and maintenance, we outlined long-term plans, transferred C-COMPASS to a GitHub organization, and included multiple research groups for ongoing support. Additionally, a scientific software developer reviewed the code and documentation.
- Addressing an earlier omission, we have updated Figure 3 to include all fractions in the HYPER-LOPIT dataset from an external study to demonstrate the broad applicability of C-COMPASS across different fractionation workflows. While the figure has been corrected, the overall results and conclusions remain unchanged.

Our point-by-point responses to the single points raised by the reviewers are provided below.

- Response to Reviewer 1
- Response to Reviewer 2

The teal boxes contain the reviewers' comments, while the yellow boxes contain the changes made in the manuscript. Within the yellow boxes, we have marked the changed part in red. To provide some context, the comments (teal) are surrounded by unchanged text (black).

Reviewer #1:

Remarks to the Author:

The manuscript by Natalie Kramer and coworkers presents C-COMPASS, an open-source software tool that utilizes neural network-based regression models to predict the spatial distribution of proteins and lipids across cellular compartments. The tool can analyze proteomics data obtained from organellar fractionation methods such as Protein Correlation Profiling (PCP) or Localization of Organelle Proteins by Isotope Tagging (LOPIT). The tool, which is provided as a user-friendly graphical interface, requires input data on protein abundances across fractions and a set of predefined or custom organelle marker proteins. It was developed with the aim of overcoming some of the limitations of existing proteomics-based organellar profiling methods, which often struggle with proteins that localize to multiple compartments as well as with the integration of protein and lipid data

We thank the reviewer for the constructive feedback and have outlined our responses and actions to address the raised points.

- Although the authors present an interesting tool to perform differential protein location analysis, they fail to present properly its differences and advantages against previous tools. For instance, in the Discussion, authors claim that "Validated with multiple datasets, C-COMPASS accurately predicts multi organelle localization and models organelle composition under different conditions, outperforming earlier single-classification methods". However, no proper comparison is performed against already published algorithms for protein localization, both for single localization, such as MetaMass (Lund-Johansen, F., de la Rosa Carrillo, D., Mehta, A. et al. MetaMass, a tool for meta-analysis of subcellular proteomics data. *Nat Methods* 13, 837–840 (2016). <https://doi.org/10.1038/nmeth.3967>) or for multiple locations such as the Bayesian mixture model described by Crook and colleagues (Crook OM, Mulvey CM, Kirk PDW, Lilley KS, Gatto L. A Bayesian mixture modelling approach for spatial proteomics. *PLoS Comput Biol*. 2018 Nov 27;14(11):e1006516. doi: 10.1371/journal.pcbi.1006516.) or the BUNDLE algorithm (Crook, O.M., Davies, C.T.R., Breckels, L.M. et al. Inferring differential subcellular localisation in comparative spatial proteomics using BUNDLE. *Nat Commun* 13, 5948 (2022). <https://doi.org/10.1038/s41467-022-33570-9>). Neither, do the authors compare C-COMPASS against the differential localization scoring used in other manuscripts from the field, such as: "Martinez-Val, A., Bekker-Jensen, D.B., Steigerwald, S. et al. Spatial-proteomics reveals phospho-signaling dynamics at subcellular resolution. *Nat Commun* 12, 7113 (2021). <https://doi.org/10.1038/s41467-021-27398-y>; or "Itzhak DN., Tyanova S., Cox J, Borner, GHH (2016) Global, quantitative and dynamic mapping of protein subcellular localization *eLife* 5:e16950. <https://doi.org/10.7554/eLife.16950>; . A much more thorough benchmark needs to be done.

Did the results of the study using C-COMPASS significantly deviate from other published tools? If so, what findings were specific to C-COMPASS and how did the authors ensure that the results of C-COMPASS were correct? And what did C-COMPASS miss that other tools found?

We thank the reviewer for this valuable suggestion and acknowledge that this aspect was previously missing from our manuscript. In the revised version, we have incorporated a summary of computational methodologies previously employed in organelle proteomics analysis, which is now presented in the Introduction. We have also added a paragraph to the Discussion section highlighting the key distinctions between C-COMPASS and other existing tools. To further address this point, we introduced Figure 3 and Table 1, which provide a detailed feature comparison across computational methods. Specifically, Table 1 summarizes the core characteristics and differentiating features of C-COMPASS relative to other tools.

We further focused our evaluation on recently published tools producing comparable output matrices. However, meaningful performance comparisons were complicated by variations in the original studies, including differences in cell models, fractionation strategies, targeted organelles, and the lack of standardized benchmarks for accurate localization in experimentally generated datasets under certain biological conditions. To overcome these limitations and enable an unbiased assessment, we generated a synthetic dataset simulating protein distribution profiles. Using this dataset, we systematically evaluated the performance of C-COMPASS alongside other recent tools producing comparable output matrices and being

applicable for various fractionation workflows. The comparative results are presented in Figure 3K-P.

Changes in the introduction lines 56-66:

Recent computational advancements which include DOM-ABC¹⁰ and SubCellBarCode¹⁵ (both employing Support Vector Machines), Metamass¹⁶ (k-means clustering), TRANSPIRE¹⁷ (stochastic variational Gaussian process classifier), BANDLE¹⁸ (Bayesian framework), and TAGM-MCMC¹⁹ (T-Augmented Gaussian Mixture Bayesian model with Markov-chain Monte Carlo techniques) address this complexity and predict protein localizations and relocalizations from fractionation data. While some of these tools use uncertainty or probabilistic estimations to interfere multiple localizations, none of these tools predicts quantitative protein distribution across compartments, integrates organelle and protein abundance data essential for analyzing organelle composition, nor combines proteomic and lipidomic data layers. Additionally, many tools require advanced coding skills, limiting accessibility for researchers without computational expertise.

Addition of Figure 4 K-P:

Fig. 4. Figure 4. Benchmarking C-COMPASS: Impact of Fraction Number and Comparison with Other Organelle Proteomics Methods. (K) Venn diagram showing the overlap of correctly classified single-localization proteins across three analysis tools. (L) Number of proteins successfully analyzed (colored bars) and those excluded due to method-specific filtering. (M) Prediction metrics for proteins simulated with single localizations across three methods. (N) Box plots of prediction errors across tools, categorized by proteins with one, two, or three simulated localizations. (O) Total prediction error per protein for each method, computed as the sum of absolute errors across compartments, divided by 2. (P) Prediction error for protein relocalization events, between predicted and simulated changes in localization between the conditions. (M-O) Medians are shown with middle lines, boxes indicate interquartile ranges (IQR), whiskers extend to 1.5 times the IQR, and dots represent outliers.

Lines 294-327:

Comparison of C-COMPASS with existing organelle proteomics analysis pipelines

Having validated the performance of C-COMPASS using simulated datasets, we next employed these datasets to compare its capabilities with other computational tools developed for spatial proteomics analysis via predictive models of protein localization (see Extended Table 1 for a detailed feature comparison). We selected BUNDLE and DOM-ABC for direct benchmarking, as both tools produce output formats most comparable to those of C-COMPASS. Each is in theory applicable across various spatial proteomics workflows and provides compartment-specific localization values per protein, although existing methods such as BUNDLE and DOM-ABC provide probability-based compartment classifications, whereas C-COMPASS quantitative estimates of protein distribution across organelles.

To assess localization accuracy, we focused on 1,139 proteins that were exclusively localized to a single compartment (complete data in Supplementary Table 4). Of these, 720 were correctly assigned by all three methods, while 16 were misclassified by each (Fig. 4K). Both C-COMPASS and BUNDLE outperformed DOM-ABC in terms of the number of correctly localized proteins. However, DOM-ABC applied a stricter internal filtering strategy, excluding a larger number of proteins to minimize false positives (Fig. 4L). When evaluated using standard classification metrics such as precision, recall, and F1 score, C-COMPASS achieved the highest performance for single-compartment localizations (Fig. 4M). To further assess tool performance, we examined prediction accuracy across proteins with increasing numbers of simulated localizations (Fig. 4N). While all methods performed comparably for proteins with single localizations, though C-COMPASS exhibited slightly higher variance. C-COMPASS consistently outperformed both BUNDLE and DOM-ABC in cases involving dual or triple localizations, demonstrating lower prediction error across these more complex scenarios (Fig. 4O). Finally, we directly compared relocalization accuracy between C-COMPASS and BUNDLE, excluding DOM-ABC due to its incompatible output format for relocalization analysis. This comparison revealed that C-COMPASS consistently achieved lower relocalization errors, confirming its effectiveness in detecting dynamic changes in protein compartmentalization under varying cellular conditions (Fig. 4P).

In summary, all tools robustly predict single localizations; however, C-COMPASS demonstrates superior performance in capturing complex multi-localization patterns by employing a distinct approach. While existing methods such as BUNDLE and DOM-ABC rely on probability-based compartment classifications, C-COMPASS provides quantitative estimates of protein distribution across organelles for analyzing dynamic and heterogeneous localization profiles.

We added detailed descriptions to the Methods how comparisons were done (L 928-971):

Statistics for comparison of fractions and tools

To evaluate the accuracy of each tool using simulated data, we quantified localization and relocalization errors by comparing the known expected values (defined by the simulation parameters) with the results from C-COMPASS, BUNDLE, and DOM-ABC. Specifically, expected protein localization across compartments was compared to resulting CC values for C-COMPASS, and to localization probabilities for BUNDLE and DOM-ABC.

BUNDLE protein predictions were evaluated with version 1.8.0 in conditions 1 and 2 as described in the Bioconductor documentation (doi: 10.18129/B9.bioc.bundle) with adaptations. To each replicate, the analysis was performed considering 4,000 Markov-chain Monte Carlo (MCMC) iterations, 400 burn-in iterations, and 9 chains. The matrix of Dirichlet priors was defined considering 0.1% of relocalization and the suggested values for penalized complexity priors for big datasets.

C-COMPASS and DOM-ABC results were distributed across the 8 compartments used during simulation. In contrast, BUNDLE included a ninth compartment corresponding to an 'outlier' probability, which reflects the likelihood that a protein localizes to a compartment not included in the marker list. The BUNDLE authors recommend a threshold of 0.99 for both the outlier probability and the compartmental assignment probability to confidently assign a protein to a single compartment¹⁸. Therefore, proteins with both an outlier probability >0.99 and a compartment probability >0.99 were considered fully localized to compartment 9.

Localization error (LE) was calculated as the absolute difference between expected and predicted localization values for each compartment, across all proteins. Similarly, relocalization error (RLE) was calculated using the difference between expected and predicted relocalization values, where relocalization is defined as the localization difference between conditions. Both LE and RLE were divided by two to account for the fact that an overestimation in one compartment necessarily leads to an underestimation in another, thereby avoiding double-counting of errors.

To further assess RLE with respect to specific origin-target compartment transitions, we applied a continuous transport-based approach that conserves predicted abundance across compartments. For each protein, we computed a transport matrix by minimizing the cost of redistributing the origin localization distribution (condition 1) into the target distribution (condition 2), using a modified Earth Mover's Distance (EMD) that ensures conservation of total signal.

This yielded a transport matrix $F = [f_{i,j}]$ per protein, where each entry reflects the amount redistributed from a specific origin compartment i to a specific target compartment j :

$$EMD^{(P,Q)} = \min_F \sum_{i=1}^m \sum_{j=1}^n f_{ij} \times d_{ij},$$

where:

$P = (p_1, \dots, p_m)$ is the origin localization distribution (condition 1),

$Q = (q_1, \dots, q_m)$ is the target localization distribution (condition 2),

d_{ij} is the cost of transporting one unit of mass from compartment i to j , and

f_{ij} is the amount of mass moved from i to j .

$Q = (q_1, \dots, q_m)$ is the target localization distribution (condition 2),

d_{ij} is the cost of transporting one unit of mass from compartment i to j , and

f_{ij} is the amount of mass moved from i to j .

The same procedure was applied to both expected and predicted distributions. Element-wise differences between the expected and resulting transport matrices were used to calculate the per-element transport error. The final global error matrix was obtained by averaging these error matrices across all proteins present in both conditions.

Addition to the Discussion lines 521-541:

C-COMPASS offers a complementary and distinct approach to existing tools for spatial proteomics analysis. Most available methods rely on probabilistic machine learning techniques, such as support vector machines, K-means clustering, or Bayesian inference, and are primarily designed for classification tasks, assigning proteins to a single compartment based on likelihood or confidence scores. In contrast, C-COMPASS employs a neural network framework for quantitative, multi-class, multi-label regression, enabling it to estimate protein distribution across multiple compartments and capture multi-localization patterns more explicitly. While several existing tools support comparative analysis between biological conditions, these are generally limited to pairwise comparisons relative to a control and tend to be qualitative in nature. C-COMPASS extends this functionality by supporting quantitative comparisons across multiple conditions, facilitating the detection and measurement of relocalization events. Additionally, it is currently the only tool among those compared that provides quantitative predictions and not probabilistic predictions for proteins localized to more than one compartment, which can be particularly useful in studies involving complex or dynamic spatial reorganization. From a usability perspective, C-COMPASS is designed to be accessible to a broad range of users. Unlike tools that require command-line proficiency, R-based environments, or manual data pre-processing, it includes a graphical user interface and is available as a PyPI package or standalone executable, simplifying installation and use. Overall, C-COMPASS provides a flexible solution for spatial proteomics analysis, particularly in contexts where quantitative multi-localization and multi-condition comparisons are of interest.

- In line 191, and again in line 241, the authors mentioned a reduction in the number of fractions used for organelle fractionation. They give some hints that such reduction has an effect on increase organelle overlap, but this effect is not properly assessed. What would be the smallest number of fractions required for an accurate performance of this strategy? Also, in this regard, would this strategy work for other subcellular proteomics strategies, which do not rely on organelle purification but rather on sequential enrichment?

The reviewer raises an interesting and relevant question. To provide general guidance, we systematically explored this question using a synthetic dataset, varying both the number of compartments and fractions. We evaluated method performance by analyzing F1 scores and true positive predictions concerning these variables (Figure 4A-J). This analysis provides insights into the influence of fraction number and gives some general guidance for experimental design.

Impact of fractionation resolution on prediction accuracy

The experimental design for C-COMPASS analyses requires balancing resolution with practical limitations, including mass spectrometry (MS) measurement time and sample preparation workload. To systematically assess how the number of fractions affects performance, we created simulated datasets that closely reflect gradient centrifugation experiments (see Methods). By varying both fraction counts and compartment numbers, we evaluated localization predictions for single- and multi-compartment proteins to determine how resolution impacts accuracy (Fig. 4A). Unlike experimental datasets, where ground truth localizations are typically unavailable, these simulations allow direct evaluation of predicted versus true compartment assignments.

Prediction accuracy consistently improved with an increasing number of fractions, as indicated by a reduction in mean prediction error (Fig. 4B). Correspondingly, the proportion of proteins with high prediction accuracy, defined by a localization error (LE) below 0.1, increased across various simulated compartment numbers (Fig. 4C). However, when the ratio of fractions to compartments was low, approximately 6% of proteins remained unassigned due to C-COMPASS's built-in filtering mechanism, which removes low-confidence predictions resulting from insufficient resolution (Fig. 4D). An assessment of F1 scores further confirmed that the number of fractions directly determines compartment resolution. While F1 values remained above 0.9 for up to 12 compartments with 16 fractions, the highest scores were achieved when the number of fractions was at least twice the number of compartments, highlighting the need of balancing these parameters in experimental design. We further evaluated C-COMPASS's performance in predicting complex localization patterns, by analyzing the simulated dataset comprising 8 compartments and 16 fractions, where proteins were assigned to multiple compartments with localization contributions mixed in 0.25 increments. The resulting predicted class contributions (CC values) closely matched the expected distributions (Fig. 4F), demonstrating the ability of C-COMPASS to accurately quantify multi-localizations. To identify sources of misclassification, we computed mean prediction values for proteins assigned to a single compartment (Fig. 4G), revealing that inaccuracies were predominantly associated with overlapping fractionation profiles (Fig. 4H), whereas well-separated compartments showed minimal false positives. We further assessed C-COMPASS's performance in detecting and quantifying protein relocalization based on simulated data, which showed strong concordance between predicted and true localization changes. The compartment-specific relocalization error (RLE) remained low, with a maximum of 0.048 (Fig. 4I). Notably, the highest errors occurred between compartments with overlapping profiles (Fig. 4J), while relocalizations between distinct compartments were more accurately resolved.

Taken together, these results indicate that limitations in both localization and relocalization accuracy primarily arise from overlapping compartment profiles. This issue can be mitigated by increasing the resolution of the fractionation. For optimal C-COMPASS performance, we recommend using at least twice as many fractions as the number of compartments to be resolved.

Fig. 4. Evaluation of C-COMPASS performance across fraction numbers and Comparison with other Organelle Proteomics Tools. (A) Simulated marker profiles shown across different numbers of fractions. Profiles are shown for simulations with minimum and maximum numbers of compartments and fractions. (B) Line plot of mean prediction error and (C) line plot of the number of proteins with a prediction error below 0.1 across increasing for numbers of simulated fractions. Errors were calculated as the sum of absolute differences across compartments, divided by 2. (D) Scatter plot showing the percentage of proteins that remained unassigned by C-COMPASS, plotted against the ratio of simulated fractions to compartments. (E) F1 scores for simulations containing 16 fractions and varying numbers of compartments. (F) Predicted CC values for proteins with different simulated localization amounts. Blue circles indicate the mean, black lines the median, and grey lines the interquartile range. (G) Matrix showing average predicted CC values per compartment (y-axis) for proteins simulated to localize to a single compartment (x-axis). Empty cells represent values of 0. H) Normalized intensity profiles of simulated marker proteins for the first condition of the dataset used in G. (I) Heatmap of relocalization errors. Matrix shows origin compartments (y-axis) and target compartments (x-axis) for relocalizing and non-relocalizing proteins. (J) Normalized intensity profiles of marker proteins for the second condition of the dataset used in G and I.

- If understood correctly, the algorithm in C-COMPASS relies on a prior list of expected organelles, with known or predefined markers. Does this mean that C-COMPASS cannot be applied to organelles, which are not defined among the ones used in this publication? For instance, can C-COMPASS analyze data of non-membrane protein condensates or stress granules?

The resolution of specific compartments depends upon two main factors: 1) the fractionation procedure and its ability to distinguish between compartments, and 2) the availability and incorporation of markers into the analysis. We have addressed these points clearly in the text lines 95-116:

C-COMPASS uses experimentally generated protein abundance profiles and is independent of LC-MS/MS acquisition methods, whether data-dependent acquisition (DDA) or data-independent acquisition (DIA), as well as labeling approaches such as label-free, Tandem Mass Tag (TMT), or Stable Isotope Labeling by Amino acids in Cell culture (SILAC). To annotate protein localizations from cellular fractionation data, C-COMPASS requires: (1) a pivot table of identified proteins and their intensities across fractions, and (2) a marker set of proteins typically localized to single compartments.

For this, we provide predefined marker lists, derived from experimentally validated datasets from resources such as microscopy or proteomics studies, including the Protein Atlas²⁰ and a subcellular map in HeLa cells²¹. These sources are more reliable and preferred over GO annotations, which often have overlapping and therefore limited informational value. Our provided list encompasses all major membrane-enclosed organelles and includes two additional categories: protein complexes and protein synthesis. Protein complexes, such as chaperone complexes or MTOR, sediment at higher densities than soluble proteins, allowing techniques like protein correlation profiling²² and differential centrifugation²¹ to resolve them. Protein synthesis, which can be categorized as a membrane-less organelle, comprises ribosomes, mRNA-binding proteins, and other components involved in translation. Alternatively, users can import custom lists, which can include both membranous and membrane-less structures. Resolution depends on two factors: 1) the inclusion of markers in the list, and 2) the ability of the fractionation procedure to separate compartments. To assess this in an unbiased manner, we recommend performing supervised hierarchical clustering followed by Gene Ontology (GO) enrichment analysis prior to applying C-COMPASS.

- In line 123, the authors mention the necessity to use total proteome information. Is this strictly required? If so, it needs to be clarified that for using this tool, total proteome is required to be measured in parallel to organelle fractions. Also, in this regard; should the total proteome have the same depth as the organelle fractionation proteome? Meaning that all proteins profiled in the organelle fractions need to be identified and quantified in the total proteome as well, to be able to normalize them?

The incorporation of the total proteome data is optional, and an increased depth is beneficial. We have clarified this in the text lines 139-146:

To further investigate how compartments adjust their composition with biological changes, C-COMPASS provides the option to integrate total proteome data. By normalizing CC values with corresponding protein levels and organelle abundance, estimated from summed intensities of organelle markers, the composition of each compartment and its changes with biological changes can be modeled, the third C-COMPASS output, offering a class-centric perspective on protein level changes within compartments (Extended Fig. 1D). An increased depth of the proteome is beneficial here, as it increases the number of proteins with comprehensive data at both the proteome and localization levels.

- It is not clear, how the statistical significance of the protein relocations is calculated neither from the results nor from the methods sections? This needs to be described in much more detail. Moreover, it is not clear if multiple comparison corrections are performed?

We have clarified this question in the methods section lines 835-860:

To assess the statistical significance of relocalization events, we performed Welch's t-tests based on the neural network outputs $w_c^{(s,r)}$, which were obtained from multiple rounds and retraining runs. For each condition g and compartment c , this yielded a distribution of predicted scores:

$$w_c^{(s,g)} = \{w_{c,1}^{(s,g)}, w_{c,2}^{(s,g)}, \dots, w_{c,R}^{(s,g)}\}.$$

The Welch's t-statistic for each compartment was computed as:

$$t = \frac{\bar{w}_c^{(s,g_n)} - \bar{w}_c^{(s,g_m)}}{\sqrt{\frac{\sigma_c^{2(s,g_n)}}{R_n} + \frac{\sigma_c^{2(s,g_m)}}{R_m}}},$$

Where $\bar{w}_c^{(g)}$ is the mean predicted score, $\sigma_c^{2(g)}$ is the sample variance, and R_m, R_n are the number of runs for g_m and g_n , respectively.

The degrees of freedom ν were estimated using the Welch-Satterthwaite equation:

$$\nu_c^{(s)} = \frac{\left(\frac{\sigma_c^{2(s,g_m)}}{R_m} + \frac{\sigma_c^{2(s,g_n)}}{R_n}\right)^2}{\frac{\left(\frac{\sigma_c^{2(s,g_m)}}{R_m}\right)^2}{R_m-1} + \frac{\left(\frac{\sigma_c^{2(s,g_n)}}{R_n}\right)^2}{R_n-1}}.$$

The resulting p-values were then calculated from the two-sided t-distribution with ν degrees of freedom, as implemented in the `scipy.stats.ttest_ind` function (with `equal_var=False`) from the SciPy Python package:

$$p_value = 2 \times (1 - CDF_{t_\nu}(|t|)),$$

where CDF_{t_ν} denotes the cumulative distribution function of the t-distribution with ν degrees of freedom.

In addition to statistical significance, we also calculated Cohen's d as a standardized effect size metric for each compartment c :

$$d_c^{(s)} = \frac{\bar{w}_c^{(s,g_n)} - \bar{w}_c^{(s,g_m)}}{\sigma_{spooled}}, \text{ where } \sigma_{spooled} = \sqrt{\frac{\sigma_c^{(s,g_m)} + \sigma_c^{(s,g_n)}}{2}}.$$

This metric accounts for both the magnitude of relocalization and the precision of the model predictions, providing a unified score to identify promising relocalization candidates.

To quantify the overall extent of relocalization between two conditions, we computed the ReLocalization Score (RLS) as the sum of the absolute relocalization values across all compartments:

$$RLS^{(s)} = \sum_{c=1}^C |RL_c^{(s)}|.$$

MINOR POINT:

- In figure 5B: which subcellular compartment does “Protein Complex” refer to?

We have added a detailed explanation lines 102-116:

For this, we provide predefined marker lists, derived from experimentally validated datasets from resources such as microscopy or proteomics studies, including the Protein Atlas²⁰ and a subcellular map in HeLa cells²¹. These sources are more reliable and preferred over GO annotations, which often have overlapping and therefore limited informational value. Our provided list encompasses all major membrane-enclosed organelles and includes two additional categories: protein complexes and protein synthesis. Protein complexes, such as chaperone complexes or MTOR, sediment at higher densities than soluble proteins, allowing techniques like protein correlation profiling²² and differential centrifugation²¹ to resolve them. Protein synthesis, which can be categorized as a membrane-less organelle, comprises ribosomes, mRNA-binding proteins, and other components involved in translation. Alternatively, users can import custom lists, which can include both membranous and membrane-less structures. Resolution depends on two factors: 1) the inclusion of markers in the list, and 2) the ability of the fractionation procedure to separate compartments. To assess this in an unbiased manner, we recommend performing supervised hierarchical clustering followed by Gene Ontology (GO) enrichment analysis prior to applying C-COMPASS.

- In the discussion, the authors mention: “Importantly, C-COMPASS extends beyond proteomics to include other omics levels such as lipidomics, transcriptomics, and metabolomics”. However, the publication only presents integration of proteomics and lipidomics. The authors should either demonstrate and benchmark the potential integration with transcriptomics, and metabolomics, or remove the sentence.

We have rephrased the sentence lines 499-501:

Here, we show that C-COMPASS can extend beyond proteomics and co-integrate lipidomics to co-map protein and lipid compositions of organelles. In the future, the pipeline could also be used for integration of additional omics layers such as metabolomics or transcriptomics.

Reviewer #2:

Dear Editors, dear authors,

I congratulate the authors for this interesting piece of research. They provided a novel data analysis tool in the field of spatial proteomics by exploiting neural networks for assigning protein intensity profiles to cellular compartments. Their classification method further allows multiple localization, which apparently is not yet possible with the current state-of-the-art despite its biological relevance. Moreover, they leverage the output of their models to further assess changes in these compartment distributions upon treatment or other experimental condition and attempt to assess organelle composition. To ensure reliability of their data, they carry out a thorough model training and validation procedure and prevent over-fitting. They further show that their method is not limited to proteomics analysis, but can also be applied to other omics modalities, here illustrated with a combined proteomics and lipidomics data set.

We thank the reviewer for pinpointing the novelty of the work and the constructive feedback. We tried to address the three major concerns in the revised manuscript and hope that these changes improve the clarity of the manuscript. We are happy to address further comments if questions remain open.

Major concern 1: clarity of the manuscript

I found several parts of the manuscript very unclear, leading me to guess what the authors mean or to ignore some parts of the manuscript. I'm convinced that investing some time improving clarity will dramatically increase the quality of the manuscript. Here is a comprehensive list of the parts that need a major revision.

- General comment: the Results section provides too many details that could be in the methods section, whereas the Methods section is too short, very difficult to read and lacks mathematical notation that would clarify what the authors attempt to compute.

In response to the reviewer's criticism, we have revised the entire methods section for improved clarity and relocated parts of the results section to the methods where appropriate.

- L142-145: quality assessment should also include the SVM results from the initial study. Currently, it is difficult to assess whether C-COMPASS performs better than SVM.

In Figure 2, we compare the C-COMPASS output with the results from the previous SVM analysis. We apologize if this was not sufficiently clear in the earlier version of our manuscript and have rephrased the relevant text for improved clarity. By comparing the number of markers correctly predicted in both the previous SVM analysis and the C-COMPASS analysis, we demonstrate the superior performance of C-COMPASS. Moreover, we have added a supplemental figure, Extended Figure S2C, which compares the F1 values of C-COMPASS and the SVM output from the original study. Furthermore, we have now included Figure 4K-P, which provides a detailed comparison of C-COMPASS with other available tools for organelle proteomic data analysis, including DOM-ABC, a tool which is based on SVMs. Lines 181-198:

To assess the reliability of C-COMPASS predictions compared to SVM-based predictions from Klingelhuber et al.²³, we evaluated the number of correctly predicted marker proteins for each method and validated uniquely assigned proteins using databases such as MitoCarta²⁵ and LD Knowledge Portal (LDKP)²⁶. C-COMPASS and SVM-based predictions showed a 53.7% overlap in mitochondrial classifications. C-COMPASS demonstrated a greater overlap with marker proteins (97.1%) compared to SVM predictions (83.9%), indicating a higher reliability than the traditional machine learning-based classification. Additionally, 26 of the 120 proteins uniquely assigned to mitochondria by C-COMPASS but absent from the marker set were validated by MitoCarta, suggesting they are likely true mitochondrial proteins (Fig. 2H). In contrast, only 17 of the 229 proteins uniquely assigned by SVMs were found in MitoCarta. For LDs, C-COMPASS predicted 320 proteins, with 38 confirmed (8 by marker proteins, 30 by LDKP) (Fig,2H), thereby indicating strongly enhanced LD protein identification compared to SVMs. Overall, C-COMPASS outperformed SVMs in correctly assigning marker proteins to compartments in 11 out of 12 cases (Extended Fig. 2B), indicating strongly enhanced performance and reliability of the predictions compared to the SVMs. The overall F1 score of C-COMPASS (0.84) was higher than for the original SVM-based predictions (0.80) (Extended Fig. 2C).

Extended Figure 2. Comparison of C-COMPASS performance versus SVM predictions. (C) Bar plot shows the F1 scores for the original predictions and C-COMPASS predictions based on highest CC values.

- L163: the authors state that 54% of the hits are confirmed by MitoCarta and this indicates improved accuracy? This reads to me as if 46% of the markers are false hits. Moreover, how does MitoCarta markers overlap with SVM results?

We apologize for any confusion caused by our previous explanation. We have changed the visualization in Figure 2 and the corresponding text to make our message clearer.

Change in Figure 2.

Change to text lines 183-194:

...Klingelhuber et al.²³, we evaluated the number of correctly predicted marker proteins for each method and validated uniquely assigned proteins using databases such as MitoCarta²⁵ and LD Knowledge Portal (LDKP)²⁶. C-COMPASS and SVM-based predictions showed a 53.7% overlap in mitochondrial classifications. C-COMPASS demonstrated a greater overlap with marker proteins (97.1%) compared to SVM predictions (83.9%), indicating a higher reliability than the traditional machine learning-based classification. Additionally, 26 of the 120 proteins uniquely assigned to mitochondria by C-COMPASS but absent from the marker set were validated by MitoCarta, suggesting they are likely true mitochondrial proteins (Fig. 2H). In contrast, only 17 of the 229 proteins uniquely assigned by SVMs were found in MitoCarta. For LDs, C-COMPASS predicted 320 proteins, with 38 confirmed (8 by marker proteins, 30 by LDKP) (Fig.2H), thereby indicating strongly enhanced LD protein identification compared to SVMs.

- L195: "...indicating high reliability in compartment predictions (Fig. 4c)". I agree, but what about the results by Mulvey et al.? Again, it is difficult to assess whether C-COMPASS performs better than the original study.

In Figure 3C (previously Figure 4C), we illustrate that C-COMPASS can handle data from various organelle purification techniques, achieving accuracy comparable to those from the original study. This point has been made clear in the revised text. Additionally, we emphasize that while the accuracy remains consistent, a significant advantage of C-COMPASS is its ability to predict multiple localizations for many proteins which were not captured in the original study. For instance, among proteins classified as "unknown" in the original study, C-COMPASS was able to map multiple localizations for over 80% of them (Figure 3E). Although we do not have a direct method to verify the accuracy of these multi-localization predictions, they are biologically plausible, since it is known that the proteins often co-localize between for example lysosomes and plasma membrane or nucleus and DNA-binding (Figure 3D). Moreover, C-COMPASS not only captures a substantial overlap in the relocalization events identified in the original study but also enhances the analysis by providing quantitative measures for these relocalizations, thus offering a methodological improvement.

Addition to text lines 226-239:

A comparison matrix between Mulvey et al.'s²⁹ original predictions and those from C-COMPASS showed a strong correlation (Figure 3E and Extended Figure 3F.), demonstrating a high level of agreement in prediction performance and confirming C-COMPASS's capability for accurate predictions across different fractionation methods beyond PCP. Notably, C-COMPASS outperformed the original methodology by identifying multiple protein localizations that the original study did not detect. For instance, proteins initially categorized solely as lysosomal were found to also localize to the plasma membrane, and those labeled as nuclear were additionally found in the ribosome, chromatin, and cytosol. The identification of biologically expected multiple localizations by C-COMPASS suggests that it accurately captures genuine multi-compartmental distributions of proteins. Moreover, 81% of proteins categorized as unknown in the original study were found by C-COMPASS to occupy multiple compartments, demonstrating its capability to overcome the challenges associated with assigning definitive localizations to these proteins through its ability to predict multilocalization patterns.

- Fig2, L280-282, L654-660: exploring the organelle composition seems very interesting, but I did not understand it. The authors should clarify the biological question they attempt to answer and how their approach tackles the question. After reading several times the part in the Methods section, I am still unable to grasp the procedure. The Methods part should be completely rewritten, and I suggest to make use of mathematical notation to avoid ambiguity. Finally, exploiting the organelle composition results with a simple cluster analysis did not convince me of its relevance.

We have added a mathematical description of how the organelle composition was calculated to the methods section lines 861-882:

Organelle composition

To analyze organelle composition across conditions, we first computed compartment-targeted protein intensities by combining predicted $CC_c^{(s,g)}$ with total proteome intensity values. For each species s and condition g , the total proteome intensity was calculated as the mean across all replicates R :

$$\bar{I}_s^{(g)} = \frac{1}{R} \sum_{r=1}^R I_r^{(s,g)},$$

Where $I_r^{(s,g)}$ is the total proteome intensity for species s from condition g in replicate r , and $\bar{I}_s^{(g)}$ is the mean intensity across all replicates R for species s from condition g .

The compartment-weighted species abundance $SA_c^{(s,g)}$ for each species s , condition g , and compartment c was then calculated as:

$$SA_s^{(c,g)} = CC_s^{(c,g)} \times \bar{I}_s^{(g)}.$$

To estimate the total abundance of each compartment in a given condition, these values were summed across all species s per compartment c :

$$CA_c^{(g)} = \sum_{s=1}^S SA_s^{(c,g)},$$

where the compartment abundance $CA_c^{(g)}$ reflects the relative representation of compartment c in condition g , based on the total contribution of all proteins assigned to it.

Each compartment-weighted species abundance $SA_s^{(c,g)}$ is then normalized by the compartment abundance $CA_c^{(g)}$ giving the normalized abundance $NA_s^{(c,g)}$:

$$NA_s^{(c,g)} = \frac{SA_s^{(c,g)}}{CA_c^{(g)}}.$$

Finally, to compute changes in the levels of species on a specific compartment, we calculated the class-centric fold change ($CFC_s^{(c,g)}$) using log₂-transformed normalized values:

$$CFC_s^{(c,g)} = \log_2(NA_s^{(c,g_n)}) - \log_2(NA_s^{(c,g_m)}).$$

We have further clarified the text section lines 329-334:

Organelle composition is determined by three main factors: I) protein localization, II) protein abundance, and III) organelle abundance. C-COMPASS integrates these parameters to assess changes in organelle-specific protein abundance levels by combining localization changes and protein expression levels derived from the total proteome. By combining these two levels of information, we could estimate changes in the organelle abundance to normalize the results.

- L634-639: I can guess what the authors mean after several reads, but the text should be clarified, especially in the Methods sections.

We have rephrased the Methods section and added mathematical formulations:

Output filtering and final network output processing

To further reduce spurious (false-positive) assignments, we apply a compartment-specific threshold. For each compartment c , we gather the outputs that appear in marker proteins of other compartments. The 95th percentile of these “false-positive” outputs defines a threshold τ_c . If a protein’s raw output $y_c^{(raw)}$ is below τ_c , we set it to 0. The remaining nonzero outputs are then rescaled so that they sum to 1:

$$w_c^{(filtered)} = \begin{cases} w_c^{(raw)}, & \text{if } w_c^{(raw)} \geq \tau_c, \text{ and } \sum_{c=1}^C w_c^{(filtered)} = 1. \\ 0, & \text{otherwise.} \end{cases}$$

This yields the final ClassContribution (CC) values. Any protein retaining exactly one nonzero compartment after this filtering is labeled as single localization in that compartment. We note that in principle, this filtering step can also be used already during network training.

- L648: student t-test is used, but given the restricted distribution range of RLs (0-2), I expect the data to be heteroskedastic. I would rather use a Welch t-test, or maybe even a non-parametric test if the authors cannot prove that the sample means are normally distributed.

We apologize for the error in the methods description. We indeed used a Welch t-test and have corrected this in the methods section.

- L641-645: it's not clear what data is used to compute precision, recall and F1, namely how the target values are defined. Also, the main text mentions false positives (L152), but how are false positives defined? I guess that the protein marker list is used as ground truth data, but could it be the that the 2 FP are new markers that were not or wrongly described before? Also, scikit-learn weighted averaging is used, but how were the weights assigned?

We have updated the Methods section to include a detailed explanation of how the values were computed, including the use of Scikit-learn's weighted averaging approach. We agree with the reviewer that using protein markers as ground truth has inherent limitations, false positives may result from proteins that were previously misannotated or that exhibit context-dependent localization. To address this, and to enable a more controlled comparison between existing tools, we have now opted to use a synthetically generated dataset in which the true localizations are precisely defined.

Addition to Methods lines 972-990:

Quality assessment

To evaluate the performance of compartment predictions, we calculated standard classification quality metrics, including precision, recall, and F1 score, using the scikit-learn Python package. Martini protein annotations served as the ground truth, and predicted compartments were determined by selecting the compartment with the highest predicted $CC_c^{(s)}$ value for each species s . An exception was made for the evaluation of the upsampling strategy, where raw neural network outputs $w_c^{(raw)}$ were used instead. For each compartment c , the quality metrics were computed as follows:

$$Precision_c = \frac{TP_c}{TP_c + FP_c}$$

$$Recall_c = \frac{TP_c}{TP_c + FN_c}$$

$$F1_c = 2 \times \frac{Precision_c \times Recall_c}{Precision_c + Recall_c}$$

where TP_c is the number of true positives, FP_c is the number of false positives, and FN_c is the number of false negatives per compartment c .

To assess overall performance across all compartments, F1 scores were weighted by the number of marker proteins evaluated in each compartment:

$$F1_{weighted} = \frac{\sum_{c=1}^C n_c F1_c}{\sum_{c=1}^C n_c},$$

where n_c is the number of marker proteins used for evaluation in compartment c .

This approach ensures that compartments with more ground-truth annotations contribute proportionally to the final quality scores.

Major concern 2: software engineering

Since the title mentions that it is "a user-friendly tool" and that publication focuses on the development of a novel computational approach, I'll be picky on the software engineering. In its **current** state, I foresee the software to have a limited lifetime and a limited user base, but I am convinced the authors can improve this.

We appreciate the reviewer's concerns regarding software quality and maintenance and their conviction that we can improve on that matter. By now, the code has undergone significant improvements in both quality and performance. This includes, for example:

- Automated code formatting
- Complete refactoring to improve readability and modularity which made the codebase more maintainable
- Compatibility with recent versions of Python and other dependencies as well as different operating systems
- Packaging as a Python package that is now automatically uploaded to PyPI and Zenodo upon new releases
- The documentation is now available at ReadTheDocs, and both the in-source and out-of-source documentation have been extended
- Improved error handling
- Various usability improvements ranging from small GUI tweaks to parallel of several analysis steps

- The code is provided through a personal GitHub account, with a single contributor and showing little commit activity and no community activity (no issues, no pull request). I doubt the code has been reviewed. I don't mean to offend the main author which has invested a lot of effort in writing the software, but rather highlights that the software is prone to being abandoned upon project's completion. Could the author comment on their maintenance and lifecycle plan for management of the software?

We moved the repository to the GitHub organization of the Hasenauer lab (<https://github.com/ICB-DCM/C-COMPASS/>). The code has meanwhile undergone full review. Code quality has improved, and there was major refactoring to improve maintainability. So far, there have been neither issues nor pull requests by external users, because C-COMPASS isn't well known yet. We expect this to change after the publication of this manuscript.

- There is no unit testing, this again questions the long term maintainability of the code.

The code has been refactored for better testability and a first set of unit tests has now been added. The unit tests along with integration tests and checks for code quality are now automatically executed via GitHub actions.

- The software executables are only provided for Windows, forcing users with MacOS, Linux or other OS to use the tools through the command line, which may have a significant impact on the user base. Since I use Linux, I have to install it manually.

The code has been made compatible with recent versions of Windows, Linux, and macOS. Single-file executables are now available for these three platforms in addition to the now possible installation from PyPI.

- Manual installation of the module lacks documentation and a streamlined workflow. For instance, I tried installing the dependencies using the CCMPS/requirement.txt, but I hit an awful lot of version clashes which led me to abandon C-COMPASS installation...

The code has now been properly packaged and adapted to recent versions of Python and its other dependencies. The package is now available on PyPI (<https://pypi.org/project/ccompass/>) and pip-installable (`pip install ccompass`). Complementarily, we provide single-file executables that do not need a prior Python installation. The different installation options have been documented.

Note that I saw the authors provided a Zenodo repository with an example session and data, but it is still private.

We have released the Zenodo repository.

Major concern 3: reproducibility

From the journal's peer review guidelines, "Is the reporting of data and methodology sufficiently detailed and transparent to enable reproducing the results?" I currently have to answer no as the Methods section is unclear and there is no script/guidelines to reproduce any of the results presented in this work. Since I couldn't run the tool, I could not assess whether these results were all generated from within C-COMPASS itself.

We understand the concern regarding reproducibility and have taken several steps to address it. The Methods section has been restructured and expanded to provide a clearer and more detailed description of the data processing steps. Mathematical calculations have been formalized and are now presented as equations, and redundant methodological content in the Results section has been removed and integrated into the Methods for clarity. We have also updated the tool to ensure compatibility with Linux systems. Furthermore, the example dataset and the C-COMPASS sessions have been updated and are now available on Zenodo. These improvements should allow to run the tool and reproduce results as presented.

Minor concerns:

- L92: could the authors provide some suggestions of good sources or data bases for marker lists? How reliable do they need to be?

We have included an explanation highlighting that GO-annotations, while commonly used, often have limited value due to their strong overlap and reliability issues. Instead, we favor using experimentally validated datasets that are either microscopy-based, such as the Protein Atlas, or proteomics-based, which provide more accurate and dependable insights (lines 102-116).

For this, we provide predefined marker lists, derived from experimentally validated datasets from resources such as microscopy or proteomics studies, including the Protein Atlas²⁰ and a subcellular map in HeLa cells²¹. These sources are more reliable and preferred over GO annotations, which often have overlapping and therefore limited informational value. Our provided list encompasses all major membrane-enclosed organelles and includes two additional categories: protein complexes and protein synthesis. Protein complexes, such as chaperone complexes or MTOR, sediment at higher densities than soluble proteins, allowing techniques like protein correlation profiling²² and differential centrifugation²¹ to resolve them. Protein synthesis, which can be categorized as a membrane-less organelle, comprises ribosomes, mRNA-binding proteins, and other components involved in translation. Alternatively, users can import custom lists, which can include both membranous and membrane-less structures. Resolution depends on two factors: 1) the inclusion of markers in the list, and 2) the ability of the fractionation procedure to separate compartments. To assess this in an unbiased manner, we recommend performing supervised hierarchical clustering followed by Gene Ontology (GO) enrichment analysis prior to applying C-COMPASS.

- L110-112: I had to read the sentences several times to understand the underlying concept. I suggest rewriting it for clarity.

We have rewritten the whole paragraph and this sentence has been removed.

- L113: "renormalized to sum to one": is it renormalized per compartment across proteins or per protein across compartments? I guess it's the latter, but it should be clarified.

The normalization was per protein across compartments. We have clarified this in line 715.

To ensure comparability, the profiles were normalized so that the total sum across fractions f equals 1:

- L117: the ensemble approach comes out of the blue in the main text. After reading the methods section, I find ensembling an interesting approach to stabilize the predictions, so it's worth clarifying this in the results.

We have now introduced the ensemble strategy in the results section lines 129-138 and added a more detailed description in the methods section.

To reduce false positives, values below a compartment-specific threshold are set to zero, and the remaining values are re-normalized to sum to one, producing class contributions (CC) that reflect protein distribution across compartments. To stabilize predictions, the process is repeated multiple times, generating ensemble averages. These condition-specific CC values form the first C-COMPASS output (Extended Fig. 1D). For comparative analysis, changes in CC between conditions, called relocalization (RL), range from -1 to 1. Using variation across ensemble runs, p-values and Cohen's D are calculated per compartment to derive two metrics: the relocalization score (RLS), summarizing total change (0–2), and the distance score (DS), ranking shifts by effect size. These global relocalization metrics make up the second C-COMPASS output (Extended Fig. 1D).

Addition to methods section lines 797-811:

Neural network optimization

The model is trained to minimize the mean squared error (MSE) between the network's predictions and known (or upsampled) reference assignments. For each replicate, the upsampled data is split into 80 % for training and 20 % for validation. Training stops early if the validation MSE does not improve after 5 consecutive epochs (*patience* = 5) to avoid overfitting.

We employ Keras Hyperband to select the best performing optimizer (Stochastic Gradient Descent, Root Mean Square Propagation, or Adaptive Moment Estimation), the learning rate (logarithmically distributed within $[10^{-4}, 10^{-1}]$) and the size of the 1st hidden layer. Each configuration is tested for up to 20 epochs (with a reduction factor of 3), and the best settings are selected. After determining the best hyperparameters, we retrain the model on the complete training dataset (including synthetic profiles) for 10 independent runs. This entire procedure (upsampling, mixing, training, hyperparameter tuning, and training and prediction) is repeated in three rounds for each condition and replicate. We then average the raw outputs across these runs and renormalize to obtain the final class contribution values.

- L122-123: I don't understand this sentence.

We have rewritten the paragraph and rephrased this part in lines 133-138:

For comparative analysis, changes in CC between conditions, called relocalization (RL), range from -1 to 1. Using variation across ensemble runs, p-values and Cohen's D are calculated per compartment to derive two metrics: the relocalization score (RLS), summarizing total change (0–2), and the distance score (DS), ranking shifts by effect size. These global relocalization metrics make up the second C-COMPASS output (Extended Fig. 1D).

We have further added the mathematical formulas for calculating RLS and DS to the Methods section lines 827-860:

Protein relocalization

Relocalization events were analyzed through pairwise comparisons between conditions (groups) g_m and g_n with $m, n \in \{1, \dots, G\}$, where G is the total number of conditions. For each compartment c , the RL value was computed as the difference in predicted compartment association between the two conditions:

$$RL_c^{(s)} = CC_c^{(s,g_n)} - CC_c^{(s,g_m)},$$

where $RL_c^{(g_m,g_n)}$ represents the amount of relocalization for compartment c between conditions g_m and g_n , and CC_c is the predicted class contribution for compartment c .

To assess the statistical significance of relocalization events, we performed Welch's t-tests based on the neural network outputs $w_c^{(s,r)}$, which were obtained from multiple rounds and retraining runs. For each condition g and compartment c , this yielded a distribution of predicted scores:

$$w_c^{(s,g)} = \{w_{c,1}^{(s,g)}, w_{c,2}^{(s,g)}, \dots, w_{c,R}^{(s,g)}\}.$$

The Welch's t-statistic for each compartment was computed as:

$$t = \frac{\bar{w}_c^{(s,g_n)} - \bar{w}_c^{(s,g_m)}}{\sqrt{\frac{\sigma_c^{2(s,g_n)}}{R_n} + \frac{\sigma_c^{2(s,g_m)}}{R_m}}},$$

Where $\bar{w}_c^{(g)}$ is the mean predicted score, $\sigma_c^{2(g)}$ is the sample variance, and R_m, R_n are the number of runs for g_m and g_n , respectively.

The degrees of freedom ν were estimated using the Welch-Satterthwaite equation:

$$\nu_c^{(s)} = \frac{\left(\frac{\sigma_c^{2(s,gm)}}{R_m} + \frac{\sigma_c^{2(s,gn)}}{R_n}\right)^2}{\frac{\left(\frac{\sigma_c^{2(s,gm)}}{R_m}\right)^2}{R_m-1} + \frac{\left(\frac{\sigma_c^{2(s,gn)}}{R_n}\right)^2}{R_n-1}}$$

The resulting p-values were then calculated from the two-sided t-distribution with ν degrees of freedom, as implemented in the `scipy.stats.ttest_ind` function (with `equal_var=False`) from the SciPy Python package:

$$p_value = 2 \times (1 - CDF_{t_\nu}(|t|)),$$

where CDF_{t_ν} denotes the cumulative distribution function of the t-distribution with ν degrees of freedom.

In addition to statistical significance, we also calculated Cohen's d as a standardized effect size metric for each compartment c :

$$d_c^{(s)} = \frac{\bar{w}_c^{(s,gn)} - \bar{w}_c^{(s,gm)}}{\sigma_{spooled}}, \text{ where } \sigma_{spooled} = \sqrt{\frac{\sigma_c^{(s,gm)} + \sigma_c^{(s,gn)}}{2}}.$$

This metric accounts for both the magnitude of relocalization and the precision of the model predictions, providing a unified score to identify promising relocalization candidates.

To quantify the overall extent of relocalization between two conditions, we computed the ReLocalization Score (RLS) as the sum of the absolute relocalization values across all compartments:

$$RLS^{(s)} = \sum_{c=1}^C |RL_c^{(s)}|.$$

- L142-145: I find the discussion of the results in Fig3a rather dry. It's striking that there is a benefit of upsampling for some compartments and no benefit in others, why? Is it because LDs, peroxisomes, endosome have less markers? Or is it because the marker patterns are more difficult to grasp hence require more profiles? The fact that some compartments are more difficult to model than others, regardless of upsampling or not, is interesting information worth to be discussed, in my opinion. The following paragraph seems to provide part of the explanation. I'm also not convinced that scores are improved for mitochondria given the error bars, I wouldn't use it as an example.

We have addressed this comment in the results section and revised the text accordingly lines 156-179. We have clarified that upsampling primarily enhances the prediction accuracy for organelles with few available markers, without negatively affecting the predictions for organelles that are well-represented by many marker proteins. We also acknowledge that some compartments are inherently more challenging to resolve than others. This difficulty may arise from an overlap in protein profiles across compartments or from biological factors, such as the high dynamics and non-unique localization of proteins within certain organelles like lipid droplets, which we believe is the case here.

We first assessed the impact of upsampling to balance marker count discrepancies across compartments. UMAP visualization confirmed that upsampling maintained distinct clusters and preserved similarity between artificial and experimental profiles (Fig. 2C). Performance was evaluated using precision (correct predictions among all predicted), recall (correct predictions among actual proteins), and F1 score (harmonic mean of precision and recall) (Fig. 2D). Proteins from compartments with fewer markers benefited most, while those from well-represented compartments retained high F1 scores. For example, lipid droplet (LD) proteins, initially poorly predicted (F1 <0.01), improved to 0.50 post-upsampling. Mitochondrial proteins, with abundant markers, consistently showed high F1 scores (0.82–0.86) without upsampling. Overall, upsampling improved precision (from 0.44 to 0.60), recall (from 0.47 to 0.66), and F1 score (from 0.46 to 0.63).

To further evaluate C-COMPASS in organelle classification in detail, we analyzed the pre-filtered distribution of class predictions for marker proteins. These markers, typically localized to single compartments, showed high prediction values for their respective compartments and low values elsewhere, aligning with biological expectations (Extended Fig. 2A). Mitochondrial proteins, known for their distinct targeting signals, exhibited strong predictions for mitochondria and minimal misclassification (Fig. 2E), resulting in the highest accuracy among compartments. Among 583 mitochondrial markers, only two false positives were identified (Fig. 2F). In contrast, LD proteins exhibited lower prediction accuracy. This may stem from the limited resolution and overlapping profiles within the gradient, or from the inherent characteristics of LDs themselves, which are highly dynamic and often multi-localized. Many LD proteins localize to multiple compartments and show higher prediction scores for both the ER and cytosol, aligning with their known targeting pathways from either the Cytosol or the ER (CYTOLD and ERTOLD)²⁴ (Fig. 2E).

- L244: "with some compartments, such as the endosomes and Golgi apparatus, showing lower accuracy". Golgi shows good results in Fig5d.

Thanks for noting this error, we have corrected it.

- L252-254: Concluding that there are relocalization events requires more evidence. This statement assumes the cellular composition of the livers across conditions to be similar, eg maybe the diet condition may impact the proportion of adipocytes or immune cells in the liver. I'm interested in the author's perspective on this and feel it needs to be added to the discussion.

We fully agree with the reviewer that the potential influence of changes in cell composition on protein profiles is an interesting consideration and could complicate data interpretation. We have added a paragraph in the discussion to address this aspect. However, we also believe that this factor does not impact the part of our study concerning steatotic liver. It generally takes approximately 27-40 weeks for mice to develop inflammation associated with changes in cell populations¹¹, a duration we do not reach in the current study.

Addition to discussion lines 557-560:

Finally, confounding variables, such as shifts in cell-type composition in tissues under specific conditions (e.g., steatotic liver), can alter organelle behavior during fractionation and impact the reliability of protein localization, further emphasizing the need for context-aware interpretation of results.

- L374-375: the sentence should reference Fig S8e.

Thanks for noting the mistake, we have corrected it.

- L568: literature and theory has shown that imputation by zero is the worst strategy for MS-based proteomics data. Could the authors comment on this? Also, why using another strategy for the total proteome?

We agree with the reviewer that zero imputation is generally not advisable for total proteome analyses, as missing values in this context often reflect low-abundance proteins rather than complete absence. Imputing such values with zeros would exclude these proteins from downstream statistical analyses and lead to biases when comparing conditions, especially in cases where division or fold-change calculations are involved. For this reason, missing values in total proteome datasets are imputed with small, non-zero values sampled from a left-shifted distribution, representing low but non-zero abundance.

However, our dataset is also based on subcellular fractionation profiles. In this context, the interpretation of missing values differs substantially. Each protein profile represents the relative distribution of the protein across gradient fractions, and the presence or absence of a signal in a given fraction carries spatial meaning. A zero value in a specific fraction suggests that the protein is absent from the organelle that predominantly sediments in that fraction. Hence, imputing missing values with zero in this context is not only methodologically appropriate but also biologically meaningful.

Moreover, unlike total proteome analyses, gradient fractionation data do not require all values to be non-zero to enable valid statistical comparisons. Zero values can be integrated into profile-based analyses without mathematical issues, such as division by zero. Adding small, non-zero imputed values would not provide additional biological insights and could even introduce noise by artificially inflating the presence of proteins in fractions where they are likely absent.

To reflect this reasoning, we have clarified our imputation strategy in the methods section (lines 706-722):

Proteome fractionation data processing

Fractionation abundance data were imported into C-COMPASS, with samples assigned to condition (group), replicate r , and fraction f . Proteins identified in at least two replicates per condition were retained, and missing values were replaced with zeros. Unlike standard proteomic analyses, where missing values typically indicate low-level expression, fractionation data reflect compartmental localization. A protein absent from a given fraction likely indicates its absence from that compartment rather than low expression. Therefore, imputing missing values with zeros is more appropriate in this context. To ensure comparability, the profiles were normalized so that the total sum across fractions f equals 1:

$$\bar{p}_f^{(s,r)} = \frac{p_f^{(s,r)}}{\sum_{f=1}^F p_f^{(s,r)}}$$

where $\bar{p}_f^{(s,r)}$ represents the normalized abundance of $p_f^{(s,r)}$. This normalization method is applied in the analysis performed by C-COMPASS. However, for visualization purposes, min-max scaling as normalization was used to rescale values into the range [0,1]:

$$\bar{p}_f^{(s,r)} = \frac{p_f^{(s,r)} - \min_f p_f^{(s,r)}}{\max_f p_f^{(s,r)} - \min_f p_f^{(s,r)}}$$

This alternative normalization ensures a more intuitive representation of the profiles in plots.

- L571: "Mean intensity values were used for analysis." Mean across what?

The mean intensity values were calculated across replicates. We have added this to the Methods section line 864.

- L599-600: Is there a safeguard when the difference between the number of markers between most and least abundant compartment is substantial? In other words, are there some compartments that are automatically dropped if not enough markers?

There is no automatic drop of compartments since the number of necessary marker proteins can vary for different experiments. Users can list the marker batch sizes in C-COMPASS for all found compartments and decide to drop specific compartments based on the proportion compared to the other batches.

- L606: two-compartment profiles are simulated at 3 mixing ratios, each represent 5% of the batch size. Could the authors comment why 5%? From the results presented, at least 30% of the proteins (and about 75% of the lipids) in the data sets seem to display multiple compartment profiles.

We thank the reviewer for this point. Indeed, our analysis suggests that in the order of 30% of the profiles are mixed. Yet, at the time of training, this was not known and it is in our opinion beneficial to avoid too much refinement for a proper assessment of the prediction power.

To understand the impact of the training set composition of the outcomes, a comprehensive testing would be necessary. This is considered beyond the scope of the work, yet, we mention this in the revised discussion. However, we hypothesize that this does not substantially impact training performance, as profiles corresponding to individual organelles remain sufficiently distinct to enable accurate classification.

We have added to the discussion lines 550-555:

Another consideration is the representation of multi-localized proteins in the training data. While our analysis suggests that approximately 30% of the profiles may represent mixed localizations, only 5% of the training data reflected such cases. Nevertheless, we hypothesize that this does not significantly impair training performance, as profiles corresponding to individual compartments remain sufficiently distinct for accurate classification.

Remarks on code availability:

See concerns #2 and #3 in the main review box.

Please see our responses for #2 and #3

Below, we present our point-by-point response to each reviewer comment. Reviewer comments are shown in teal boxes, our corresponding manuscript changes in yellow boxes, with the revised text highlighted in red. To ensure clarity and context, the unchanged surrounding text is shown in black.

Reviewer #1:

The manuscript by Natalie Kramer and coworkers presents C-COMPASS, an open-source software tool that utilizes neural network-based regression models to predict the spatial distribution of proteins and lipids across cellular compartments. The tool can analyze proteomics data obtained from organellar fractionation methods such as Protein Correlation Profiling (PCP) or Localization of Organelle Proteins by Isotope Tagging (LOPIT). The tool, which is provided as a user-friendly graphical interface, requires input data on protein abundances across fractions and a set of predefined or custom organelle marker proteins. It was developed with the aim of overcoming some of the limitations of existing proteomics-based organellar profiling methods, which often struggle with proteins that localize to multiple compartments as well as with the integration of protein and lipid data

Response:

We sincerely thank the reviewer for the positive evaluation and constructive feedback. We appreciate the acknowledgement of our efforts in benchmarking C-COMPASS against other available tools, as well as the assessment of fraction number and separation resolution impacts. We are particularly pleased that the reviewer finds our GitHub repository and associated tutorials valuable.

Regarding the request for additional explanation on biologically interpreting outcomes generated by C-COMPASS, we agree this is a critical aspect. We included the following, more detailed explanation of interpreting these results within our GitHub tutorials which already provided in-depth discussions on datasets:

Class Contribution (CC) values represent the proportion of a protein assigned to each compartment.

And:

Relocalization Values (RL) quantify how the localization of a protein changes in each compartment between two conditions. A positive RL value indicates greater localization in the second condition compared to the first, while negative values indicate reduced localization.

We explicitly referenced these resources within the manuscript itself (L134-135):

Detailed descriptions of all C-COMPASS output values are available in the online documentation.

In their benchmark against other tools, the authors claims that C-COMPASS shows better performance for predicting dual and multiple locations. However, I failed to understand how the prediction error is calculated in these cases. Should the user provide a list of predefined proteins with multiple locations such that the true positives and false positives can be properly assessed in those cases?

Response:

We thank the reviewer for pointing out that our explanation of how C-COMPASS's performance in predicting dual and multiple localizations was evaluated is not sufficiently clear. In the revised manuscript, we clarified that, due to the lack of reliable databases or facilitated lists of proteins with confirmed multiple localizations, performance assessments for multi-localized proteins were conducted using simulated dataset. The simulated data were constructed from the profiles of single-compartment marker proteins.

A detailed explanation and clarification were added in the Results section (L231-238):

Assessing prediction quality from experimental datasets is challenging, since the ground truth for localizations is typically not available, especially for multiple localization patterns. Therefore, to estimate performance, we used a theoretical model that reflects ideal experimental behavior. We performed simulations in which multi-localization profiles are generated by mixing compartment-specific features in defined ratios. These simulations allow us to evaluate C-COMPASS's performance under controlled conditions. For real datasets, we assume comparable accuracy, provided that the experimental data quality is optimal.

From this revised version, I can understand how the relocalization score can be used to identify relocation events. Is the magnitude of this value indicative of the magnitude of the event? Does the program provide an accompanying statistical value for this? Moreover, does the program indicate between which compartments the protein relocation is happening?

Response:

We thank the reviewer for raising this point. Our in-silico tests indicate that the relocalization values reflect the percentage of localization change per organelle and that the total Relocalization Score (RLS) summarizes absolute changes across all compartments. To support interpretability, we also provide a statistical p-value based on variability across predictions from an ensemble of neural network models. In the revised manuscript, we clarify that the relocalization values include organelle-specific directionality, where positive or negative values indicate increasing or decreasing localization proportions for a given protein between two conditions for all organelles independently. Therefore, the relocalization values are describing changes in the spatial distribution of a protein rather than a discrete movement of a protein.

We added the following section (L118-124):

In cases where only two compartments are involved, the direction of relocalization can be clearly determined based on opposing shifts in localization values. For more complex cases involving multiple compartments, only changes in spatial distribution between conditions can be assessed. While a movement score can approximate source and target compartments, it provides a broader overview rather than tracking distinct molecular transitions.

In the lines 248 to 253 the authors claim: “Furthermore, a reorganization of the Pi3K pathway, known to be activated by LPS32, was observed. In concordance with the original study using HyperLOPIT, which utilized a T-Augmented Gaussian Mixture Model with Bayesian computation performed using a Markov Chain Monte Carlo (TAGM-MCMC) model for protein localization and relocalization, C-COMPASS detected a high number of translocation events, especially from the lysosome to the plasma membrane and from the cytosol to the nucleus (Fig. 3G), confirming its ability to detect known and validated protein relocalizations under specific treatment conditions.” How does the detecting of more relocalization events confirm that their software can detect known and validated protein relocalization events? The authors should provide examples to prove this claim.

Response:

We agree that detection alone does not confirm correctness. We revised the statement to clarify that C-COMPASS identifies relocalization patterns consistent with those reported in the original study, thus supporting the reliability of the proposed approach. We also had to apply additional changes in this section due to the limitation of words. The revised paragraph is now as follows (L214-223):

We further validated C-COMPASS for identifying protein relocalization. We filtered proteins with a Relocalization Score (RLS) >1 after LPS treatment, indicating a 50% change in organelle assignment, and plotted the C-COMPASS distance metric against the RLS p-value to identify highly reliable hits (Extended Fig. 3G). These proteins were enriched in pathways known to be activated by LPS, including, MHC1³⁰ and the unfolded protein response³¹ in addition to vesicular transport and cytoskeletal remodeling described in the original study (Fig. 3F). In concordance with the original study, C-COMPASS detected a high number of relocalization events, especially from the lysosome to the plasma membrane and from the cytosol to the nucleus (Fig. 3G). These consistent patterns support the reliability of C-COMPASS in detecting biologically validated relocalization events under specific treatment conditions.

For completeness, we also provide a comprehensive list of predictions from the original study and our results in the supplementary table 02.

In lines 499 to 501 the authors have rephrased their initial text to “Here, we show that C-COMPASS can extend beyond proteomics and co-integrate lipidomics to co-map protein and lipid compositions of organelles. In the future, the pipeline could also be used for integration of additional omics layers such as metabolomics or transcriptomics” This statement implies that C-COMPASS can be fully compatible with other omics layers. I can understand that from a metabolomics perspective a similar data matrix as in proteomics or lipidomics can be retrieved from subcellular fractionation experiments. However, is that also the case for transcriptomics? Does it make biological sense to perform subcellular fractionation at that level?

Response:

We thank the reviewer for the critical assessment. While it has been shown that the spatial distribution of transcripts within the cell is functionally relevant and of biological interest¹⁻³, the interpretability of the integrated data is indeed difficult. Hence, while the software is in principle adaptable to any type of analysis of omic-level, we removed the following part regarding transcriptomics from the manuscript:

Reviewer #2:

I am impressed by the dramatic improvement of the manuscript. All my previous comments regarding manuscript clarity have been thoroughly addressed as well as my additional minor comments. I am most impressed by the improved quality of the software, which I previously could not even install and now could fully run from start to end without errors. I can attest of the user-friendliness of the software and acknowledge the efforts taken to improve the long-term maintainability of the code. Below are still a few minor comments that should be quick to fix.

Response:

We sincerely thank the reviewer for the positive evaluation and the additional comments.

L114-116: "To assess this in an unbiased manner, we recommend performing supervised hierarchical clustering followed by GO enrichment analysis prior to applying C-COMPASS." I find this sentence too short to be informative and therefore superfluous.

Response:

We agree that the sentence in its previous form did not provide sufficient guidance. In the revised version, we replaced it with the following statement (L99-101):

In case an individual marker list is needed, we recommend creating it based on published, experimentally validated datasets (see Supplement).

In the supplementary material, we have included a more detailed description outlining how such marker lists can be generated using hierarchical clustering followed by GO-term and organelle marker enrichment analysis, following an approach that has already been applied in previous publications^{4,5} (Supplementary_Methods.docx):

Procedural suggestion for generating a custom marker list

In certain cases, it may be necessary to generate a custom list of spatial marker proteins, particularly when reference data for specific tissues, organisms, or biological conditions is unavailable. However, it is possible to derive marker proteins that reflect compartment-specific profile patterns directly from the dataset itself. To do so, we recommend the following pre-processing steps on the gradient data: 1) Impute missing values with zero. 2) Apply MinMax scaling to normalize all protein profiles to a range between 0 and 1.

Next, perform hierarchical clustering across the protein dimensions (rows) only, without clustering the fraction dimension (columns). After clustering, identify protein clusters that are enriched for annotations associated with specific organelles or compartments. We recommend using Gene Ontology Cellular Component (GOCC) annotations for this purpose. This enrichment can be assessed manually or via algorithmic approaches that identify clusters with the strongest enrichment for the compartments of interest. If performed manually, we recommend prioritizing clusters with distinctive profile patterns over large cluster sizes.

The resulting set of proteins can then be compiled into a marker list, with each protein annotated according to the enriched compartment. To be used in C-COMPASS, this list must contain at least two columns: one serving as an identifier (e.g. gene names), and one indicating the assigned compartment annotation. A comprehensive list of marker proteins used in our experiments can be found in the supplementary material.

Below, we provide a list of publications in which this strategy has been successfully applied to derive compartment-specific marker proteins:

We also added a comprehensive list of valuable experimentally validated resources for organelle markers (Markerlist_Haas_et_al.xlsx). This is the same marker list that we used for our analyses.

L703-706: this section seems obsolete or out of place.

Response:

We agree and removed this section from the manuscript:

~~In the following sections, we use this basic framework to describe how we address class imbalance (“Upsampling”), design and train of our neural network (“Neural Network Architecture” and “Neural Network Optimization”), and finally perform filtering and output processing (“Output Filtering and Final Network Output Processing”).~~

L711-714: I do not quite agree with the answer regarding data imputation. In MS experiments, a protein is either detected or not detected in a sample. Whether the protein is not detected because it is truly absent in the sample or because of technical reasons is difficult to assess. Replacing all missing values by zero assumes the absence of technical missingness, which I highly doubt happens even in a fractionation context. However, as missing data management is outside the scope of this work and I do not foresee a dramatic impact on this manuscript's conclusions, I put this comment for the track record and do not require action by the authors.

Response:

We thank the reviewer for the valuable comment. In the revised manuscript, we mention in the Discussion that the handling of missing values with C-COMPASS might be further improved in the future but was outside the scope of this manuscript (L458-460):

C-COMPASS currently uses zero imputation for missing values, which we consider appropriate for fractionation data, though future versions could support alternative strategies.

L832-833: inconsistency in notation either it is $RL^{\{(s)\}}_c$ or $RL^{\{(g_m, g_n)\}}_c$

Response:

We thank the reviewer for the careful reviewing. We changed the equation to (L1043-1045):

$$RL_c^{(s, g_n, g_m)} = CC_c^{(s, g_n)} - CC_c^{(s, g_m)}$$

here $RL_c^{(s, g_m, g_n)}$ represents the amount of relocalization for compartment c between conditions g_m and g_n , and CC_c is the predicted class contribution for compartment c .

L841-842: inconsistency in notation either it is $\bar{w}_c^{(g)}$ or $\bar{w}_c^{(s,g)}$

Response:

This is indeed inconsistent and correctly formulated in former L841. Accordingly, we revised former L842 and changed the sentence to the following (L1053-1054):

where $\bar{w}_c^{(s,g)}$ is the mean predicted score, $\sigma_c^{2(s,g)}$ is the **squared** sample variance, and R_m , R_n are the numbers of runs for g_m and g_n , respectively.

L854: spooled or pooled?

Response:

We thank the reviewer for pointing out this typo; the correct naming is 'pooled'. While reviewing the equation, we also noticed an additional missing square term. We revised the equations as follows (L1065):

$$d_c^{(s)} = \frac{\bar{w}_c^{(s,g_n)} - \bar{w}_c^{(s,g_m)}}{\sigma_{pooled}}, \text{ where } \sigma_{pooled} = \sqrt{\frac{\sigma_c^{2(s,g_m)} + \sigma_c^{2(s,g_n)}}{2}}.$$

L871 & L874: inconsistency in notation, $CC_c^{(s,g)}$ should be used instead of $CC_c^{(c,g)}$. Same goes for SA.

Response:

The notation is indeed incorrect here. We revised the equations to the following (L1082-1085):

$$SA_c^{(s,g)} = CC_c^{(s,g)} \times \bar{I}_s^{(g)}.$$

To estimate the total abundance of each compartment in a given condition, these values were summed across all species s per compartment c :

$$CA_c^{(g)} = \sum_{s=1}^S SA_c^{(s,g)},$$

Note that I may have missed additional inconsistencies.

Response:

We systematically checked and revised all equations to ensure consistency throughout the manuscript, including points mentioned by the reviewer and any additional issues we found.

L980-984: could you add a brief sentence in the methods that describes how you identify false positives, false negative and true positive for experimental data? While I am satisfied with your answer to my previous comment, I feel this could be a bit more explicit for the reader as well.

Response:

To include a clear explanation of how these values were identified, we revised the Methods by replacing the following sentence (former L983):

~~where TP_ϵ is the number of true positives, FP_ϵ is the number of false positives, and FN_ϵ is the number of false negatives per compartment c .~~

by (L1194-1199):

For each compartment c , TP_c is the number of true positives (species correctly predicted to compartment c), FP_c is the number of false positives (species predicted to compartment c but originally annotated to a different compartment $\neq c$), and FN_c is the number of false negatives (species originally annotated to compartment c but predicted to a different compartment $\neq c$). The predicted compartment for each marker species was determined by selecting the compartment with the highest $CC_c^{(s)}$ value.